

# A comprehensive biomass burning emission inventory with high spatial and temporal resolution in China

Ying Zhou[1,2], Xiaofan Xing[1,2], Jianlei Lang[1,2], Dongsheng Chen[1,2], Shuiyuan Cheng[1,2,3], Lin Wei[1,2], Xiao Wei[4], Chao Liu[5]

[1]Key Laboratory of Beijing on Regional Air Pollution Control, Beijing University of Technology, Beijing 100124, China
[2]College of Environmental & Energy Engineering, Beijing University of Technology, Beijing 100124, China
[3]Collaborative Innovation Center of Electric Vehicles, Beijing 100081, China
[4]Beijing Municipal Research Institute of Environmental Protection, Beijing 100037, China
[5]Environmental Meteorological Center of China Meteorological Administration, Beijing 100081, China

*Correspondence to*: Ying Zhou (y.zhou@bjut.edu.cn) and Shuiyuan Cheng (chengsy@bjut.edu.cn)

**Abstract.** Biomass burning injects many different gases and aerosols into the atmosphere, which could have a harmful effect on air quality, climate change and human health. In this study, a comprehensive biomass burning emission inventory including crop straw domestic combustion and in field burning, firewood and livestock excrement combustion, forest and grassland fire was developed for mainland China in 2012 based on county-level activity data and updated source-specific emission factors (EFs). The emission inventory within $1 \times 1$ km grid was generated using geographical information system (GIS) technology according to source-based spatial surrogates. A range of key information related to emission estimation (e.g., province-specific proportion of crop straw domestic burning and open burning, detailed firewood combustion quantities, uneven temporal distribution coefficient) was obtained from field investigation, systematic combing of the latest research and regression analysis of statistical data. The established emission inventory includes the major precursors of complex pollution, greenhouse gases and heavy metal released from biomass burning. The results show that the emissions of $SO_2$, $NO_x$, $PM_{10}$, $PM_{2.5}$, VOC, $NH_3$, CO, EC, OC, $CO_2$, $CH_4$ and Hg in 2012 were 332.8 Gg, 972.5 Gg, 3676.0 Gg, 3479.4 Gg, 3429.6 Gg, 395.8 Gg, 33987.9 Gg, 367.1 Gg, 1151.7 Gg, 665989.0 Gg, 2076.5 Gg and 3.65 Mg, respectively. Indoor and outdoor burning of straw and firewood combustion are identified as the dominant biomass burning sources. The largest contributing source is different for various pollutants. Straw indoor burning is the major source of $SO_2$, CO, $CH_4$ and Hg emission; firewood contributes most to EC and $NH_3$ emission. Corn, rice and wheat represent the major crop straws, with their total emission contribution exceeding 80% for each pollutant. Corn straw burning has the greatest contribution to EC, $NO_x$ and $SO_2$ emissions; rice straw burning is dominant contributor to $CO_2$, VOC, $CH_4$ and $NH_3$ emissions. Heilongjiang, Shandong, and Henan provinces located in northeast and central-south region of China have higher emissions. Gridded emissions, which were obtained through spatial allocation based on the gridded rural population and fire point data from emission inventory at county resolution, could better represent the actual situation. Higher biomass burning emissions are concentrated in the areas with greater agricultural and rural activity. The temporal distribution shows that higher emissions occurred in April, September, and October during the whole year. There's regional difference in monthly variation due to the diversity of main planted crop and the climate





conditions. Furthermore, $PM_{2.5}$ component results showed that OC, $Cl^-$, EC, $K^+$, $NH_4^+$, K element and $SO_4^{2-}$ are the main $PM_{2.5}$ species accounting for 80% of the total emissions. The species with relatively higher contribution to VOCs emission including ethylene, propylene, toluene, mp-xylene and halocarbons which are key species for the formation of secondary air pollution. The detailed biomass burning emission inventory generated by this study could provide useful information for air quality

modelling and support the development of appropriate pollution control strategies.

**Keywords:** Biomass burning; Emission inventory; High resolution; Species

## 1 Introduction

Biomass burning is considered a significant source of gas and particulate matter (PM), resulting in a major impact on atmospheric chemistry, climate change and human health. Active trace gases (e.g., $SO_2$, $NO_x$, VOCs, $NH_3$) released from

biomass burning are the major precursors of secondary inorganic/organic aerosols and tropospheric ozone ($O_3$) in the atmosphere (Penner et al., 1992; Kaufman and Fraser, 1997; Koppmann et al., 2005; Langmann et al., 2009). Several studies have indicated that observed local and regional air pollution could be attributed to the chemical species emitted from biomass burning (Huang et al., 2012b; Zha et al., 2013; Cheng et al., 2014; Yan et al., 2014; Zong et al., 2016). The emission factor (EF) of some biomass burning pollutants is even greater than coal burning, which is widely recognized as a major pollution

source (Zheng et al., 2009;Fu et al., 2013). Primary particles (e.g., BC and OC) discharged by biomass burning not only impact visibility, but also have an influence on climate change due to the positive effects of the absorption of light and cloud condensation (IPCC, 2011). Biomass burning is also a critical source of greenhouse gases such as methane ($CH_4$) and carbon dioxide ($CO_2$) (Andreae and Merlet, 2001), which contribute to global warming (Sun et al., 2016). Moreover, several reports (Fernandez et al., 2001; Huang et al., 2012b; Shi and Yamaguchi, 2014) reveal that the long-term or short-term exposure to

PM (e.g., BC emitted from indoor biomass burning) can cause adverse effects to human health, such as decreased lung function, increased respiratory diseases and lung cancer mortality. Furthermore, studies have identified that indoor biomass burning could bring adverse health effects on residents (Jiang and Bell, 2008; Fullerton et al., 2008).

Prior to its rapid economic development, China was a largely agricultural country and thus once consumed a large amount of biofuels (e.g., crop residues and firewood). With the dramatic urbanization that accompanied the economic development,

the pattern of energy consumption in rural areas has been gradually transformed. In particular, in some agricultural areas with relatively high income, crop residues were more burned directly in the field (Sun et al., 2016). Beginning in 1999, the Chinese government has issued a series of laws and regulations to ban the open burning of straw residues and to encourage straw comprehensive utilization, such as returning to field, livestock feeding, industrial raw materials manufacturing, briquette fuel processing, etc. (MEP, 1999). However, the effect of this legislation was not satisfactory because the processes of straw

comprehensive utilization not only required high labor costs but also delayed sowing of the next crop. Thus, the phenomenon of straw outdoor burning continued to occur. The amount of straw outdoor burning in China in 2009 is 0.215 billion tons (MA, 2011). Accordingly, a comprehensive and detailed emission inventory of biomass burning representing the current status in



China, is important to provide valuable information for researchers and policymakers. Examples of potential applications include research to understand the influence of biomass burning on indoor air quality and the outdoor atmospheric environment, and the development of effective management decisions to relieve the environmental burden and reduce health risk.

Since the early research conducted by Crutzen et al. (1979), a series of efforts have been made to develop a biomass burning emission inventory, especially in developed countries (Reddy and Venkataraman, 2002; Ito and Penner, 2004; van der Werf et al., 2006; Nelson et al., 2012; Shon, 2015). Compared with the developed countries, research by Chinese scientists on this issue started relatively late. The initial studies on biomass burning emission inventory across China (Streets et al., 2001; Tian et al., 2002; Streets et al., 2003; Cao et al., 2005) or in certain regions (Zheng et al., 2009; Huang et al., 2011) were developed mainly based on EFs developed for foreign nations (Turn et al., 1997; Andreae and Merlet, 2001; U.S. EPA, 2002) because of the lack of local measurements in China. However, this approach could introduce relative great uncertainty in emission estimates because of the differences in crop types and the combustion conditions between China and other counties.

In recent years, various research activities have focused on the emission characteristics of biomass burning in China, including local EF and chemical species profile tests. Li et al. (2007b) and Li et al. (2009) conducted field measurements to determine the EF for several of the main household biofuels in Beijing, Chongqing, Henan and Shandong. Li et al. (2007c) determined the EF for wheat and maize burning in field and Cao et al. (2008) measured EFs for the indoor burning of rice straw, wheat straw, corn stovers and cotton stalks. Zhang et al. (2008) measured $CO_2$, CO, NO, $NO_2$, $NO_x$ and PM EFs of rice, wheat and corn straw and Wang et al. (2009) launched a study on characteristics of gaseous pollutants from biofuel stoves in China. More recently, Zhang et al. (2013b) carried out experiments on EFs for open burning of sugar cane leaves and rice straw in southeast China. Ni et al. (2015) conducted laboratory burn tests to determine the EFs of wheat straw, rice straw and corn stalks, considering the impacts of the fuel moisture content.

Based on the local EFs, emission inventories that focused on certain provinces (Li et al., 2015; He at al., 2015) or city group regions (He at al., 2011; Fu et al., 2013) were developed. In our previous study, we reported an emission inventory with high resolution in the Beijing–Tianjin–Hebei region of China (Zhou et al., 2015). To produce a national emission inventory, several studies of biomass burning have been carried out without distinguishing the detailed crop straws (Lu et al., 2011; Yan et al., 2006; Tian et al., 2011). Moreover, there are several studies that have focused on certain pollutant (Huang et al., 2012d; Chen et al., 2013; Zhang et al., 2013a; Kang et al., 2016; Li et al., 2016), and certain crop straws (Zhang et al., 2008; Hong, et al., 2016; Sun, et al., 2016). In recent years, the comprehensive biomass emission inventory is limited. Most of recent studies are concentrated upon biomass open burning, including the multi-year trend analysis on certain or multiple pollutants (Wang and Zhang, 2008; Song et al., 2009; Huang et al., 2012c; Shi et al., 2014; Shon, 2015; Xu et al., 2016; Zhang et al., 2016). Few studies have covered recent firewood burning (see next paragraph for details regarding the reason for this). In addition to the EF, detailed activity data are also important for a reliable emission inventory, such as straw domestic or in field burning ratios, which are not currently publicly available. Gao et al. (2002) produced a study on the percentage of straw used as fuel and for direct incineration in 2000. Wang et al. (2008) investigated the percentage of crop open burning in 2006 of six regions in China, which were divided according to the similarities of agriculture, climate, economy and region. Tian et al. (2011) estimated the



proportion of crop straw domestic burning and open burning in 2007 for seven and three regions of China, respectively. Thus, there is limited information about the ratio of straw used as fuel to that burnt in the field that reflects the status of China in recent years for different provinces. Moreover, because of the lack of firewood consumption in the energy statistical yearbook after 2007, there are few reports containing a comprehensive biomass burning emission inventory for China.

5 Consequently, we have identified several weaknesses in the current biomass burning emission inventories. First, not all biomass burning sources have been included in recent years, especially after 2007, because of the lack of firewood consumption data in the statistical yearbook. Second, the source-specific EFs used in emission estimation need to be updated based on the systematic combing of local tests in the latest research. Third, the proportion of crop straw domestic burning and open burning, which could reflect the recent situation of different provinces in China needs to be investigated. Fourth, the current biomass 10 burning emission inventory for China is generally at province resolution because detailed activity data cannot be directly obtained from the yearbook. Activity data at coarse resolution are likely to be associated with greater uncertainty in grid emissions generated according to source-based gridded spatial surrogates (e.g., population) using GIS technology (Zheng et al., 2014). As a result, it is of great importance to develop an integrated and model-ready biomass burning emission inventory with high spatial and temporal resolution.

15 In this study, a comprehensive biomass burning emission inventory including crop straw domestic combustion and in field burning, firewood and livestock excrement combustion, forest and grassland fire was developed for the Chinese mainland (excluding Hong Kong, Macao, and Taiwan) in 2012, based on detailed activity data. In addition, we attempt to take full account of the source-specific EFs measured in China. A range of important information for emissions estimation (e.g., province-specific straw domestic combustion/in field burning ratio, detailed firewood combustion quantities and uneven 20 temporal distribution coefficient) were obtained from a field investigation, systematic combing of latest research and regression analysis of statistical data. A 1-km resolution emission inventory was generated using GIS software. The gaseous and particulate pollutants examined in this research included $SO_2$, $NO_x$, $PM_{10}$, $PM_{2.5}$, VOC, $NH_3$, CO, EC, OC, $CO_2$, $CH_4$ and Hg, covering the major precursors of complex pollution, greenhouse gases and heavy metals released from biomass burning. The detailed emission inventory given by this paper could provide valuable information to support the further biomass burning 25 pollution research and the development of a targeted control strategy of all regions across the Chinese mainland.

The remainder of this paper is structured as follows. Section 2 describes the methodology including the emission estimation method, the selection and handling of activity data and corresponding parameters, determination of EFs, spatial and temporal allocation and speciation of $PM_{2.5}$ and VOCs. Section 3.1 describes the total emission in China, and the contribution of various biomass burning sources and crop straws. Section 3.2 describes the emission from different regions, and contributions of 30 different biomass sources and crop straws of each province. Spatial and temporal distribution of biomass burning emissions is discussed in Secs. 3.3 and 3.4, respectively. Section 3.5 presents the emissions of $PM_{2.5}$ and VOC species. Uncertainty in biomass burning emission estimates is described in Section 3.6. The comparison between this study and other studies appears in Section 3.7. Section 4 summarizes the conclusions.





## 2 Methodology

### 2.1 General description

The biomass burning considered in this study is mainly divided into two categories including domestic combustion and in field burning. Domestic combustion mainly involves crop straw, firewood and livestock excrement (mainly used in pastoral and semi-pastoral areas). In field burning includes seasonal crop residue waste burning, grassland and forest fires. Details of the source classifications are shown in Table 1.

A bottom-up approach was used to develop the biomass burning emission inventory for all districts or counties. The annual biomass burning emissions ($E_i$) were calculated using Eq. (1) as follows:

$$E_i = \sum (A_i \times EF_{i,j})/1000, \tag{1}$$

where subscripts i and j represent the type of pollutant and biomass burning source; E is the annual typical pollutant emission (Mg/a); A is annual amount of dry biomass burned (Mg/a), for which the detailed calculation method is shown in Sec. 2.2; and EF is the emission factor (g/kg), for which a detailed description is presented in Sec. 2.3.

### 2.2 Activity data

#### 2.2.1 Straw burning

The burning mass of straw indoor and outdoor burning can be calculated using Eq. (2) as follows:

$$A_{i,k} = P_{i,k} \times N_k \times R_{i,k} \times D_k \times CE_k, \tag{2}$$

where subscripts i and k represent region (district or county) and crop type, respectively; $A_{i,k}$ is the annual burning mass of crop straw (ton/year); $P_{i,k}$ is the amount of crop-specific yields per year (ton/year); $N_k$ is the residue-to-production ratio of each straw type (ton/ton); $R_{i,k}$ is percentage of crop straw burned as fuel or in field burning; $D_k$ is dry matter fraction of each straw type; and $CE_k$ is the combustion efficiency of each straw type.

There are currently no statistics on the amount of each crop yield at the county resolution ($P_{i,k}$) in the statistical yearbook. Therefore, in this study, we conducted a correlation analysis between grain yield and crop yield at prefecture resolution, and found a good correlation ($R = 0.747$, detailed analysis is provided in the Supplement, Fig. S1). Next, the $P_{i,k}$ was calculated based on the various types of crop yield at prefecture resolution and grain yield at county resolution, which was summarized from a range of statistical yearbooks in 2012 for each province (including the city yearbooks which are publically available for some cities), NBSC (2013a) and NBSC (2013b). The total straw amount of China in 2012 calculated in this study is 832.5 Tg, which is similar to NDRC (2014) (817.4 Tg). The map at prefecture and county resolution is shown in Fig. S2 in the Supplement.

The variable $R_{i,k}$ is important for biomass burning emission estimation, and the information representing the recent status in China needs to be updated because of the continued economic development and the gradual implementation of national control policies for straw residue open burning. In this study, we conducted a detailed investigation of recent literature to collect the





percentage of crop straw burned as domestic fuel and burned as waste for each province. For some provinces where the current reporting is limited (e.g., Heilongjiang, Zhejiang, Guangdong, Inner Mongolia, and Hebei), a questionnaire survey was launched. Details of the questionnaire survey are presented in the Supplement (S3). The percentage of crop straw indoor and outdoor burning for each province is summarized in Table 2. According to our estimation, the amount of straw indoor and

outdoor burning for China in 2012 was 0.26 billion tons and 0.19 billion tons, respectively, which is similar to other recently published results for 2012 (0.26 billion tons indoor burning, Tian et al., 2014) and 2009 (0.215 billion tons outdoor burning, MA, 2011).

The $N_k$, $D_k$ and $CE_k$ values were obtained according to the literature collection. Detailed parameters used in this study are summarized in Table 3.

**2.2.2 Firewood**

Firewood consumption is recorded as non-commodity energy in the China energy statistical yearbook. However, detailed firewood consumption has not been publicly available since 2007. For more recent years, we obtained the total firewood consumption for China in 2012 and for each province in 2010 (Tian et al., 2014; IEA, 2012). However, these data could not support the development of an emission inventory at high resolution. There are several detailed statistics available in the

yearbook, such as the rural population, gross agricultural output and timber yield, which are likely to have a relationship with the firewood consumption. Therefore, we produced a correlation analysis between the three statistics and the firewood consumption of each province for different years in which the firewood consumption data were available, as shown in Fig. 1. The best correlation relationship was found between rural population and firewood consumption. The correlation coefficient for the different years ranged from 0.66 to 0.82. The firewood consumption at county resolution was obtained based on the

rural population at county resolution and the total firewood consumption reported by Tian et al. (2014) and IEA (2012). China's rural population, gross agriculture output and timber yield of each province come from NBSC (1999-2008a). Firewood consumption comes from NBSC (1999-2008b).

**2.2.3 Biomass burning of forest/grassland fires**

The burning mass of forest/grassland can be calculated using Eq. (3) as follows:

$A = AR \times B \times \eta,$                                                   (3)

where A is the annual burning amount of forest/grassland (ton/a); AR is the damaged area of grassland or forest fire per year ($hm^2$/a); B represents the dry biomass density of grassland or forest (ton/$hm^2$); and η is the combustion efficiency of grassland or forest.

The damaged area of forest fire and grassland fire could be obtained from NBSC (2013c) and NBSC (2013d), respectively.

The specific locations at which fire occurred can be determined according to Moderate Resolution Imaging Spectroradiometer (MODIS) fire point data; the B and η of forest is different from the climatic zones, which is shown in Table 4; B for grassland





is 1800 kg/ha (Lu et al., 2011), and η for grassland is 80.0% according to the 'Guide for the Preparation of Atmospheric Pollutant Emission Inventories for Biomass Burning' (EPD, 2014).

### 2.2.4 Livestock manure

The mass of biomass burned by animal waste was calculated using Eq. (4) as follows:

$$A = S \times Y \times C \times R, \tag{4}$$

where, A is the annual discharge of livestock manure burned (ton/a); S represents the amount of each livestock type in pastoral and semi-pastoral land at the end of the year (head/a); Y is a single livestock annual fecal output per year (ton/head); C represents livestock manure dry matter fraction; and R is the proportion of total livestock manure directly combusted.

The S values were taken from the EOCAIY (2013) and NBSC (2013c). The Y values were related to the large animals only. Among these, single cattle annual manure output was 10 tons and single horse annual manure output was 7.3 tons (Li and Zhao, 2008). The livestock annual manure output of other animals was set at 8 tons, according to Tian et al. (2011). The C value was set as 18% (Tian et al., 2011) and R was 20% (Li, 2007a; Liu and Shen, 2007). Since not all regions use livestock manure in biomass burning, we consider only the pastoral and semi-pastoral areas including Tibet, Inner Mongolia, Gansu, Xinjiang, Qinghai province in this study (Tian et al., 2011).

### 2.3 Determination of EFs

In order to ensure the accuracy of the emission inventory as much as possible, it is important to choose the appropriate EF. The EFs used in this study were mainly based on localized measurements. When selecting the EFs, we applied the following principles: first, for a certain type of biomass source or crop type, we prioritized the use of localized measured EFs from the literature. Second, for the biomass sources or crop type which lacked localized measurements, we prioritized results from developing foreign countries similar to our country above those of developed countries. Third, when localized measured data of a certain crop type were missing, the average value of the mainstream literature in the foreign country was used as an estimate. After extensive literature research on EFs, the resultant EFs for indoor and outdoor biomass burning for each chemical species and each source are summarized in Tables 5 and 6, respectively.

### 2.4 Spatial distribution

In order to obtain the detailed spatial distribution characteristics of biomass emission, and to provide grid based data for the air quality model simulation, the biomass burning inventory in this study is assigned into $1 \times 1$ km grid cells based on the source-specific surrogate. We applied GIS software as the main tool to produce the spatial distribution. In this paper, the approaches used to determine spatial distribution varied between biomass sources; thus, we selected different methods of spatial allocation according to the homologous source characteristics. The regions in which open biomass burning occurred can be located according to the MODIS fire counts data (van der Werf et al., 2006; Liu et al., 2015). Detailed description about the MODIS fire data are shown in Supplement (S4). The outdoor biomass sources (e.g. outdoor straw burning, forest fire, and





grassland fire) were treated as point sources that were located based on MODIS fire data, and land use data provided by Ran et al. (2010). Farmland fire point, forest fire point and grassland fire point are the spatial surrogates of outdoor straw burning, forest fire and grassland fire, respectively. The emissions of straw, firewood, and livestock excrement combustion were treated as area sources and the spatial surrogates used to distribute these biomass sources were population density of different land

use types (e.g. rural population density, grassland population density) (Zheng et al., 2009; Huang et al., 2012c). The population density of different land use types is according to the land use data provided by Ran et al. (2010) and population distribution data provided by Fu et al. (2014). Detailed calculation method and equation of gridded emission are presented in Supplement (S4).

## 2.5 Temporal distribution

According to the temporal resolution of MODIS fire data, the monthly/daily emission of outdoor straw burning, forest fire and grassland fire emission can be estimated based on the number of typical fire points. For indoor biomass source, the monthly uneven coefficient was mainly derived from our survey questionnaire. Details of the questionnaire survey are presented in the Supplement (S3). The daily indoor emission is equally allocated from the monthly emission.

## 2.6 Speciation of VOCs and PM$_{2.5}$

The detailed species emission of VOCs and PM$_{2.5}$ is necessary information of model simulation for different chemical mechanism selection (e.g., CB05). The speciation of VOCs and PM$_{2.5}$ is the main research object of the chemical composition of the atmospheric emission source, which has received extensive attention by domestic scholars in recent years (Song et al., 2007; Li et al., 2007c; Liu et al., 2008).

In this study, the species emission was mainly estimated based on the total emission, and VOC and PM$_{2.5}$ source profiles
(mass fraction) of biomass sources collected from literature review. In terms of the data selection, we prioritized domestic measurement with the species as much as possible. Therefore, the VOCs source profile mainly refers to data from Liu et al. (2008), including 91 species covering alkane, alkene, alkyne, benzene series compounds and so on; the PM$_{2.5}$ source profile data is cited from the work of Li et al. (2007c) and Watson et al. (2001), including 36 species, such as element, ion and so on.

## 3 Results and discussion

### 3.1 Total emissions in China

### 3.1.1 Contributions by biomass burning sources

The annual emissions of biomass burning in mainland China are presented in Table 7; The total annual emissions of SO$_2$, NO$_x$, PM$_{10}$, PM$_{2.5}$, VOC, NH$_3$, CO, EC, OC, CO$_2$, CH$_4$ and Hg for Chinese mainland in 2012 are 332.8 Gg, 972.5 Gg, 3676.0 Gg, 3479.4 Gg, 3429.6 Gg, 395.8 Gg, 33987.9 Gg, 367.1 Gg, 1151.7 Gg, 665989.0 Gg, 2076.5 Gg and 3.65 Mg, respectively. The





contribution of different sources to the total emissions to various chemical species is shown in Fig. 2. It shows that the straw indoor, outdoor burning and firewood combustion are the dominant biomass burning sources with the total contribution ranging from 95.9% to 99.1% for various species. However, the largest contributing sources to different species are not similar. Compared with other sources, straw indoor burning contributed most to $SO_2$, CO, $CH_4$ and Hg, accounting for 58.5%, 58.7%, 53.6% and 42.2% of total emissions, respectively. Straw indoor burning has a direct impact on residents and the prolonged exposure under high indoor biomass burning emission (e.g., $SO_2$, CO, $CH_4$ and Hg) can cause many adverse health effects (e.g. acute respiratory infections and chronic bronchitis) (Emily and Martin, 2008). The contribution of firewood to each species cannot be neglected, especially for EC (51.6%) and $NH_3$ (41.8%). According to the localized measurement of EF by Li et al. (2009), the average EF for firewood (1.49 g/kg) is 3.5 times of crop residue (0.43 g/kg). EF of firewood $NH_3$ is larger than the average of various straws. This results in a large contribution by firewood for these two species. The contribution of straw indoor and outdoor burning to $NO_x$, $PM_{10}$, $PM_{2.5}$, VOC, OC and $CO_2$ is nearly equal. Straw burning has an important influence on indoor air quality and the outdoor atmospheric environment.

In addition to the sources mentioned above, the contribution of livestock excrement burning, forest and grass fires is relatively small. The contribution of livestock excrement burning to $PM_{10}$, $PM_{2.5}$, $NH_3$, EC, OC, $CO_2$ and $CH_4$ is 2.55%, 2.51%, 3.49%, 1.53%, 2.03%, 1.69% and 2.12%, respectively. The contribution of forest and grass fires to biomass burning emissions to most chemical species in China is negligible (0.19–0.66%), except for the contribution of forest fire to Hg emissions (2.65%).

**3.1.2 Contributions by various crop straw**

As mentioned in Section 3.1.1, straw burning is the important biomass burning source with considerable influence on the chemical species that most strongly impact the air quality, climate change and human health. Furthermore, the major crop straw type contribution was analysed. Figure 3 shows the contributions of 12 different types of crop straw indoor and outdoor burning to various chemical species in 2012 from the perspective of the whole country. Figure 3c indicates that corn, rice and wheat straw are the major crops straw burned as fuel or as waste in China. The contribution is more than 80% to the total emissions of each chemical species. Corn, rice and wheat are the three major food crops in China with large planting area (the output of these three kinds of grain accounts for 70% of the total grain output in China, NBSC, 2013c), resulting in a large amount of straw production. Among the various crops, corn straw burning has the greatest contribution to all of the chemical species except for $CH_4$. Rice straw is major contributor to $CO_2$, VOC, $CH_4$ and $NH_3$ emissions, accounting for 32.90%, 32.43%, 31.61% and 30.12%, respectively; wheat straw has a considerable contribution to Hg, $SO_2$ and OC emissions, accounting for 29.46%, 26.47% and 25.91%, respectively. Compared with the three kinds of crop mentioned above, the total contribution of soybean, cotton, sugar cane, potato, peanut and rape to the various chemical species is relatively small, accounting for 8.1– 19.2% of the total emission; the contribution of sesame, sugar beet and hemp burning to various chemical species emission is negligible, never exceeding 0.5%. In addition, Fig. 3a and Fig. 3b indicate that for most of the chemical species, the contribution of corn straw outdoor burning is larger than that of indoor burning, except for $SO_2$, EC and $CO_2$. Contrary to that for corn straw, emissions of all chemical species (except for $SO_2$, $NO_x$ and EC) from wheat straw indoor burning is greater




than those from outdoor burning. For rice straw, the contribution of outdoor burning to NO$_x$, PM$_{10}$, PM$_{2.5}$, VOC, EC and OC emissions is larger than indoor burning.

### 3.2 Emissions from different regions

#### 3.2.1 Total emissions for different provinces

The total biomass burning emissions in the 31 provinces in 2012 are presented in Table 7. These results indicate that Heilongjiang, Shandong, Henan, Hubei, Anhui, Sichuan, Jilin, Inner Mongolia, Hunan and Jiangsu province are the major contributors, with the total contributions of various pollutant emissions between ranging from 54% to and 66% for various species. The province with most contribution to total emission of NO$_x$, PM$_{10}$, PM$_{2.5}$, VOC, NH$_3$, OC, CH$_4$ and CO$_2$ is Heilongjiang; while Shandong province has the highest emission of SO$_2$, CO, EC and Hg. It could be attributed to different

types of biomass consumption in each province due to geographical location, climate conditions and population density. Detailed discussion about the contribution by biomass source and crop straw type of different regions is shown below.

#### 3.2.2 Contributions by biomass sources of each province

The emission of detailed biomass sources of each province is presented in Fig. 4. The provinces with major contribution to total pollutants emissions for each biomass source are various. Straw burning emissions mainly distributed in Shandong, Henan,

Heilongjiang, Hebei, Anhui, Sichuan, Jilin and Hunan province. The total contribution of these provinces to various pollutants is more than 58%. It is due to the large amount of cultivated land in the north plain region as cultivated land in this region prioritizes economic crops that produce rich straw resources. Several regions in which firewood produce a large emission are Hunan, Yunnan, Hubei, Hebei, Sichuan, Guangdong, Shaanxi, Liaoning and Jiangxi province. More than 54% firewood combustion emission is contributed by these provinces. These areas are mainly distributed in the south of China, a mountainous

region in which the forest cover is higher than 30% (NBSC, 2013c). Livestock excrement combustion emissions mainly distributed in Tibet, Inner Mongolia, Gansu, Xinjiang, and Qinghai province, since only pastoral and semi−pastoral areas burn livestock manure as fuel in China. Emissions from forest and grassland fires are mainly distributed in Yunnan, Zhejiang, Inner Mongolia and Sichuan province. This is owing to the vegetation and climatic conditions in these areas.

The contribution of biomass sources to total emissions in each province is also distinct. Straw burning has a large

contribution to various pollutants emissions in Heilongjiang (79−97%), Ningxia (87−97%), Shandong (74−95%), Jilin (73−95%), Henan (60−91%), Anhui (51−91%) and Sichuan (57−89%) province. The economic income of the rural areas in these provinces is relatively low. A large number of crop residue is consumed as main non−commodity energy. In addition, firewood resource is scarce in these areas and as a result the usage of straw is very high. Figure 4 also indicates that, for most provinces (e.g. Beijing, Tianjin, and Hebei), the contribution of the indoor straw burning is greater than outdoor straw burning.

This is mainly attributable to the gradual response to the prohibition of burning straw and the introduction of straw resource utilization measures. The emission contribution of straw burned in fields is higher than that of straw domestic burning in Hebei,





Heilongjiang and Anhui province. It suggests that the prohibition of burning straw measures in these provinces still needs to be strengthened. Several regions in which firewood produce a large component of total emissions of various pollutants are Beijing (56−90%), Guangdong (31−83%), Yunan (37−81%), Fujian (29−80%), Hainan (26−64%) and Guizhou (27−74%) province. The straw amounts in the rural areas of these provinces are relative low. Firewood is the mainly non−commodity

energy used by rural people. It is worth noting that though the biomass fuel consumption in Beijing is small, the firewood is main bio−fuel due to the server restriction of straw open burning. Compared to straw burning emission contribution (10%−45%), firewood emission (55%−90%) represents a large proportion of the total biomass burning in Beijing. Firewood gradually replaces straw as the main non−commodity biomass energy source in suburban Beijing in recent years (Wang, 2010; Liu, 2012). In addition, Tibet and Inner Mongolia are the major provinces where livestock excrement produces a large

component of total pollutant emissions. Less crop straw and little firewood is used as a fuel source and thus fierce has a large contribution to total biomass emissions in these provinces. Forest and grassland fires have a small contribution to pollutant emissions in each province. The contribution of Hg emission by forest fire in Jiangsu, Zhejiang, Fujian, Hunan, Yunan, Tibet and Qinghai province is considerable (exceeding 10%), which due to the high EF of Hg for forest fire.

### 3.2.3 Contributions from different crop straws of each province

As the largest biomass source, crop straw burning represents a major contribution to the total emissions from biomass burning. The 12 different types of straw burning emission of each province are further analysed in Fig. 5. The corn straw burning emission is concentrated in Heilongjiang, Shandong, Inner Mongolia, Hebei, Henan, Shanxi and Sichuan province, with the total contribution more than 72%. Wheat crop straw emissions mainly distributed in Henan, Shandong, Anhui, Hebei, Jiangsu, Sichuan, Shaanxi, Hubei and Shanxi province. More than 89% wheat combustion emission is contributed by these provinces.

Rice crop straw combustion emissions mainly distributed in Heilongjiang, Hunan, Jiangsu, Sichuan, Anhui, Hubei, Guangxi, Guangdong and Zhejiang province, with the total contribution more than 71%. The water condition, light and heat are better for the cultivation of rice in the south. Low temperature, long sunshine duration, and the large temperature difference between day and night are suitable for wheat growing in the north. In addition, soybean, cotton, sugar cane, potato, peanut and rape straw have a small contribution to the various chemical species, and these straw are mainly distributed in Heilongjiang,

Xinjiang, Guangxi, Sichuan, Henan and Sichuan province, respectively.

### 3.2.4 Emissions intensity at county resolution

At county resolution, we found that the spatial distributions of emissions for various chemical species are similar, taking $PM_{2.5}$ as an example to analyse the emission intensity (e.g., per unit area, per capita) at county resolution. Figure 6a shows the county-level geographic distribution of $PM_{2.5}$ emissions in 2836 counties. The numbers of counties within different emission ranges were shown in Fig. 6d. The spatial diversity of various counties emission is obvious. There are 403 counties without biomass

burning, because they are mainly distributed in the urban areas of developed cities, such as the Dongcheng and Xicheng districts in Beijing, the Jing'an district in Shanghai. The emission of the 33.4% of the total districts and counties (948) were



less than 0.25 Gg. The cumulative frequency analysis result indicated that the emission in most of the counties (i.e., more than 90%) were less than 3.2 Gg, including the regions with low crop yield or scarce population. The emission of the 31.1% of the total counties (883) were more than the average emission across all counties (1.227 Gg). The two largest emission (approximately 16 Gg) appeared in Longjiang and Wuchang which are major grain-producing counties in Heilongjiang

province.

Figure 6b shows the $PM_{2.5}$ emissions intensities per unit area. The most of high values (more than 3 Mg km$^{-2}$ a$^{-1}$) mainly appeared in the north and central region of China (e.g., Hebei, Jiangsu, Shandong, Anhui, Jiangxi, Hunan), where the land is relatively flat and has predominantly agricultural function, with a substantial amount of crop straw from a relatively small area. The most counties with lower intensity concentrated in Tibet, Qinghai and Xinjiang province. In addition, it could be found

that some rural counties in Heilongjiang, Jilin, and Liaoning provinces show substantial emissions, but relative lower intensity (e.g., Nenjiang in Heilongjiang,Dunhua in Jilin, Chaoyang in Liaoning) due to the large area of these counties.

$PM_{2.5}$ emissions intensities per capita is illustrated in Fig. 6c. Because of the diversity of population density and biomass energy utilization, the emissions intensities per capita among various counties presents obvious difference. The counties with emission intensity more than 20 kg per$^{-1}$ a$^{-1}$ are mainly distributed in Heilongjiang, Jilin and Sichuan province. The high

emission intensity in northeast China are mainly attributed to the large amount of biomass burning emissions from straw and firewood burning. The high emission intensity in southwest China mainly because these regions are less economically developed and population are relatively low. The most counties with lower emissions intensities per capita concentrated in Henan, Guangdong, and Shanxi provinces.

### 3.3 Spatial distribution of biomass burning emissions

As chemical species showed a similar distribution, $PM_{2.5}$ is taken as an example to discussion the grid emission distribution. Figure 7 shows the 1 ×1 km grid distribution. It illustrates that high biomass emissions are distributed in Henan, Heilongjiang, Shandong, Anhui, Hebei and Sichuan provinces; the areas with higher emission are mainly scattered in northeast to central–south China's major agricultural region, showing a zonal distribution. The biomass burning emissions are concentrated in the regions with greater agricultural and rural activity, and lower economic income. These regions characterized by dense

population, cultivated areas and tree resources. Low emissions are mainly distributed in the part of southwest, northwest regions and downtown areas of the majority of urban areas. The scarce populations and crop yields in part of southwest and northwest areas, and lower agricultural activity in downtown areas result in lower emissions. In addition, it should be noted that grid distribution result allocated from emission at county resolution could reflect the fact that there is no grid emission within several urban regions in eastern China. The error will be brought in grid emissions if they are allocated from the emission

inventory at coarse preliminary resolution (e.g., provincial or prefectural resolution before spatial allocation) based on the gridded surrogates (e.g., rural population). Consequently, gridded emissions, which were obtained through spatial allocation from emission inventory at county resolution, could better represent the actual situation.



### 3.4 Temporal variation in biomass burning emission

Figure 8 shows the 12 species emissions in each month, indicating that there are differences in the chemical species emissions in different months. The chemical species showing large monthly variations were $NO_x$, $PM_{10}$, OC, VOC and $PM_{2.5}$. This is because the main contribution of these species emission sources is from straw outdoor burning. The outdoor burning straw
mainly occurs in the harvest season and thus shows the obvious monthly variation features. The sources of $NH_3$, CO and EC emissions are dominated by straw and firewood indoor burning and the contributions of these two kinds of source to the total emissions of these species are 74.08%, 76.8%, and 87.51%, respectively. The temporal distribution of these two sources was more uniform at the monthly scale, and thus monthly emissions of these three chemical species showed less temporal distinction. Despite the temporal variations of some pollutants at the monthly scale, the overall trends of emissions to each
species show a certain similarity: April, June and October are months with peak emissions and the contribution of these three months to the totals for various pollutant are 8.7–11.2%, 8.4–12.7% and 9.0–12.8%, respectively, due to the outdoor burning of the main crop in the harvest season.

Burning activity mainly occurs in the harvest season (crop residue burning) or crop sowing season (clearing the cultivated land and increasing the soil fertility for the next sowing). Because of the differences in climate conditions and sowing practices
in each region, monthly emission features vary regionally and to consider this, we divided China into seven areas, again taking $PM_{2.5}$ as an example to analyse the pollutant emission characteristics (Fig. 9). Regions located in south China (including Fujian, Guangdong, Hainan and Guangxi provinces) and southwest China (including Chongqing, Sichuan, Guizhou, Yunnan and Tibet provinces) have climates that are highly suited to arable agriculture because of the sufficient heat and abundant rainfall. As indicated by Fig. 9, the south regions have relatively small peak of $PM_{2.5}$ emissions in February, April and August, these
periods are consistent with local sowing and harvest times in south region. As a result of the climate discrepancies, crops in these areas are sown earlier than in northern areas. February and April are the sowing season of beans, the harvest season of the first-round and second-round crop (e.g., rice), respectively (CAAS, 1984). For the southwest region, the emission peaks are mainly distributed in February, May and August, which differ from south regions due to the inclusion of May, which is the harvest season of rapeseed.

For the central region (including Henan, Hubei and Hunan provinces), the main crops are winter wheat and summer corn and the harvest season of these two crops are the end of May and the end of September (MOA, 2000). The peak emissions in the east region (including Shanghai, Jiangsu, Zhejiang, Anhui and Jiangxi provinces) are mainly distributed from May to July, while May, June and July are the harvest seasons of rapeseed, wheat and rice in east region, respectively. The northern plains of China (including Beijing, Tianjin, Hebei, Shanxi, Inner Mongolia and Shandong provinces), include the largest agricultural
area in the country, accounting for 34% of the rural population, 27% of the farmland and 35% of the harvest crops (NBSC, 2013c). This region differs from the eastern and central parts firstly in the usage of firewood, since here firewood is also used as heating energy and therefore the consumption of firewood in winter is greater than in summer. In addition, for the burning of outdoor straw, northern winter wheat and corn are mainly harvested in June and October, respectively, and April and May



are the sowing seasons of spring rice and soybeans. Northeast region (including Liaoning, Jilin and Heilongjiang provinces) shows high value in October, April and November. The high value in April was a result of burning activity. The peak in October was mainly due to the harvesting of corn and November is the harvest season for rice. In the northwest region (including Shaanxi, Gansu, Qinghai, Ningxia and Xinjiang provinces), the peak in March-April and October is due to burning
activities for next sowing and corn harvesting, respectively.

Furthermore, the daily PM$_{2.5}$ emissions are estimated according to the monthly emissions and the biomass sources daily non-uniformity coefficient, and are shown in the Supplement (Fig. S3). It could be found that the main emission peak is in early April, early June and the whole month of December. This is due to (1) burning activities for the next sowing in the south, southwest, and northeast regions; (2) the harvest season of winter wheat in the central, east, and north regions; and (3) the
harvest season of corn in the central, northeast, northwest regions.

### 3.5 Emissions of PM$_{2.5}$ and VOC species

The total PM$_{2.5}$ emission of biomass burning emission in this study is 3479 Gg. According to our calculation based on the method described in Sec. 2.6., OC is the largest contributor of PM$_{2.5}$ accounting for 33.1% of total emission. Cl$^-$, EC, K$^+$, NH$_4^+$, K, and SO$_4^{2-}$ are also the major species of PM$_{2.5}$, and the contribution of these species is 48.29%. Besides, there are several
species have less emission (e.g. Al, Si, Mg). Detailed PM$_{2.5}$ components emissions are presented in Supplement (Fig. S4).

The total VOC emission is 3429.6 Gg in this study. The alkenes are the major contributor of biomass burning VOC emissions. The contribution of alkenes to the total VOC emission is approximately 39%, more than it of alkane (25%), benzene series compounds (18.4%), alkynes (11.1%) and halocarbons (6.5%). Among the 91 species, ethylene, acetylene, propylene and 1-butylene are the major species of alkenes and alkynes, with the total contribution accounting for 29.7%. Methyl chloride,
ethane, n-propane, isobutene, and n-butane are the main species of alkanes and halocarbons, with the total contribution accounting for 17.5%. Benzene, toluene, mp-xylene and styrene are the major species of benzene series compounds, with the total contribution of 12.1%. Several species mentioned above are key for the formation of secondary air pollution, such as ethylene, propylene, toluene, mp-xylene and halocarbons (Huang et al., 2011). It illustrates that the biomass burning emission control is very necessary for the air quality improvement. Detailed VOC species emission is shown in the Supplement (Fig.
S5).

### 3.6 Uncertainties in biomass burning emission estimates

The Monte Carlo method is used to analyse the uncertainty of this emission inventory, which was used in uncertainties estimation for many inventories studies (e.g., Streets et al., 2003; Zhao et al., 2011; Zhao et al., 2012). Activity data are assumed to be uniform distributions (Zheng et al., 2009) and EFs are assumed to be lognormal distributions (Zhao et al., 2011).
We ran 10000 Monte Carlo simulations to estimate the range of emissions with a 95% confidence interval. Uncertainty ranges of different pollutants in emission estimation are in Table 8. The uncertainty of NH$_3$, SO$_2$, EC, PM$_{10}$, and OC are large compared to other chemical species. The total uncertainty for emissions of these species are (−78%, 76%), (−76%, 75%),



(−103%, 37%), (−37%, 80%), and (−78%, 24%), respectively. $NH_3$, EC, and $SO_2$ exist the highest uncertainties in livestock excrement combustion. The parameters used in activity data estimation of livestock excrement exist large uncertainties, such as the proportion of total livestock manure directly combusted. Besides, the localized measurements of EF for livestock manure are limited, resulting in the large uncertainty of emission. In addition to livestock excrement source, straw burning is another

large uncertainty contributor to $NH_3$, EC, and $SO_2$ due to the parameter used in the estimation of straw burning amount (e.g., percentage of crop straw burned as fuel or in−field burning). The $PM_{10}$ uncertainty is attributed in straw burning with the uncertainty ranging from −46.65% to 101.94%. It is mainly due to the limited localized measurements of EF. The uncertainty of OC is mainly attributed to firewood burning (−179.82% to 60.75%). Because the firewood consumption could not be directly obtained from yearbook after 2007, the data used in this study is estimated by statistical regression. This process may bring

uncertainty to some extent. Though the uncertainty exists in this study, compared with the limited research of national and comprehensive emission with uncertainty analysis (Table 8), our emission inventory is relatively reliable due to the selection of localized and specific crop EFs.

### 3.7 Comparison with other studies

In this paper, the national biomass burning emission inventory published after 2000 has been compared with this study (Fig.

10). It could be found that the relatively high difference (range from −80% to 426% for various species) occur between our estimation and earlier studies (e.g., published paper before 2006) due to the economic development and EF localization. Compared with recent studies, the $SO_2$, $NO_x$, $PM_{2.5}$, EC, and OC emissions of our estimation are close to those derived from Lu et al. (2011), with the difference ranging from −32% to 16%. While the $PM_{10}$, VOC, $CH_4$ and $NH_3$ emission in this study is lower than Lu et al. (2011). The EFs of $PM_{10}$, VOC, $CH_4$ and $NH_3$ for various crop types used in this study is generally

lower than the EF without disgusting crop types in Lu et al. (2011). The $SO_2$, $NO_x$, $CH_4$ and $CO_2$ emissions in this study are close to those in Tian et al. (2011), with the difference ranging from −48% to 42%. The difference of CO emission is relatively high. The major emission difference of the straw indoor burning, straw outdoor burning, and firewood combustion between our paper and Tian's et al. (2011) research are −78%, −17%, and −122%. The reason is also the selection of EF. Our localized EF for crop and firewood is lower than EFs in Tian et al. (2011). In addition, for $NH_3$ emission, compared with the earlier

studies, our estimation is close to that derived from recent research (Kang et al., 2016). The difference is less than 17%. For Hg emission, our estimation is lower than Huang et al. (2012d), but is close to Chen et al. (2013). The EF of Hg is classified by stem and leaves (40 ng/g and 100 ng/g for firewood; 35 ng/g and 319 ng/g for crop residues) in Huang et al. (2012d), which is higher than the localized EF classified by specific crop (mean EF is 6.08 ng/g) and firewood (7.2 ng/g).

### 4 Conclusions

In this study, a comprehensive biomass burning emission inventory with high spatial and temporal resolution was developed for mainland China in 2012, based on the county-level activity data and updated source-specific EFs. The emission involves



crop straw domestic combustion and in field burning, firewood and livestock excrement combustion, forest and grassland fires. The total annual emissions of $SO_2$, $NO_x$, $PM_{10}$, $PM_{2.5}$, VOC, $NH_3$, CO, EC, OC, $CO_2$, $CH_4$ and Hg are 332.8 Gg, 972.5 Gg, 3676.0 Gg, 3479.4 Gg, 3429.6 Gg, 395.8 Gg, 33987.9 Gg, 367.1 Gg, 1151.7 Gg, 665989.0 Gg, 2076.5 Gg, and 3.65 Mg, respectively.

The straw indoor and outdoor burning and firewood combustion are the major biomass burning sources, while the largest contributing source to various pollutants is different. Straw indoor burning is the major source of $SO_2$, CO, $CH_4$ and Hg emission, while firewood contributes most to EC and $NH_3$ emission. Corn, rice and wheat straw are the major crop types, with the total contribution exceeding 80% for each pollutant. Corn straw burning has the greatest contribution to EC, $NO_x$ and $SO_2$ emissions; rice straw burning is dominant contributor to $CO_2$, VOC, $CH_4$ and $NH_3$ emissions. Straw burning emissions are concentrated in agricultural provinces. Firewood burning emissions are mainly distributed in southern regions of China, where the tree resource is abundant. The corn and wheat straw burning emission are mainly distributed in the northern China, while the rice straw burning emission is concentrated in the southern China. Gridded emissions result show that high emission is concentrated in northeast and central−south region of China with more agricultural and rural activity. It also illustrates that gridded emissions, which were obtained through spatial allocation from emission inventory at county resolution instead of province or prefecture resolution, could better reflect the actual situation. Monthly distributions reveal the higher emissions in April, September and October due to the burning activity before sowing and harvesting of main crops. Regional differences of temporal distribution are attributed in the diversity of main planted crop and the climate conditions in each region. OC, $Cl^-$, EC, $K^+$, $NH_4^+$, K, and $SO_4^{2-}$ are the major $PM_{2.5}$ species, with the total contribution of 80%. Several species with higher contribution to VOCs (e.g., ethylene, propylene, toluene, mp-xylene and halocarbons) are key species for the formation of secondary air pollution. The comparison with other studies presents that the emission inventory in this study is relatively reliable. The detailed emission inventory given by this paper could provide detailed information to support the further biomass burning pollution research and the development of a targeted control strategy of all regions across the Chinese mainland.

EF and speciation of chemical species are the key parameter in the emission estimation. More localized EF of different biomass fuel types indoor and outdoor burning with diverse burning conditions, $PM_{2.5}$ and VOC source profiles that contain as much components as possible still needs to expand in the future. In addition, the higher temporal resolution (e.g. hourly resolution) satellite data are necessary to provide hourly emission information for the numerical simulation of biomass burning pollution research and effective control.

*Acknowledgements.* The MODIS Thermal Anomalies/Fire products were provided by Land Process Distributed Active Archive Center (LPDAAC). The China Land Cover product was provided by National Science & Technology Infrastructure of China, National Earth System Science Data Sharing Infrastructure. The 1 km population distribution dataset was provided by National Science & Technology Infrastructure of China, National Earth System Science Data Sharing Infrastructure. This study was supported by the Natural Sciences Foundation of China (No. 51408014), the National Science and Technology Support Project of China (No. 2014BAC23B02 & 2014BAC23B04) and the Graduate Student Science and Technology Fund (ykj-2015-12315). The authors are grateful to the anonymous reviewers for their insightful comments.



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

**Table Captions List:**

**Table 1.** The classification of biomass burning emission source.

**Table 2.** Indoor and outdoor straw burning proportions of each province.

**Table 3.** Residue-to-production ratio ($N_k$), dry matter fraction ($D_k$) and combustion efficiency ($CE_k$) used in this study.

**Table 4.** Forest biomass and combustion efficiency of each province.

**Table 5.** Emission factors used in the estimation of indoor biomass burning emissions.

**Table 6.** Emission factors used in the estimation of outdoor biomass burning emissions.

**Table 7.** Biomass burning emission inventory in the 31 provinces or municipalities of China in 2012.

**Table 8.** Uncertainty ranges of different pollutants in emission estimates (min, max). (Unit for emission estimate: Gg)

**Figure Captions List:**

**Figure1.** Regression analysis between firewood consumption and (1) rural population, (2) gross agricultural output, and (3) timber yield, respectively.

**Figure 2.** Contribution of different source to the total biomass burning emissions in China, 2012.

**Figure 3.** Contributions of 12 crop straws burning to total pollutant emissions for various species.

**Figure 4.** Contributions of different biomass sources to the emission in each province (Gg).

**Figure 5.** Contributions of different crop straw types to the emission in each province (Gg).

**Figure 6.** Biomass emission inventory at county resolution and intensity (PM$_{2.5}$).

**Figure 7.** Gridded distribution of PM$_{2.5}$ annual emissions.

**Figure 8.** Monthly variation of different biomass sources emission for each chemical species.

**Figure 9.** Monthly variation of different biomass sources emission for PM$_{2.5}$ emissions in different regions.

**Figure 10.** Comparison of the emissions inventory derived by this study with the emissions estimated by previous research.



**Table 1. The classification of biomass burning emission source.**

| I | II | III |
|---|---|---|
| | Firewood | Firewood |
| | | Cattles |
| | | Horses |
| | Livestock excrement | Donkeys |
| | | Mules |
| Domestic combustion | | Camels |
| | Crop straw | Corn, wheat, cotton, sugar cane, potato, peanut, rapeseed, sesame, sugar beet, hemp, rice, soybean |
| | Crop straw | Corn, wheat, cotton, sugar cane, potato, peanut, rapeseed, sesame, sugar beet, hemp, rice, soybean |
| | Grassland fire | Grassland fire |
| In field burning | | Tropics |
| | | South subtropical zone |
| | | Central subtropical zone |
| | | North subtropical zone |
| | Forest fire | Warm temperate zone |
| | | Temperate zone |
| | | Cold temperate zone |
| | | Tibetan region |



**Table 2. Indoor and outdoor straw burning proportions of each province.**

| Province | Straw indoor burning ratio | Straw outdoor burning ratio | Province | Straw indoor burning ratio | Straw outdoor burning ratio |
|---|---|---|---|---|---|
| Beijing | 0.0923[a] | 0.096[b] | Hubei | 0.283[j] | 0.197[o] |
| Tianjin | 0.42[a] | 0.165* | Hunan | 0.4[c] | 0.2[c] |
| Hebei | 0.35* | 0.165* | Guangdong | 0.17* | 0.197* |
| Shanxi | 0.45[c] | 0.2[c] | Guangxi | 0.2226[k] | 0.2273[k] |
| Inner Mongolia | 0.338* | 0.246* | Hainan | 0.45[c] | 0.2[c] |
| Liaoning | 0.396[e] | 0.2[c] | Chongqing | 0.4922[l] | 0.1211[l] |
| Jilin | 0.3[c] | 0.259[f] | Sichuan | 0.45[c] | 0.2[c] |
| Heilongjiang | 0.26* | 0.5* | Guizhou | 0.35[m] | 0.2[c] |
| Shanghai | 0.2[c] | 0.148* | Yunnan | 0.2[c] | 0.1* |
| Jiangsu | 0.3[g] | 0.225[g] | Xizang | 0.338[d] | 0.148[d] |
| Zhejiang | 0.3* | 0.3* | Shaanxi | 0.338[d] | 0.159[o] |
| Anhui | 0.29[h] | 0.319* | Gansu | 0.338[d] | 0.159[o] |
| Fujian | 0.3[c] | 0.188[i] | Qinghai | 0.338[d] | 0.159[o] |
| Jiangxi | 0.23* | 0.2[c] | Ningxia | 0.338[d] | 0.159[o] |
| Shandong | 0.45[c] | 0.2[c] | Xinjiang | 0.143[n] | 0.137[n] |
| Henan | 0.3[c] | 0.2[c] | | | |

[a] Fang et al. (2015). [b] Zhao et al. (2015). [c] Tian et al. (2011). [d] Bao et al. (2014). [e] Chang et al. (2012). [f] Liu et al. (2010). [g] Wang and Zhao
10 (2011). [h] Qin and Ge (2012). [i] Huang (2012a). [j] Liu et al. (2014). [k] Li et al. (2013a). [l] Li et al. (2013b). [m] Zhang et al. (2015). [n] Hou et al.
(2013). [o] EPD (2014).

* The result from our questionnaire.





**Table 3. Residue-to-production ratio ($N_k$), dry matter fraction ($D_k$) and combustion efficiency ($CE_k$) used in this study.**

| Crops | $N_k$ | $D_k$ | $CE_k$ |
|---|---|---|---|
| Corn | 1.269[a] | 0.87 | 0.92 |
| Wheat | 1.3[b] | 0.89 | 0.92 |
| Cotton | 3[b] | 0.83 | 0.9 |
| Sugar cane | 0.3[c] | 0.45 | 0.68 |
| Potato | 0.5[d] | 0.45 | 0.68 |
| Peanut | 1.5[b] | 0.94 | 0.82 |
| Rapeseed | 1.5[d] | 0.83 | 0.9 |
| Sesame | 2.2[d] | 0.83 | 0.9 |
| Sugar beet | 0.1[b] | 0.45 | 0.9 |
| Hemp | 1.7[e] | 0.83 | 0.9 |
| Rice | 1.323[a] | 0.89 | 0.93 |
| Soybean | 1.6[d] | 0.91 | 0.68 |

[a] Zhang et al. (1990). [b] Bi et al. (2010). [c] Han et al. (2002). [d] NATESC (1999). [e] Gao et al. (2009). [f] He et al. (2015).




**Table 4. Forest biomass and combustion efficiency of each province.**

| Province | Climatic zone | Forest biomass* | Combustion efficiency* |
|---|---|---|---|
| Hainan | Tropics | 348 | 0.2 |
| Guangdong | South subtropical zone | 178 | 0.2 |
| Guangxi | | 178 | 0.2 |
| Yunnan | South subtropical zone, Central subtropical zone | 138 | 0.2 |
| Fujian | Central subtropical zone | 143 | 0.2 |
| Jiangxi | | 143 | 0.2 |
| Chongqing | | 143 | 0.2 |
| Zhejiang | Central subtropical zone, North subtropical zone | 121 | 0.2 |
| Guizhou | | 121 | 0.2 |
| Sichuan | Central subtropical zone, Tibetan region | 118 | 0.2 |
| Shanghai | North subtropical zone | 98 | 0.2 |
| Anhui | | 98 | 0.2 |
| Hubei | | 98 | 0.2 |
| Hunan | | 98 | 0.2 |
| Jiangsu | North subtropical zone, Warm temperate zone | 77 | 0.2 |
| Beijing | Warm temperate zone | 55 | 0.1 |
| Tianjin | | 55 | 0.1 |
| Hebei | | 55 | 0.1 |
| Shanxi | | 55 | 0.1 |
| Shandong | | 55 | 0.1 |
| Henan | | 55 | 0.1 |
| Shaanxi | | 55 | 0.1 |
| Xinjiang | Warm temperate zone, Temperate zone | 106 | 0.1 |
| Inner Mongolia | Temperate zone | 157 | 0.1 |
| Liaoning | | 157 | 0.1 |
| Jilin | | 157 | 0.1 |
| Heilongjiang | | 157 | 0.1 |
| Gansu | | 157 | 0.1 |
| Ningxia | | 157 | 0.1 |
| Tibet | Tibetan region | 121 | 0.2 |
| Qinghai | | 121 | 0.2 |

* Tian et al. (2003).



**Table 5. Emission factors used in the estimation of indoor biomass burning emissions.**

| Material | | $SO_2$ | $NO_x$ | $PM_{10}$ | $PM_{2.5}$ | VOC | $NH_3$ | CO | EC | OC | $CO_2$ | $CH_4$ | Hg |
|---|---|---|---|---|---|---|---|---|---|---|---|---|---|
| | | g/kg* | | | | | | | | | | | ng/g* |
| Indoor burning | Corn | 1.33[h] | 1.86[a,b,h] | 7.39[h] | 6.87[h] | 7.34[h] | 0.68[h] | 82.37[a,b,h,l,m] | 0.95[a] | 2.25[a] | 1491[b,l] | 3.91[b,m] | 7.94[n,o] |
| | Wheat | 1.2[a,h] | 1.19[a,b,h,l] | 8.86[h] | 8.24[h] | 9.37[h] | 0.37[h] | 136.46[a,b,h,l,m] | 0.42[a] | 3.46[f] | 1246.7[b,l] | 8.3[b] | 11.09[n,o] |
| | Cotton | 0.53[d,k,e,g,h] | 2.49[a] | 7.69[h] | 7.15[h] | 8.82[b,j] | 1.3[b,e] | 121.7[b,h] | 0.82[a] | 1.83[a] | 963.42[b] | 6.08[b] | 3.12[n,o] |
| | Sugar cane | 0.53[d,k,e,g,h] | 1.12[d,e,f,g] | 7.69[h] | 7.15[h] | 8.82[b,j] | 1.3[b,e] | 121.7[b,h] | 0.51[d,e,g,c] | 2.21[d,e,c] | 963.42[b] | 6.08[b] | 6.5[n,o] |
| | Potato | 0.53[d,k,e,g,h] | 1.12[d,e,f,g] | 7.69[h] | 7.15[h] | 8.82[b,j] | 1.3[b,e] | 121.7[b,h] | 0.51[d,e,g,c] | 2.21[d,e,c] | 963.42[b] | 6.08[b] | 6.5[n,o] |
| | Peanut | 0.53[d,k,e,g,h] | 1.12[d,e,f,g] | 7.69[h] | 7.15[h] | 8.82[b,j] | 1.3[b,e] | 121.7[b,h] | 0.51[d,e,g,c] | 2.21[d,e,c] | 963.42[b] | 6.08[b] | 4.82[n,o] |
| | Rape | 1.36[h] | 1.65[f] | 13.73[h] | 12.77[h] | 7.97[h] | 0.52[h] | 133.5[f] | 0.51[d,e,g,c] | 2.21[d,e,c] | 963.42[b] | 6.08[b] | 6.5[n,o] |
| | Sesame | 0.53[d,k,e,g,h] | 1.12[d,e,f,g] | 7.69[h] | 7.15[h] | 8.82[b,j] | 1.3[b,e] | 121.7[b,h] | 0.51[d,e,g,c] | 2.21[d,e,c] | 963.42[b] | 6.08[b] | 6.5[n,o] |
| | Sugar beet | 0.53[d,k,e,g,h] | 1.12[d,e,f,g] | 7.69[h] | 7.15[h] | 8.82[b,j] | 1.3[b,e] | 121.7[b,h] | 0.51[d,e,g,c] | 2.21[d,e,c] | 963.42[b] | 6.08[b] | 6.5[n,o] |
| | Hemp | 0.53[d,k,e,g,h] | 1.12[d,e,f,g] | 7.69[h] | 7.15[h] | 8.82[b,j] | 1.3[b,e] | 121.7[b,h] | 0.51[d,e,g,c] | 2.21[d,e,c] | 963.42[b] | 6.08[b] | 6.5[n,o] |
| | Rice | 0.48[h] | 1.92[a,b,f,l] | 6.88[h] | 6.4[h] | 8.4[h] | 0.52[h] | 79.7[a,b,f,h] | 0.49[a] | 2.01[a] | 1147.4[a,b,l] | 4.8[b] | 5.56[n,o] |
| | Soybean | 0.53[d,k,e,g,h] | 1.12[d,e,f,g] | 7.69[h] | 7.15[h] | 8.82[b,j] | 1.3[b,e] | 80.7[f] | 0.51[d,e,g,c] | 2.21[d,e,c] | 963.42[b] | 6.08[b] | 4.48[n,o] |
| | Feces | 0.28[h] | 0.58[h] | 8.84[h] | 7.15[h] | 3.13[h] | 1.3[h] | 19.8[h] | 0.53[g] | 2.2* | 1060[g] | 4.14[g] | - |
| | Firewood | 0.4[e,g] | 1.49[b,f,h] | 5.66[h,i] | 5.22[h,d] | 3.13[j] | 1.3[e] | 48.25[b,f,h] | 1.49[c] | 1.14[c] | 1445.2[b,m] | 2.48[b,m] | 7.2[n,o] |

10  Note: Lowercase letters indicate the data source.

Sources are from the following: [a] Cao et al. (2008). [b] Wang et al. (2009). [c] Li et al. (2009). [d] Reddy and Venkataraman (2002). [e] Andreae and Merlet (2001). [f] Tang et al. (2014). [g] Tian et al. (2011). [h] EPD (2014). [i] Cao et al. (2004). [j] Wei et al. (2008). [k] Turn et al. (1997). [l] Zhang et al. (2008). [m] Zhang et al. (2000). [n] Chen et al. (2013). [o] Zhang et al. (2013a).

* The unit of emission factor.





**Table 6. Emission factors used in the estimation of outdoor biomass burning emissions.**

| Material | | $SO_2$ | $NO_x$ | $PM_{10}$ | $PM_{2.5}$ | VOC | $NH_3$ | CO | EC | OC | $CO_2$ | $CH_4$ | Hg |
|---|---|---|---|---|---|---|---|---|---|---|---|---|---|
| | | | | | | g/kg* | | | | | | | ng/g* |
| | Corn | 0.44[a,b,c] | 4.3[a,b,c] | 11.95[c] | 11.7[b,c] | 10[b] | 0.68[b,c] | 53[a,c,h] | 0.3[b,h] | 4.35[b,h] | 1350[b,h] | 4.4[b] | 7.94[n,o] |
| | Wheat | 0.85[a,b,c] | 3.3[a,b,c] | 7.73[c] | 7.58[b,c] | 7.5[b,c] | 0.37[b] | 55.8[a,b,c,d] | 0.37[b,h] | 3.9[b,h] | 1390[b,h] | 3.4[b] | 11.09[n,o] |
| | Cotton | 0.53[e,f,b,g] | 3.16[e,f,b,g] | 6.93[c] | 6.79[c] | 9.5[c,f,i] | 1.3[m] | 66.1[b,c,i] | 0.42[b] | 3.3[b] | 1410[b] | 3.9[b] | 3.12[n,o] |
| | Sugar cane | 0.53[e,f,b,g] | 3.16[e,f,b,g] | 6.93[c] | 6.79[c] | 11.02[d] | 1[m] | 40.08[d] | 0.42[b] | 3.3[b] | 1410[b] | 3.9[b] | 6.5[n,o] |
| | Potato | 0.53[e,f,b,g] | 3.16[e,f,b,g] | 6.93[c] | 6.79[c] | 9.5[c,f,i] | 0.53[b,c] | 66.1[b,c,i] | 0.42[b] | 3.3[b] | 1410[b] | 3.9[b] | 6.5[n,o] |
| Outdoor burning | Peanut | 0.53[e,f,b,g] | 3.16[e,f,b,g] | 6.93[c] | 6.79[c] | 9.5[c,f,i] | 0.53[b,c] | 66.1[b,c,i] | 0.42[b] | 3.3[b] | 1410[b] | 3.9[b] | 4.82[n,o] |
| | Rape | 0.53[e,f,b,g] | 1.12[g] | 6.93[c] | 6.79[c] | 9.5[c,f,i] | 0.53[b,c] | 34.3[g] | 0.23[g] | 1.08[g] | 1410[b] | 3.9[b] | 6.5[n,o] |
| | Sesame | 0.53[e,f,b,g] | 3.16[e,f,b,g] | 6.93[c] | 6.79[c] | 9.5[c,f,i] | 0.53[b,c] | 66.1[b,c,i] | 0.42[b] | 3.3[b] | 1410[b] | 3.9[b] | 6.5[n,o] |
| | Sugar beet | 0.53[e,f,b,g] | 3.16[e,f,b,g] | 6.93[c] | 6.79[c] | 9.5[c,f,i] | 0.53[b,c] | 66.1[b,c,i] | 0.42[b] | 3.3[b] | 1410[b] | 3.9[b] | 6.5[n,o] |
| | Hemp | 0.53[e,f,b,g] | 3.16[e,f,b,g] | 6.93[c] | 6.79[c] | 9.5[c,f,i] | 1.3[m] | 66.1[b,c,i] | 0.42[b] | 3.3[b] | 1410[b] | 3.9[b] | 6.5[n,o] |
| | Rice | 0.53[c] | 1.42[c,g] | 5.78[c] | 5.73[c,g,h] | 7.25[c,d] | 0.53[b,c] | 46.03[d,g,h] | 0.16[g,h] | 2.03[g,h] | 1393[h] | 3.9[b] | 5.56[n,o] |
| | Soybean | 0.53[e,f,b,g] | 1.08[g] | 6.93[c] | 6.79[c] | 9.5[c,f,i] | 0.53[b,c] | 32.3[g] | 0.13[g] | 1.05[g] | 1410[b] | 3.9[b] | 4.48[n,o] |
| | Forestfire | 0.79[c,e,k] | 2.3[c] | 11.28[c] | 11.05[c] | 6.9[c] | 2.9[c,l] | 105.5[c] | 0.66[k] | 6.8[k] | 1618[k] | 5.13[e] | 113[n] |
| | Grassfire | 0.35[c,e,j] | 3.9[c,i,j] | 5.51[c] | 5.4[c] | 3.4[c] | 0.7[c,l] | 65[c] | 0.65[k] | 5.1[k] | 1621[k] | 2.3[i,j] | 80[n] |

10   Note: Lowercase letters indicate the data source.

Sources are from the following: [a] Li et al. (2015). [b] Li et al. (2007). [c] EPD (2014). [d] Zhang et al. (2013b). [e] Tian et al. (2011). [f] Wang and Zhang (2008). [g] Tang et al. (2014). [h] Ni et al. (2015). [i] Streets et al. (2003). [j] Andreae and Merlet (2001). [k] Chang et al. (2010). [l] Christian et al. (2003). [m] Kanabkaew and Nguyen (2011). [n] Chen et al. (2013). [o] Zhang et al. (2013a).

*The unit of emission factor.





**Table 7. Biomass burning emission inventory in the 31 provinces or municipalities of China in 2012.**

| Province | SO$_2$ | NO$_x$ | PM$_{10}$ | PM$_{2.5}$ | VOC | NH$_3$ | CO | EC | OC | CO$_2$ | CH$_4$ | Hg |
|---|---|---|---|---|---|---|---|---|---|---|---|---|
| | unit:Gg | | | | | | | | | | | unti:Mg |
| Beijing | 0.5 | 1.8 | 6.6 | 6.1 | 4.4 | 1.2 | 55 | 1.3 | 1.7 | 1457 | 3.0 | 0.01 |
| Tianjin | 1.2 | 3.1 | 11.6 | 10.9 | 10.4 | 1.3 | 116 | 1.4 | 3.7 | 2136 | 6.6 | 0.01 |
| Hebei | 21.3 | 52.8 | 199.9 | 188.3 | 178.4 | 21.4 | 2021 | 22.7 | 65.7 | 36263 | 115.3 | 0.22 |
| Shanxi | 9.3 | 22.7 | 82.8 | 78.2 | 74.3 | 8.0 | 771 | 8.7 | 27.0 | 14545 | 43.7 | 0.08 |
| Inner-Mongolia | 15.9 | 45.1 | 215.3 | 203.3 | 151.9 | 24.0 | 1299 | 16.8 | 64.4 | 32079 | 103.2 | 0.14 |
| Liaoning | 13.8 | 40.6 | 144.3 | 136.1 | 128.1 | 17.4 | 1277 | 18.0 | 42.4 | 27366 | 72.4 | 0.14 |
| Jilin | 16.5 | 54.3 | 179.5 | 171.4 | 165.4 | 15.7 | 1394 | 14.7 | 58.3 | 29507 | 84.7 | 0.16 |
| Heilongjiang | 29.8 | 116.9 | 393.7 | 379.8 | 391.7 | 32.1 | 2849 | 22.6 | 129.5 | 65092 | 199.4 | 0.32 |
| Shanghai | 0.3 | 0.8 | 3.1 | 2.9 | 3.6 | 0.2 | 33 | 0.2 | 1.0 | 566 | 2.1 | 0.00 |
| Jiangsu | 14.6 | 39.7 | 154.2 | 146.6 | 166.6 | 12.3 | 1612 | 10.2 | 52.5 | 27488 | 102.1 | 0.16 |
| Zhejiang | 3.8 | 12.4 | 48.5 | 45.9 | 48.1 | 6.2 | 454 | 5.5 | 13.7 | 10029 | 28.7 | 0.05 |
| Anhui | 19.7 | 56.4 | 209.6 | 199.4 | 209.4 | 19.6 | 2042 | 17.6 | 71.0 | 38462 | 127.5 | 0.22 |
| Fujian | 3.1 | 10.8 | 41.2 | 38.6 | 36.8 | 6.5 | 392 | 6.4 | 10.9 | 8985 | 23.2 | 0.05 |
| Jiangxi | 8.0 | 27.4 | 105.2 | 99.3 | 102.8 | 14.3 | 1000 | 13.4 | 28.7 | 22478 | 62.2 | 0.11 |
| Shandong | 34.7 | 77.8 | 304.1 | 287.3 | 296.5 | 25.1 | 3317 | 24.9 | 108.8 | 50466 | 192.1 | 0.33 |
| Henan | 33.1 | 82.1 | 312.6 | 296.1 | 300.9 | 26.6 | 3290 | 26.0 | 112.3 | 52839 | 194.3 | 0.34 |
| Hubei | 13.2 | 40.7 | 158.1 | 148.9 | 147.4 | 19.9 | 1529 | 19.1 | 44.3 | 31146 | 91.4 | 0.16 |
| Hunan | 15.7 | 52.2 | 201.8 | 189.9 | 199.6 | 24.8 | 1973 | 23.0 | 55.6 | 39843 | 119.7 | 0.21 |
| Guangdong | 5.9 | 21.2 | 79.2 | 74.5 | 70.2 | 12.8 | 729 | 12.5 | 21.3 | 17668 | 43.6 | 0.09 |
| Guangxi | 8.1 | 29.2 | 106.1 | 100.4 | 108.1 | 15.3 | 1005 | 12.6 | 32.0 | 21369 | 61.8 | 0.11 |
| Hainan | 1.4 | 5.0 | 19.2 | 18.0 | 17.7 | 3.0 | 191 | 2.9 | 5.1 | 4040 | 11.1 | 0.02 |
| Chongqing | 5.5 | 15.6 | 61.3 | 57.4 | 58.7 | 7.4 | 620 | 7.3 | 17.0 | 11573 | 35.6 | 0.06 |
| Sichuan | 19.1 | 52.5 | 209.5 | 197.3 | 203.4 | 21.9 | 2096 | 20.9 | 61.5 | 37819 | 124.2 | 0.21 |
| Guizhou | 6.4 | 19.6 | 75.0 | 70.4 | 63.1 | 10.5 | 682 | 10.8 | 20.0 | 14985 | 39.0 | 0.08 |
| Yunnan | 8.2 | 26.1 | 99.8 | 93.4 | 78.7 | 16.6 | 906 | 17.0 | 26.1 | 21173 | 50.7 | 0.13 |
| Tibet | 0.5 | 1.3 | 9.4 | 8.8 | 6.3 | 1.1 | 56 | 0.5 | 2.7 | 1306 | 5.3 | 0.00 |
| Shaanxi | 8.4 | 22.1 | 85.6 | 80.3 | 70.7 | 11.0 | 825 | 12.0 | 25.3 | 16585 | 46.7 | 0.09 |
| Gansu | 6.2 | 15.6 | 65.7 | 61.7 | 51.8 | 8.0 | 572 | 7.8 | 19.5 | 11657 | 35.0 | 0.06 |
| Qinghai | 0.7 | 1.5 | 8.7 | 8.2 | 6.1 | 1.0 | 70 | 0.8 | 2.3 | 1322 | 4.7 | 0.01 |
| Ningxia | 1.6 | 4.1 | 14.9 | 14.1 | 14.7 | 1.2 | 143 | 1.2 | 5.0 | 2502 | 8.4 | 0.01 |
| Xinjiang | 6.1 | 21.0 | 69.7 | 65.9 | 63.7 | 9.5 | 668 | 8.4 | 22.4 | 13239 | 38.9 | 0.07 |
| Total | 332.8 | 972.5 | 3676.0 | 3479.4 | 3429.6 | 395.8 | 33988 | 367.1 | 1151.7 | 665989 | 2076.5 | 3.65 |



**Table 8. Uncertainty ranges of different pollutants in emission estimates (min, max). (Unit for emission estimate: Gg)**

| Species | Emission estimate | Uncertainty ranges * | Previous study<br>Streets et al.(2003) |
|---|---|---|---|
| $SO_2$ | 333 | (−76%, 75%) | (−245%, 245%) |
| $NO_X$ | 972 | (−62%, 24%) | (−220%, 220%) |
| $PM_{10}$ | 3676 | (−37%, 80%) | |
| $PM_{2.5}$ | 3479 | (−54%, 34%) | |
| VOC | 3430 | (−55%, 21%) | (−210%, 210%) |
| $NH_3$ | 396 | (−78%, 76%) | (−240%, 240%) |
| CO | 33988 | (−41%, 47%) | (−250%, 250%) |
| EC | 367 | (−103%, 37%) | (−430%, 430%) |
| OC | 1152 | (−78%, 24%) | (−420%, 420%) |
| $CO_2$ | 665989 | (−38%, 43%) | |
| $CH_4$ | 2076 | (−55%, 12%) | (−195%, 195%) |
| Hg | 0.00365 | (−26%, 51%) | |

* 95% confidence interval.





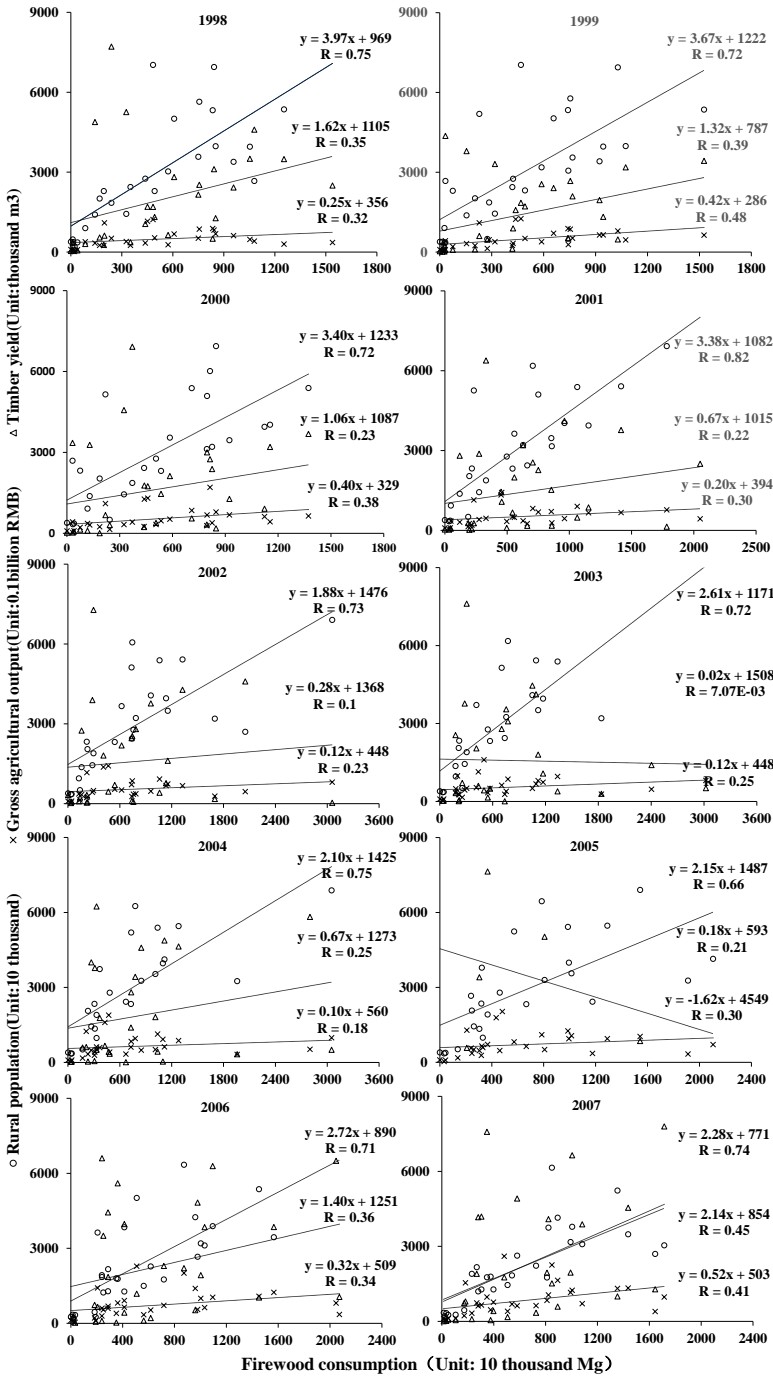

**Figure 1. Regression analysis between firewood consumption and (1) rural population, (2) gross agricultural output, and (3) timber yield, respectively.**

Note: It is referred by circles, crosses and triangles, respectively. The regression equation of each figure is provided in the top, middle and bottom, respectively.



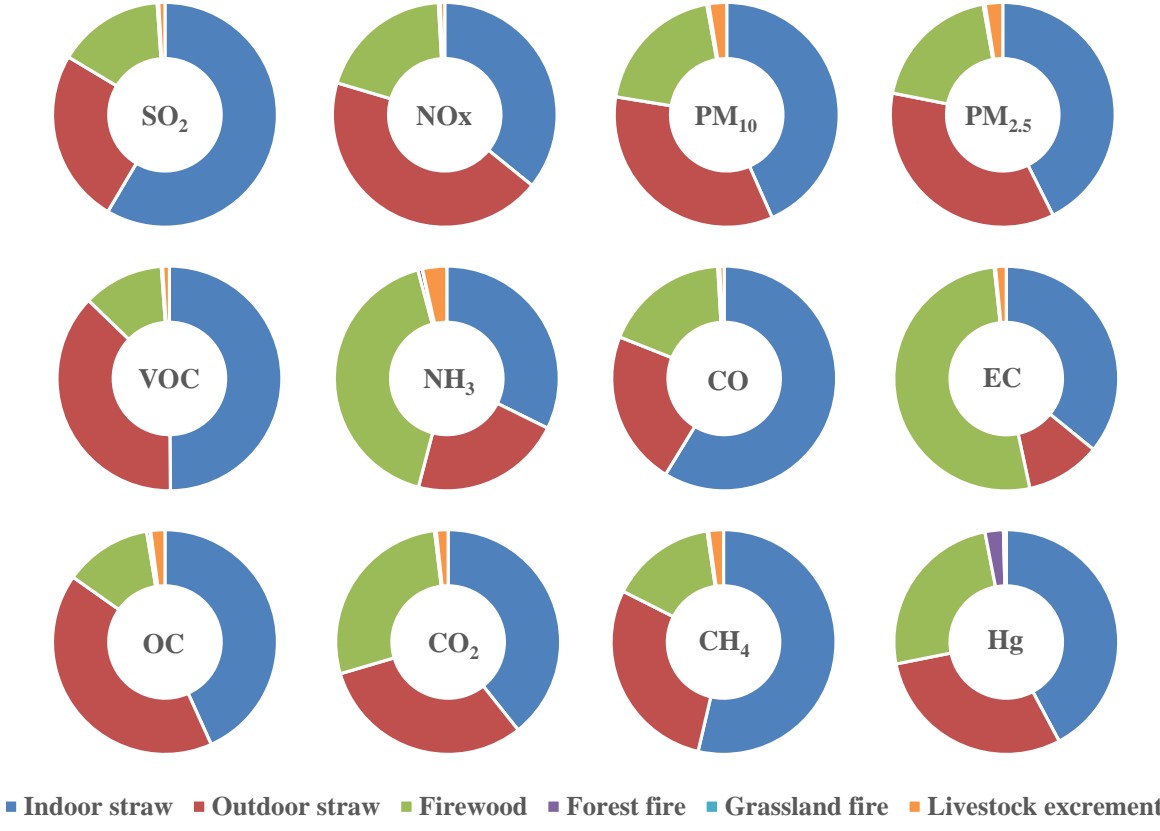

**Figure 2. Contribution of different source to the total biomass burning emissions in China, 2012.**





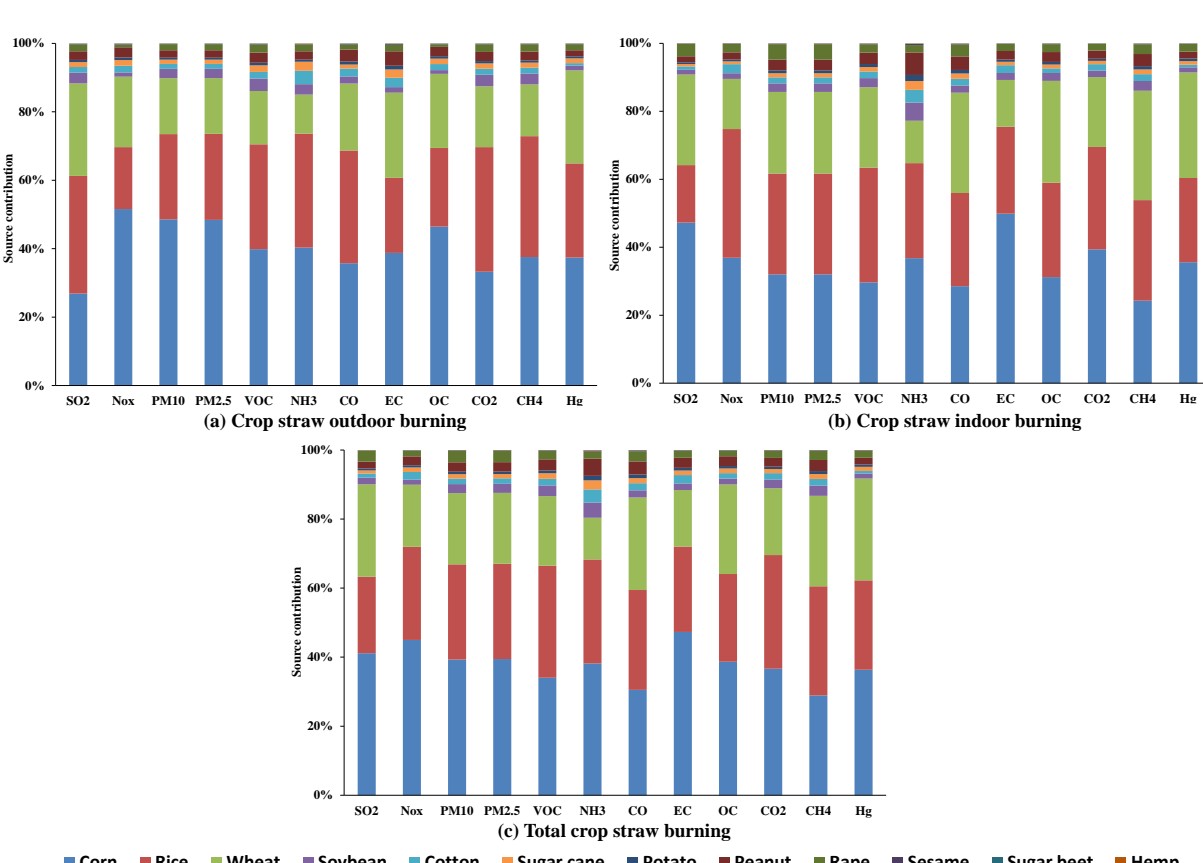

**Figure 3. Contributions of 12 crop straws burning to total pollutant emissions for various species.**





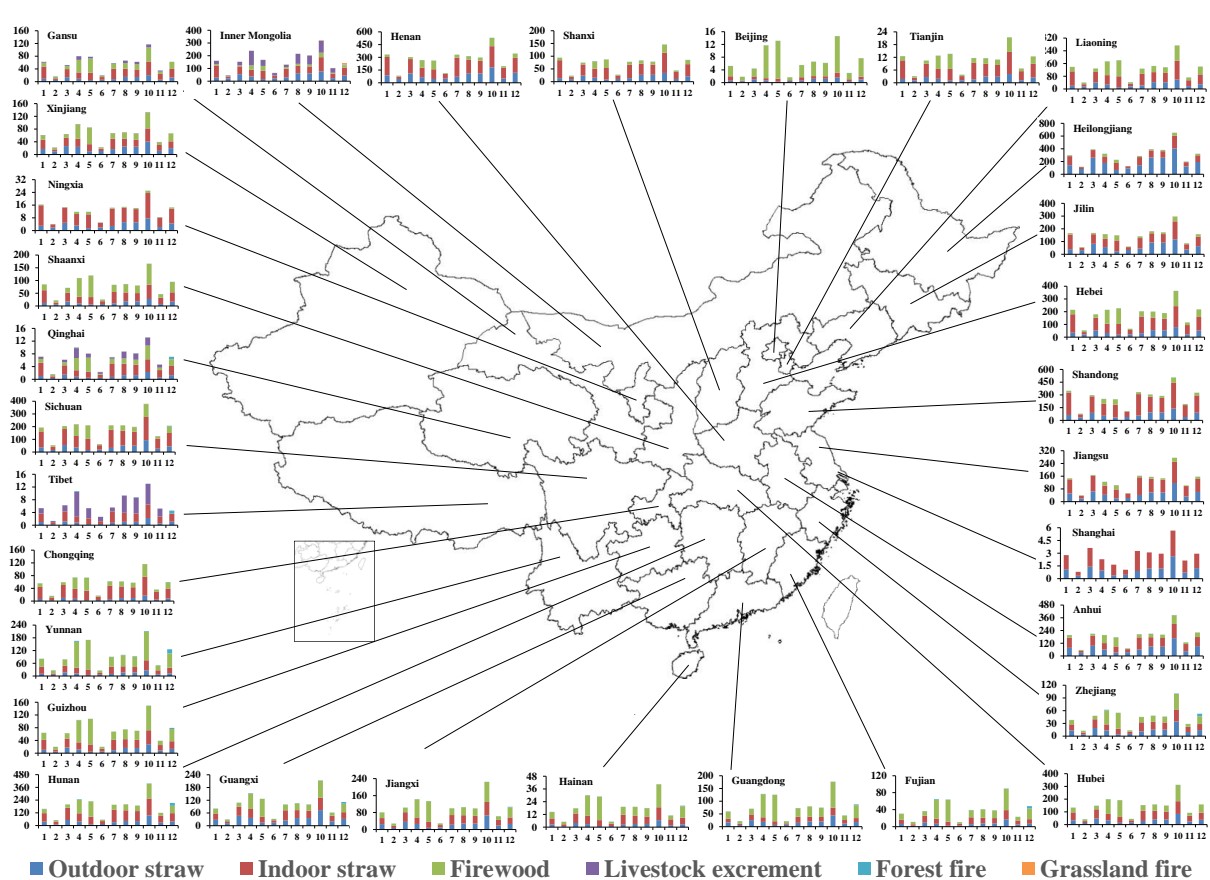

**Figure 4. Contributions of different biomass sources to the emission in each province (Gg).**

Note: The numbers 1−12 represent the species $SO_2 \times 10$, $NO_x$, VOC, $NH_3 \times 10$, $EC \times 10$, OC, CO/10, $PM_{10}$, $PM_{2.5}$, $CO_2/100$, $CH_4$ and $Hg \times 1000000$, respectively.



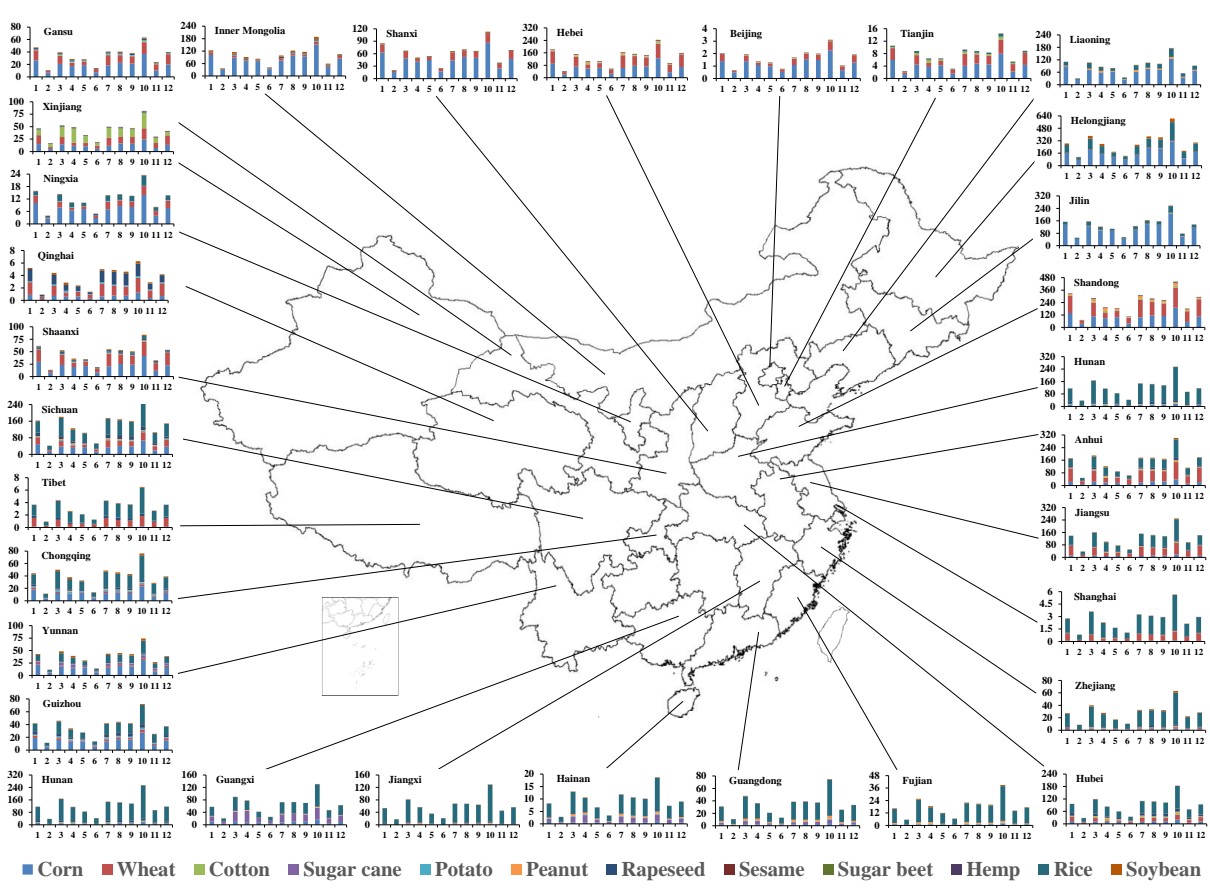

**Figure 5. Contributions of different crop straw types to the emission in each province (Gg).**

5    Note: The numbers 1−12 represent the species $SO_2 \times 10$, $NO_x$, VOC, $NH_3 \times 10$, $EC \times 10$, OC, CO/10, $PM_{10}$, $PM_{2.5}$, $CO_2$/100, $CH_4$ and $Hg \times 1000000$, respectively.



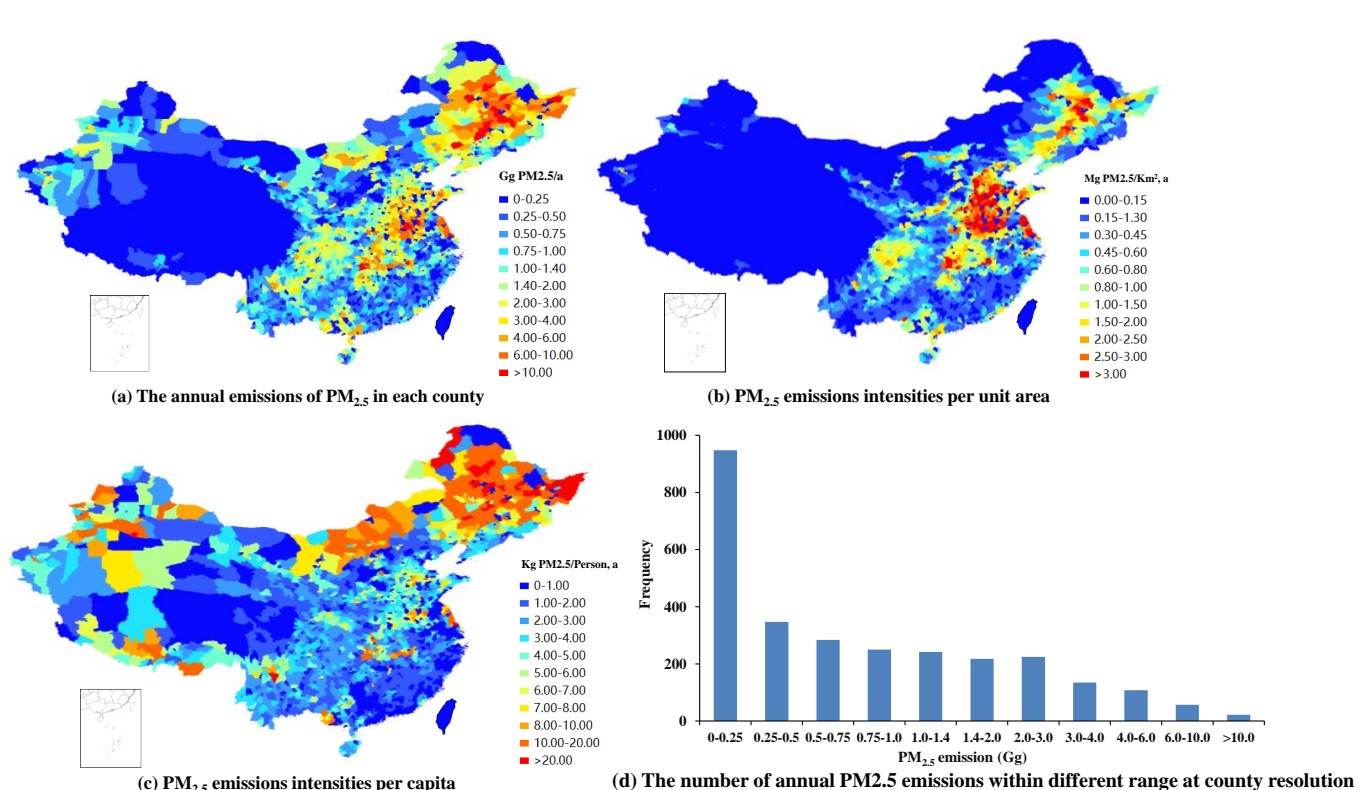

(a) The annual emissions of PM$_{2.5}$ in each county

(b) PM$_{2.5}$ emissions intensities per unit area

(c) PM$_{2.5}$ emissions intensities per capita

(d) The number of annual PM2.5 emissions within different range at county resolution

**Figure 6.** Biomass emission inventory at county resolution and intensity (PM2.5).

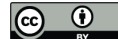





5    **Figure 7. Gridded distribution of PM$_{2.5}$ annual emissions.**





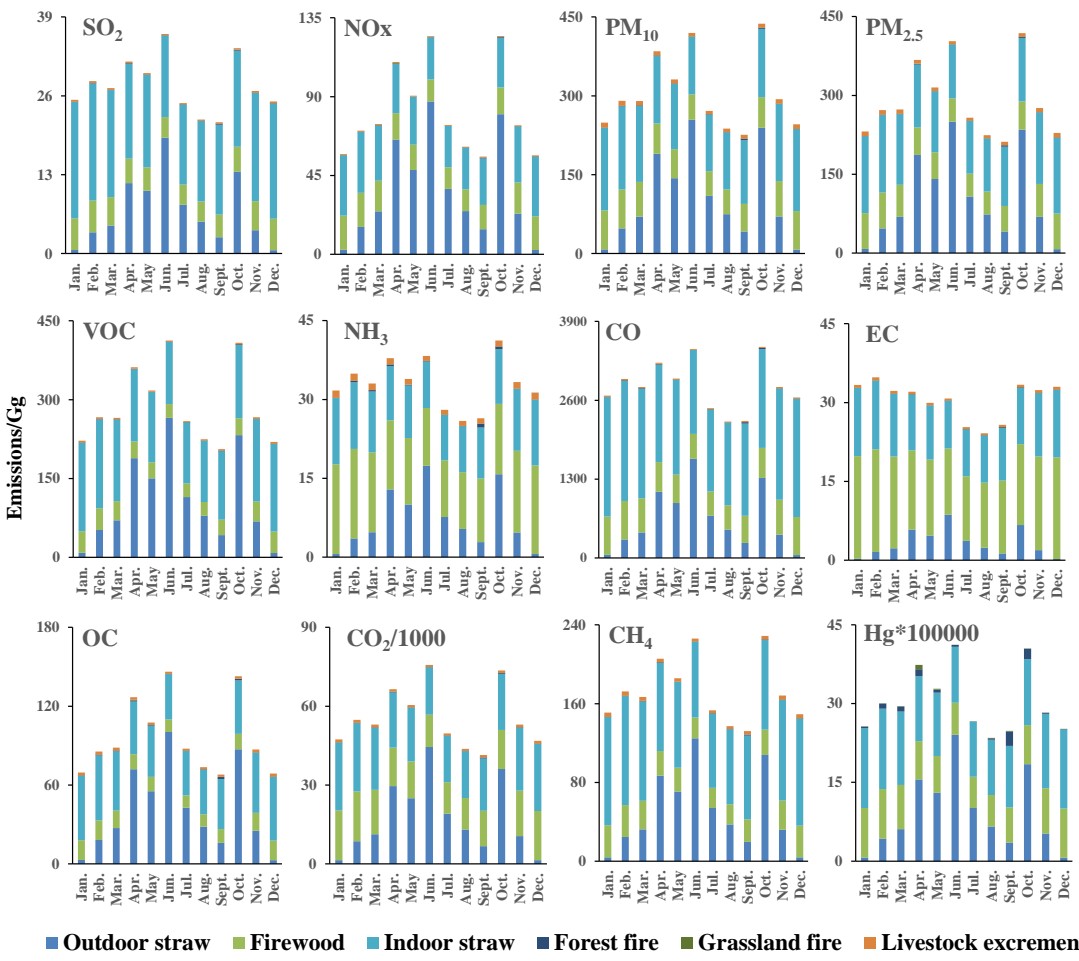

**Figure 8. Monthly variation of different biomass sources emission for each chemical species.**





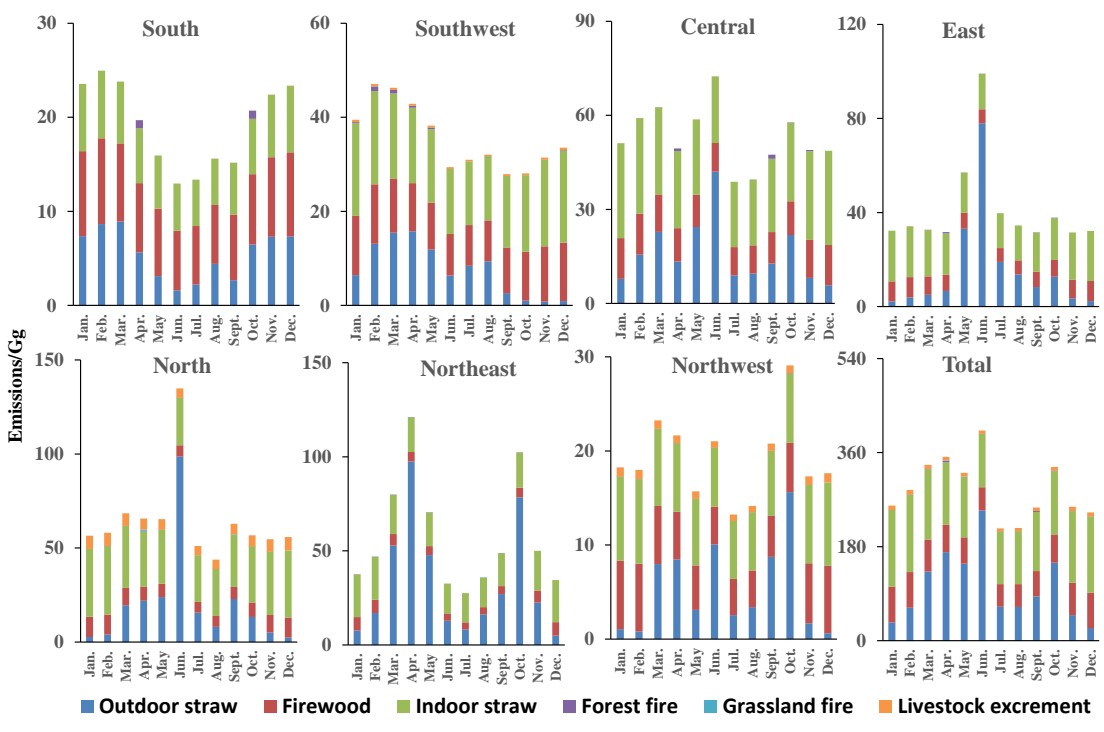

10  **Figure 9. Monthly variation of different biomass sources emission for PM$_{2.5}$ emissions in different regions.**





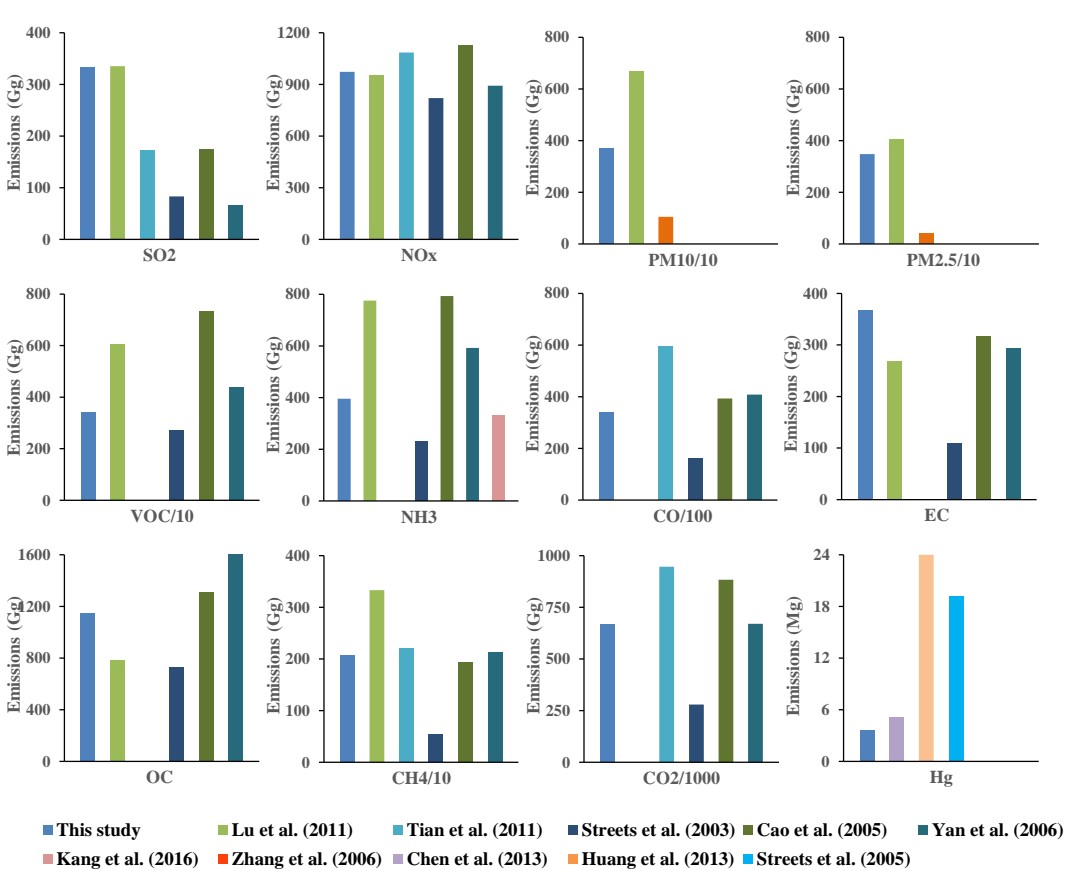

**Figure 10. Comparison of the emissions inventory derived by this study with the emissions estimated by previous research.**