# Peer review of "A comprehensive biomass burning emission inventory with high spatial and temporal resolution in China"

_Atmospheric Chemistry and Physics, 2016_

## Referee Comment (RC1) · Anonymous Referee #1 · 15 Aug 2016

"general comments" In general, this study quantified a comprehensive biomass burning emissions including indoor and outdoor biomass burning emissions and fits the requirement of East Asia emissions assessment. However, it is difficult to find anything new to the scientific world. Since there is nothing new on used methods or data. And some methods and data usually reduce some errors and uncertainties.

"specific comments" The biomass burning includes firewood burning and in-field burning. There is another large contributor of human waste burning that should not be overlooked, especially in rural area of the developing countries. Since this is a comprehensive inventory, I suggested the authors can add this part. Wiedinmyer, C.; Yokelson, R. J.; Gullett, B. K.Global emissions of trace gases, particulate matter, and hazardous

air pollutants from open burning of domestic waste Environ. Sci. Technol. 2014, 48 (16) 9523–9530, DOI: 10.1021/es502250z Shi, Y., Matsunaga, T., and Yamaguchi, Y., High-resolution mapping of biomass burning emissions in three tropical regions, Environmental Science and Technology, Environ. Sci. Technol., 2015, 49 (18), pp 10806–10814. DOI: 10.1021/acs.est.5b01598

2.2.3 Biomass burning of forest/grassland fires The estimation of burned biomass in this very simple method have lots of problems. AR is the damaged area, in fact, it is the burned area, they are far different. Burned area data were usually derived from satellite data for such a large area of China. It is basically wrong that the authors used the statistics data to allocate them according to the fire counts. Since fire counts does not linearly correspond with the burned area. Please refer to MCD64A1 burned area product with 500 m resolution, which has been validated in many ecosystems. Fire consumes great amount of biomass when burning happens. And this biomass usually cost several years to recover to its previous condition. The authors failed in considering the reduction of biomass of this month due to fire as the beginning of the next month. Therefore, I suggested the authors should consider the reduction of biomass when it is used as the base for the next month. Besides, the biomass used in this study within each province are even. The biomass density was constrained by precipitation, air temperature and vapor pressure controlled gross primary production, respiration, etc. The used constant data cannot reflect the heterogeneity of the biomass. Combustion factor is strongly controlled by fuel types and moisture conditions and vary widely from pixel to pixel. The authors set the combustion factors for each fuel type as constant, which cannot depict the differences between moisture and dry fuels types. Since dry fuels can burn mostly while wet fuels burn less completely. I suggested the authors should consider the moisture condition of the fuel types and revised them into spatial and temporal variable parameters, which can really reflect the condition of each pixel. Since there are many available satellite products on burned area, ecosystem productivity model estimated biomass density and moisture condition, we really do not suggest the authors used the county-level data and allocate them into each pixel. The

estimation of biomass burning emissions by using the bottom-up method should use the pixel-based high-resolution datasets to describe its process. van der Werf, G. R., Randerson, J. T., Giglio, L., Collatz, G. J., Mu, M., Kasibhatla, P. S., Morton, D. C., DeFries, R. S., Jin, Y., and van Leeuwen, T. T.: Global fire emissions and the contribution of deforestation, savanna, forest, agricultural, and peat fires (1997–2009), Atmos. Chem. Phys., 10, 11707-11735, doi:10.5194/acp-10-11707-2010, 2010.

As for 2012, this study estimated 665.989Tg $CO_2$ list in Table 7 and there is almost no forest and grassland fires based on Figure 2 $CO_2$ chart and Figure 8 $CO_2$ chart. But actually, by using the ecosystem production model integrated with fire emission process, Global Fire Emissions Database v4 (GFED4) estimated outdoor biomass burning emissions (forest, savanna and agriculture) with 54 Tg $CO_2$ in 2012. Authors should explain this large differences due to their used methods and datasets.

Figure 7: In North China Plain, there are many polygons in blue with small amounts of PM2.5, which were far lower than their surrounding areas, the sudden reduction of these polygon values may be attributable to the used county-level data, we suggested the authors changed this dataset since it is unreasonable of these polygons with small amount.

Figure 10: This study estimates SO2, NOx, which are comparable to Lu et al., (2011). What is the reason for the underestimation of PM10, VOC, NH3, CH4 and overestimation of EC and OC relative to Lu et al.,(2011). Why these emissions agreed well in NOx, but large differences on other gases?

---

## Referee Comment (RC2) · Anonymous Referee #2 · 30 Aug 2016

Review of "A comprehensive biomass burning emission inventory with high spatial and temporal resolution in China" This study addresses what the authors believe are key weaknesses of current biomass burning emission inventory for China: 1. Missing sources (in particular firewood), 2. Incomplete or source specific EF, 3. Estimates of crop straw utilization and it's variability across regions/provinces, 4. Province level resolution of available inventories is not appropriate for modeling / evaluation emission impacts of atmospheric chemistry, climate or health.

SPECIFIC COMMENTS:

P1, Ln 25-26 "Corn, rice and wheat represent the major crop straws, with their total emission contribution exceeding 80% for each pollutant." Please clarify for which pollutants ("each pollutant") crop straw combustion accounts for 80% of total emissions. Do they refer to SO2, CO, CH4, and Hg? Or all pollutants listed in lines 20-21? Do the authors mean that the combined emissions of corn, rice, and wheat account for 80% of the inventory total emissions of specific pollutants?

Statements regarding emissions of EC and NH3 are contradictory, please clarify / correct: P1, Ln 24-25 "...firewood contributes most to EC and NH3 emission." P1, Ln 26-27: "Corn straw burning has the greatest contribution to EC, NOx and SO2 emissions; rice straw burning is dominant contributor to CO2, VOC, CH4 and NH3 emissions"

P1, Ln 31: "The temporal distribution shows that higher emissions occurred in April, September, and October during the whole year." This statement is unclear. Do the authors mean that the combined emissions from April, September, and October exceeded emissions for the remainder of the year? Please clarify.

P2, Ln3: "haolocarbon" important to secondary chemistry?

P2, Ln17 change "critical" to "significant"

P2, L2: "The amount of straw outdoor burning in China in 2009 is 0.215 billion tons (MA,2011)." This should statement should be qualified e.g. change: "is" to "was estimated as". Also, please provide a couple lines describing the data and the source of data, since the citation is not readily accessible.

P4 L3: Please provide a citation for the "energy statistical yearbook" and provide a brief (1 sentence) description of the yearbook. P4, L7 "statistical yearbook" is this the "energy statistical yearbook"? Please clarify. P4, L11 "yearbook" is this the "energy statistical yearbook"? Please clarify.

P5, L1 delete "including" before "domestic combustion" The authors use the term "field burning" to refer to burning of crop residue in the field, and grassland and forest fires. The widely used terminology in biomass burning research refers to in-field crop residue burning, grassland, and forest fires as "open burning". The author should use this terminology not "field burning" when referring to the combined crop residue, grassland, and forest burning. The use of "field burning" by the authors is inconsistent with accepted terminology and is confusing, a forest is not a "field".

P5, L11 What is the unit "a" in Mg/a?

P5, L23: Please clarify the source of data in Figure S1 (statistical yearbook?) and note that it is prefecture level.

Table 3. Are the superscripts denoting reference for Nk also the references for Dk and CEk? Please clarify.

Section 2.2.2 Firewood. Please clarify exactly which regression equation(s) were used to predict firewood consumption.

P6, L26 What is the unit "a" in ton/a? Is a = "annum"? If so I recommend using year ("yr") instead. Also, is this metric ton?

P6, L26: "damaged area" = "burned area"? I assume by "damage area" the authors mean burned area. I recommend the authors use "burned area" instead of "damaged area" for consistency with biomass burning literature and accepted terminology. From an ecological standpoint a burned forest or grassland is not generally "damaged" since fire is a natural part of many if not most ecosystems.

P6, L29: More details are needed on the data used and the method used to determine the spatial and temporal distribution of burned area. 1. Describe the "damaged area" data from NBSC (2013c) and NBSC (2013d). a. Is the data county level, prefecture level, province level? b. What is the time resolution of the data (annual, monthly, daily)? c. How was the data collected, e.g. is it based on administrative reports from local land management agencies? d. Does the dataset include both wildfires and fires used for ecosystem management, e.g. clearing logging debris or rangeland burning for grazing? 2. Provide a web link to where the references NBSC (2013c) and NBSC (2013d) can be accessed.

P6, L31: A more detailed description of the forest and grassland biomass and combustion efficiency data is needed: 1. Provide a web link to where the references Tian et al. (2003), Lu et al. (2011), and EPD (2013) can be accessed. If they are not accessible, the work is not reproducible. 2. What forest components does the biomass number listed in Table 4 include? Organic soil/duff, litter, down dead wood, understory herbs and shrubs? 3. Is the forest biomass derived from forest inventory data? 4. The combustion efficiency of forests is 0.1 – 0.2. Is this because much of the forest biomass numbers include boles and branches of live trees which do not burn? 5. Does grassland category include shrub lands? 6. Please comment on how the biomass loadings and consumption estimates used in this study compare with those used in previous global emission inventories (e.g. GFED, van der Werf et al., 2010; FiNN, Wiedinmyer et al., 2011) and surveys of fuel consumption (e.g. van Leeuwen et al., 2014) and studies of grassland biomass in China (e.g. Ni, 2004; Ma et al. 2016; . Zhao et al., 2014) 7. The value of 1800 kg/ha (180 g/m2) used in this study compares reasonably well with Ni (2004) study of northern China Northern (325.5 g/m2).

P7, L1 Units for grassland biomass are kg/ha while other quantities are listed as kg/hm2. While these units are equivalent, please be consistent by using either ha or hm2 throughout the manuscript.

Table 4. Note the units for forest biomass.

P7, L10: Provide a web link to where the references EOCAIY (2013) and NBSC (2013b) can be accessed

Section 2.3 EFs The VOC emissions factors for forest, grasslands, open residue burning, and feces burning seem quite low compared to those reported in extensive reviews such as Akagi et al. (2011). I imagine the difference is that the VOC category in Table 5 & 6 include only a subset of VOC present in biomass smoke and measured in other studies. Please comment in the differences.

Section 2.4 Spatial Distribution Is the land use data of Ran et al. (2010) publicly available? If so, please provide a web link where it may be accessed.

Figure 2 is not very useful. It should be replaced with or augmented with table that provides the total emissions and percent of each species by source.

Results and discussion Please include a table providing total annual fuel consumption and emissions for the 12 species in Table 7 by source. This will is needed to compare current paper to previous studies that may have focused on only a subset of sources.

P9 L15-16 Please not the total annual burned area of forest and grasslands.

Figures 4 & 5 are difficult to read and the data would be better presented as tables, perhaps in the supplement.

P13, L13-14: Are specific crops typically burned harvest season, sowing season, or both? Or does it vary by region and practice?

Section 3.6 Please describe how parameters were estimated for the PDFs used in the Monte Carlo simulation.

Figure S4. Please note the data sources used to derive the non-carbon PM components.

TECHINCAL The manuscript contains many minor grammatical errors, here are a few: P13, Ln3-4: Change "This is because the main contribution of these species emission sources is from straw outdoor burning" to "This is because straw outdoor burning is the main source for these species" P13, Ln 4: change "The outdoor burning straw mainly occurs in..." to "The outdoor burning of straw occurs mainly in..." P13, L19 insert "a" between "have" and "relatively" and change "peak" to "peaks" P13, L20: change "discrepancies" to "differences" P13, L28: change "while" to "where" P14, L7: change 'peak" to "peaks" P14, L14: Change "Besides" to "Additionally" P16, L23-25 Sentence beginning "More localized EF of..." is jumbled and must be rewritten

REFERENCES Akagi et al. (2011) Atmos. Chem. Phys. 11, 4039-4072. Ma, A. et

al. Carbon storage in Chinese grassland ecosystems: Influence of different integrative methods. Sci. Rep. 6, 21378; doi: 10.1038/srep21378 (2016). Ni (2004) Plant Ecology, 174, 217-234. van der Werf et al. (2010) Atmos. Chem. Phys., 10, 11707-11735 van Leeuwen et al. (2014) Biogeosciences, 11, 7305-7329. Wiedinmyer et al. (2011) Geosci. Model Dev., 4, 625-641 Zhao et al. (2014) Remote Sens. 6, 5368-5386.

Please also note the supplement to this comment:
http://www.atmos-chem-phys-discuss.net/acp-2016-560/acp-2016-560-RC2-supplement.pdf

---

## Author Comment (AC1) · 2 Nov 2016

We have carefully taken all comments into consideration in revision. Please see the attached point-by-point responses.

Please also note the supplement to this comment: http://www.atmos-chem-phys-discuss.net/acp-2016-560/acp-2016-560-AC1-supplement.zip

---

## Author Response (AR1)

**Authors' Responses to Reviewer Comments**

Manuscript: A comprehensive biomass burning emission inventory with high spatial and temporal resolution in China (Ref. No.: acp-2016-560)

We are very grateful to the respectful referee for your careful and insightful review. Your comments and suggestions have contributed greatly to improving our paper. Our point-by-point responses to your comments are listed as follows.

**Anonymous Referee #1**

General comments: In general, this study quantified a comprehensive biomass burning emissions including indoor and outdoor biomass burning emissions and fits the requirement of East Asia emissions assessment. However, it is difficult to find anything new to the scientific world. Since there is nothing new on used methods or data. And some methods and data usually reduce some errors and uncertainties.

**Response:**

We thank you for your comments, which were very helpful for revising and improving our paper. Generally, two approaches are employed for developing a biomass burning emission inventory: a "top-down" approach and a "bottom-up" approach. Regarding a consideration of new methods, we point out the overwhelming importance of employing refined and updated data to develop a more accurate emission inventory with a much higher temporal-spatial resolution. In fact, highly detailed emission information is extremely important for investigating the causes of air pollution (e.g., air quality modeling) and developing a targeted strategy for the control of pollution accompanying biomass burning, particularly for conditions prevalent in recent years owing to the altered pattern of energy consumption in rural areas and the increasing pollution problems associated with dramatic urbanization caused by the economic development in China. Currently, as far as we are aware, few studies have developed a comprehensive biomass burning emission inventory in China, particularly after 2007,

because of a lack of detailed statistical data regarding firewood consumption. Furthermore, the source-specific emission factors (EFs) used in emission estimation need to be updated based on a systematic combination of localized measurements conducted in China. In addition to EFs, the activity data is also a key factor for improving an emission inventory. Moreover, several key sources of information related to biomass emission estimation must be updated, such as the proportion of crop straw domestic combustion and in-field burning, and the uneven temporal distribution coefficient, which reflects recent conditions in different regions of China. In fact, the current biomass burning emission inventory for China is generally at a province-level resolution because detailed activity data is not publicly available. It is obvious that the resolution of activity data determines the preliminary resolution of an emission inventory. An emission inventory with a coarse preliminary resolution could result in greater uncertainty in grid emissions generated according to source-based gridded spatial surrogates (e.g., population) using GIS technology.

The main contributions of our work are summarized as follows.

First, a comprehensive biomass burning emission inventory, including crop straw domestic combustion and in-field burning, firewood and livestock excrement combustion, and forest and grassland fires for mainland China was developed based on detailed data (county-level data and satellite data) and updated source-specific EFs for the first time.

Second, a range of important information representing the recent status for emissions estimation in China were obtained from field investigation, a systematic combination of the latest research, and regression analysis (e.g., province-specific straw domestic combustion/in-field burning ratio, detailed firewood combustion quantities, and non-uniform temporal distribution coefficient).

Third, the high-resolution temporal (monthly and daily) and spatial (1 km $\times$ 1 km) biomass burning emission inventory presented in this study includes major precursors of complex pollution systems, greenhouse gases, and heavy metals released from biomass burning such as $SO_2$, $NO_x$, $PM_{10}$, $PM_{2.5}$, VOCs, $NH_3$, CO, EC, OC, $CO_2$, $CH_4$, and Hg with further breakdowns of $PM_{2.5}$ particles and VOCs.

In addition, we have carefully taken the reviewer's remaining comments related to methods and data into consideration during the revision of our paper (e.g., emission estimation of forest and grassland fires). Please see the following point-by-point responses.

"Specific comments"

The biomass burning includes firewood burning and in-field burning. There is another large contributor of human waste burning that should not be overlooked, especially in rural area of the developing countries. Since this is a comprehensive inventory, I suggested the authors can add this part. Wiedinmyer, C.; Yokelson, R. J.; Gullett, B. K.Global emissions of trace gases, particulate matter, and hazardous air pollutants from open burning of domestic waste Environ. Sci. Technol. 2014, 48(16) 9523–9530, DOI: 10.1021/es502250z Shi, Y., Matsunaga, T., and Yamaguchi, Y., High-resolution mapping of biomass burning emissions in three tropical regions, Environmental Science and Technology, Environ. Sci. Technol., 2015, 49 (18), pp10806–10814. DOI: 10.1021/acs.est.5b01598

**Response:**

We thank you very much for your suggestion. In view of your proposal, we have conducted an extensive literature review.

First, in this study, we developed a detailed emission inventory of biomass burning, including crop straw domestic combustion and in-field burning, firewood and livestock excrement combustion, and forest and grassland fires. These are important sources of biomass burning that are considered in the literature, including studies focused on domestic combustion and open burning in China (He et al., 2011; Chen et al., 2013; Zhang et al., 2013) and other regions (Shon et al., 2015; van der Werf et al., 2010; Bhardwaj et al., 2016) in recent years.

Second, with the increasing economic development and income of rural residents, tremendous changes have taken place in the pattern of rural resident consumption. Industrial products are increasingly used in the lives of rural residents. Therefore, the composition of rural human waste tends to be largely a product of urbanization Human

waste in rural areas of China is mainly inorganic garbage, such as waste plastics, waste paper, waste glass, and hazardous waste, and some organic garbage (e.g., crop residue waste and kitchen waste) (Ma et al., 2002; Chai et al., 2012; Yan et al., 2014). More than 80% of the garbage in rural areas is discarded carelessly without any further processing (Guan and Qiu, 2008; Wang et al., 2011; Yao et al., 2009).

Third, among the sources of rural human waste, the primary biomass wastes that may be burnt in the rural areas of China are crop residue waste and livestock excrement (Tian et al., 2011; Zhou et al., 2013; Zhang et al., 2013). These wastes have been considered in our study.

Fourth, we have added a review of studies focused on the emissions of human waste burning, including the two articles recommend by the reviewer, such as Park et al. (2013), Wiedinmyer et al. (2014), Shi et al. (2015), and Maasikmets et al. (2016). Few EFs of human waste burning are categorized into specific waste types (particularly with regard to biomass waste). In addition, the specific waste production in rural areas is difficult to obtain currently in China. If we attempt to estimate the emissions of human waste burning based on non-specific EFs and waste production in our biomass burning study, the results of the biomass burning emission inventory will be overestimated due to introduced emissions that are independent of biomass burning.

Owing to the reasons discussed above, we did not estimate the emissions of human waste burning in the current study. Further studies on the specific characteristics of human waste burning emission must be conducted, which would then allow the development of an elaborate emission inventory of waste burning related to biomass based on detailed EFs and relevant activity data investigations.

2.2.3 Biomass burning of forest/grassland fires. The estimation of burned biomass in this very simple method have lots of problems. AR is the damaged area, in fact, it is the burned area, they are far different. Burned area data were usually derived from satellite data for such a large area of China. It is basically wrong that the authors used the statistics data to allocate them according to the fire counts. Since fire counts does not linearly correspond with the burned area. Please refer to MCD64A1 burned area

product with 500 m resolution, which has been validated in many ecosystems. Fire consumes great amount of biomass when burning happens. And this biomass usually cost several years to recover to its previous condition. The authors failed in considering the reduction of biomass of this month due to fire as the beginning of the next month. Therefore, I suggested the authors should consider the reduction of biomass when it is used as the base for the next month. Besides, the biomass used in this study within each province are even. The biomass density was constrained by precipitation, air temperature and vapor pressure controlled gross primary production, respiration, etc. The used constant data cannot reflect the heterogeneity of the biomass. Combustion factor is strongly controlled by fuel types and moisture conditions and vary widely from pixel to pixel. The authors set the combustion factors for each fuel type as constant, which cannot depict the differences between moisture and dry fuels types. Since dry fuels can burn mostly while wet fuels burn less completely. I suggested the authors should consider the moisture condition of the fuel types and revised them into spatial and temporal variable parameters, which can really reflect the condition of each pixel. Since there are many available satellite products on burned area, ecosystem productivity model estimated biomass density and moisture condition, we really do not suggest the authors used the county-level data and allocate them into each pixel. The estimation of biomass burning emissions by using the bottom-up method should use the pixel-based high-resolution datasets to describe its process. van der Werf, G. R., Randerson, J. T., Giglio, L., Collatz, G. J., Mu, M., Kasibhatla, P. S., Morton, D. C., DeFries, R. S., Jin, Y., and van Leeuwen, T. T.: Global fire emissions and the contribution of deforestation, savanna, forest, agricultural, and peat fires (1997–2009), Atmos.Chem. Phys., 10, 11707-11735, doi:10.5194/acp-10-11707-2010, 2010.

**Response:**

We thank you for pointing out these problems with the methods and data employed in the emission estimation of forest and grassland fires. As the reviewer mentioned, the damaged area is different from the burned area. Our description in the original manuscript was unfortunately confusing, and we have revised the description to clarify our meaning. Furthermore, as the reviewer pointed out, the burned area should be

obtained from satellite data rather than statistical data and allocation based on the fire counts. According to the reviewer's suggestion, we have updated our methods and data employed in this study, and have re-calculated the pixel-based emission of forest and grassland fires using the bottom-up method. The burned area data is derived from the MCD64A1 burned area satellite product with a 500 m resolution. As for the biomass density, we agree with the reviewer that the value within each province should not be constant because this cannot reflect the heterogeneity of the biomass. The vegetation type of each pixel where forest and grass fires have occurred can be determined according to the land cover data. Therefore, the biomass density was determined according to the vegetation type of different provinces based on localized studies in China. As for the biomass density reduction over continuous months due to fires, we compared the distribution of burned areas due to forest and grassland fires in different months for 2012 in China, and found that little overlap between burning areas is observed among various months. This is mainly due to the fact that forest and grassland fires are accidental events, which can occur only with the confluence of three elements, i.e., forest and grassland fuel, fire, and meteorological conditions (Wei et al., 2014). Therefore, we did not consider the reduction of biomass density in different months. A similar consideration of biomass density can be found in recent studies (Song et al., 2009; Zhang et al., 2013; Qiu et al., 2016). As for the combustion factor (CF), we also noticed that CFs for each fuel type should not be constant. Considering the information we could obtain, we used specific CFs according to the vegetation type in each pixel based on literature review (Michel et al., 2005; Kasischke et al., 2000; Hurst et al., 1994). He et al. (2011), Zhang et al. (2013), and Chen et al. (2013) used a similar approach to consider CFs. The methodology employed for the estimation of the biomass burning emission owing to forest and grassland fires in Sect. 2.2.3 of the manuscript was modified accordingly, as follows.

"2.2.3 Estimation of biomass burning emission of forest/grassland fires

The burning mass of forest/grassland can be calculated from the annual mass of forest/grassland burned (kg/yr) as follows:

$$A = \sum_{j=1}^{10} BAx,j \times FLx,j \times CFj, \tag{3}$$

where subscripts $j$, and $x$ represent the land cover type, and location, respectively, $BA_{x,j}$ is the burned area (m$^2$) at $x$ where belongs to $j$, $FL_{x,j}$ is the biomass fuel loading (the aboveground biomass density in this study; kg/m$^2$) at $x$ where belongs to $j$, and $CF_j$ is the combustion factor (the fraction of burned aboveground biomass) at $j$.

Burned area data for 2012 were derived from the moderate-resolution imaging spectroradiometer (MODIS) direct broadcast burned area product (MCD64A1; http://modis-fire.umd.edu). This product employs an automated algorithm for mapping MODIS post-fire burned areas, and deriving the approximate burn date within each burn cell combined with surface reflectance, land cover products, and daily active fires. The MCD64A1 product has a primary spatial resolution of 500 m and a temporal resolution of 1 month. The extent of burning over a Julian day and its temporal uncertainty are specified for each burn cell. The burned areas within an approximate Julian day can be extracted from the original 500 m resolution map.

Earlier research on the estimation of $FL$ values for forest and grassland typically employed an averaged value of aboveground biomass density. However, these values do not well reflect the spatial variations of $FL$ for each vegetation type. In this study, numerous local $FL$ were collected for each province and vegetation type. The type of vegetation burned in each pixel was determined by the 1 km resolution MODIS Land Cover product produced by Ran et al. (2010). We considered 10 vegetation types as forest and grassland (i.e., evergreen needleleaf forest, evergreen broadleaf forest, deciduous needleleaf forest, deciduous broadleaf forest, mixed forest, closed shrublands, open shrublands, woody savannas, savannas, and grassland). The values of $FL$ employed in this study are listed in Table 4. As for $CF$, it has usually been set as a constant in previous literature. In our paper, $CF$ values were collected for each vegetation type, and the $CF$ in each pixel was determined by the MODIS Land Cover product and the $CF$ of typical vegetation. The $CF$ of forest, closed shrublands, open shrublands, woody savannas, and grassland were set as 0.25, 0.5, 0.85, 0.4, and 0.95, respectively (Michel et al., 2005; Kasischke et al., 2000; Hurst et al., 1994)."

The corresponding figures and tables have been revised.

As for 2012, this study estimated 665.989Tg CO2 list in Table 7 and there is almost no forest and grassland fires based on Figure 2 CO2 chart and Figure 8 CO2 chart. But actually, by using the ecosystem production model integrated with fire emission process, Global Fire Emissions Database v4 (GFED4) estimated outdoor biomass burning emissions (forest, savanna and agriculture) with 54 Tg CO2 in 2012. Authors should explain this large differences due to their used methods and datasets.

**Response:**

We thank you very much for your comment. According to the reviewer's suggestion, we have re-estimated forest and grassland fire emission based on the bottom-up approach and high-resolution satellite data. The total annual forest and grassland fire emission of $CO_2$ was determined in this way to be 10.9 Tg, where it was 1.68 Tg in the original manuscript. The total biomass burning emission of $CO_2$ is therefore 675 Tg. Figures 2, 8, and other relevant figures have been revised accordingly. As for the large differences between the $CO_2$ emissions reported in the Global Fire Emissions Database v4 (GFED4) and those reported in the present study, these could be attributed to the following reasons.

First, the biomass burning emission inventory in this study included the crop straw domestic combustion, an important biomass burning source in China, with a contribution accounting for 68% of the total $CO_2$ emission. This source was not included in the GFED4.

Second, the emissions of crop straw open burning were estimated in the present study based on the specific EFs and the amount burned for each type of crop straw. The emission in GFED4 was estimated according to the burned area based on a constant EF. The agricultural burned area was derived from the MODIS MCD64A1 product (~500 m resolution). Despite the efforts made to improve the direct broadcast mapping algorithm employed in the MCD64A1 product, the product has a minimum detectable burn area size, which is greater than the size of many agriculture waste burn sites. Therefore, numerous small and scattered agricultural fires would not be detected

(McCarty et al., 2007; Giglio et al., 2013; Shi et al., 2015). This is particularly the case in China, where the open burning of crop residue tends to be conducted by individual families (Liu et al., 2015), resulting in agricultural burning that often occurs over small areas, which is then undetected. The results in the present study were compared with the results of other research. The $CO_2$ emission of agricultural crop residue open burning in China in 2012 was estimated at 184 Tg by Sun et al. (2016), which is similar to the 207 Tg value obtained in our study. In addition, the total $CO_2$ emission by biomass burning published in most literature (Cao et al., 2005; Yan et al., 2006; Lu et al., 2011; Tian et al., 2011) is similar to the value obtained in our work, with differences ranging from 0.7% to 40.0%. This comparison indicates that the $CO_2$ emission estimated in our paper is relatively credible.

Figure 7:

In North China Plain, there are many polygons in blue with small amounts of $PM_{2.5}$, which were far lower than their surrounding areas, the sudden reduction of these polygon values may be attributable to the used county-level data, we suggested the authors changed this dataset since it is unreasonable of these polygons with small amount.

**Response:**

We thank you very much for your comment. Figure 7 presents a 1 km × 1 km grid reflecting biomass burning emission, including crop straw domestic combustion and in-field burning, firewood and livestock excrement combustion, and forest and grassland fires. Actually, the North China Plain includes several urban areas surrounded by suburban and rural areas. The main fuel used in these urban areas is commodity energy (e.g., coal, natural gas, and electricity) rather than biomass fuel. Therefore, these urban areas produce little biomass burning emission. However, previous studies could not account for these actual conditions because the gridded emission was allocated from an emission inventory with coarse preliminary resolution (e.g., a provincial or prefectural level resolution prior to spatial allocation) based on gridded surrogates (e.g., rural population). In the present study, the grid emission was allocated from an emission

inventory with improved preliminary resolution (i.e., county-level resolution), which could reflect the low use of biomass fuel (e.g., crop straw, firewood, and livestock excrement) within several urban regions of the North China Plain. Consequently, the use of the relatively high-resolution emission inventory allocation could better represent the actual conditions.

We have emphasized the description in Sect. 3.3 (Lines 1-15 on Page 16) of the manuscript:

"The scarce population and crop yield in part of southwest and northwest areas, and lower agricultural activity in downtown areas result in lower emissions. Specially, some urban areas in the north China Plain are surrounded by suburban and rural areas, the main fuel used in these urban areas is commodity energy. Besides, there is no agricultural activity in the field. Therefore, these areas produce little biomass burning emission. However, error will be brought in grid emissions if they are allocated from the emission inventory at coarse preliminary resolution (e.g., provincial or prefectural resolution before spatial allocation) based on the gridded surrogates (e.g., rural population). Consequently, gridded emissions, which were obtained through spatial allocation from emission inventory at county resolution, could better represent the actual situation."

Figure 10: This study estimates SO2, NOx, which are comparable to Lu et al., (2011). What is the reason for the underestimation of PM10, VOC, NH3, CH4 and overestimation of EC and OC relative to Lu et al.,(2011). Why these emissions agreed well in NOx, but large differences on other gases?

**Response:**

We thank you very much for your question. The differences in the cited pollutant emissions between this study and that of Lu et al. (2011) are most likely attributable to the selection of EFs. The higher estimations of EC and OC relative to those obtained by Lu et al. (2011) are mainly due to our use of higher EC and OC EFs for crop straw and firewood domestic combustion. The EC and OC EFs employed in the present study were selected from the work of Li et al. (2007), which were measured in representative

rural areas across China to determine the characteristics of household biofuel combustion emission. The EFs employed in Lu et al. (2011) were constant values for different crop straw types, and were derived from measurements conducted outside of China (Reddy and Chandra, 2002). The lower values of $CH_4$ and $NH_3$ emissions relative to those obtained by Lu et al. (2011) is mainly due to the lower EFs employed in the present study for crop domestic burning. The crop domestic EFs for $CH_4$ and $NH_3$ employed in Lu et al. (2011) were constant values for different crop straw types. The EFs used in the present study were specific for each type of crop straw, which were updated according to published reports of localized measurements conducted in China. The lower estimations of VOC and $PM_{10}$ relative to those obtained by Lu et al. (2011) are the result of the employment of lower EFs for in-field crop residue burning and firewood combustion. The VOC and $PM_{10}$ EFs employed in Lu et al. (2011) did not distinguish between different crop straw types, and were derived from measurements conducted outside of China (Street et al., 2003; Reddy and Chandra, 2002). The specific EFs for various crop straws employed in our study were derived from the Chinese guide for compiling atmospheric pollutant emission inventories for biomass burning published in 2014.

**Anonymous Referee #2**

General comments:

Review of "A comprehensive biomass burning emission inventory with high spatial and temporal resolution in China" This study addresses what the authors believe are key weaknesses of current biomass burning emission inventory for China: 1. Missing sources (in particular firewood), 2. Incomplete or source specific EF, 3. Estimates of crop straw utilization and it's variability across regions/provinces, 4. Province level resolution of available inventories is not appropriate for modeling / evaluation emission impacts of atmospheric chemistry, climate or health.

**Response:**

We thank you very much for your careful and insightful review. We have taken the following comments into consideration in revision. Please see the following point-by-point responses.

Specific Comments:

P1, Ln 25-26 "Corn, rice and wheat represent the major crop straws, with their total emission contribution exceeding 80% for each pollutant." Please clarify for which pollutants ("each pollutant") crop straw combustion accounts for 80% of total emissions. Do they refer to $SO_2$, CO, $CH_4$, and Hg? Or all pollutants listed in lines 20-21? Do the authors mean that the combined emissions of corn, rice, and wheat account for 80% of the inventory total emissions of specific pollutants?

**Response:**

We thank you very much for your comment. We are truly sorry for the confusing sentence. In this sentence, each pollutant refers to all pollutants listed in lines 18-19.

We have revised this sentence in the revised version in Lines 26-27 on Page 1:

*"Corn, rice and wheat represent the major crop straws. The combined emissions of these three straw types account for 80% of the total straw burned emissions for each specific pollutants mentioned in this study."*

Statements regarding emissions of EC and $NH_3$ are contradictory, please clarify / correct: P1, Ln 24-25 "...firewood contributes most to EC and NH3 emission." P1, Ln 26-27: "Corn straw burning has the greatest contribution to EC, NOx and $SO_2$ emissions; rice straw burning is dominant contributor to $CO_2$, VOC, $CH_4$ and $NH_3$ emissions.

**Response:**

We thank you very much for your comment. We are truly sorry about this unclear description. This sentence means that firewood contributes most to EC and $NH_3$ emission compared with other sources (indoor straw, In-field crop residue, livestock excrement, forest and grassland fire). As for the various crop straw types, corn straw burning has more contribution to EC, NOx and $SO_2$ emissions, and rice straw burning has more contribution to $CO_2$, NMVOC, $CH_4$ and $NH_3$ emissions compared with other straw types. We have revised this sentence in the revised version in Line 27 on Page 1 and Lines 1-2 on Page 2:

*" As for the straw burning emission of various crops , corn straw burning has the largest contribution to EC, NOx and $SO_2$ emissions; rice straw burning has higher contribution to $CO_2$, NMVOC, $CH_4$ and $NH_3$ emissions; wheat straw burning has higher contribution to CO and Hg emissions."*

P1, Ln 31: "The temporal distribution shows that higher emissions occurred in April, September, and October during the whole year." This statement is unclear. Do the authors mean that the combined emissions from April, September, and October exceeded emissions for the remainder of the year? Please clarify

**Response:**

We thank you very much for your comment. We are truly sorry for the confusing sentence. This sentence means that as for the emission from each month, these months

have higher emission. According to our calculation, April, May, June and October are the top four months with higher emissions, due to the in-field crop residue burning. While as for EC, the emission in February, January, October and December are relatively higher due to the biomass domestic burning in heating season. We have made the corresponding revision in the revised manuscript in the P2L6-7:

*"...April, May, June and October are the top four months with higher emissions, due to the in-field crop residue burning. While as for EC, the emission in February, January, October and December are relatively higher due to the biomass domestic burning in heating season."*

P2, Ln3: "haolocarbon" important to secondary chemistry?

**Response:**

We thank you very much for your comment. We are truly sorry for the confusing description. Haolocarbon is unimportant to secondary chemistry. Ethylene, propylene, toluene, mp-xylene and ethyl benzene are major species of NMVOC, which are important to secondary chemistry. We have made the corresponding modification in the revised manuscript in Lines 10-12 on Page 2, Lines 18-25 on Page 18 and Lines 18-19 on Page 21:

Lines 10-12 on Page 2*: "The species with relatively higher contribution to NMVOC emission include ethylene, propylene, toluene, mp-xylene and ethyl benzene, which are key species for the formation of secondary air pollution."*

*Lines 18-25 on Page 18: "The total NMVOC emission is 3474 Gg in this study. The alkenes are the major contributor of biomass burning NMVOC emissions. The contribution of alkenes to the total NMVOC emission is approximately 34%, more than that of alkane (28%), aromatics (24%), alkynes (13%) and others (1%). Among these species, ethylene, acetylene, propylene and 1-butylene are the major species of alkenes and alkynes, with the total contribution accounting for 40.1%. Ethane, n-propane, n-butane, and n-dodecane are the main species of alkanes, with the total contribution accounting for 14.0%. Benzene, toluene, styrene, mp-xylene and ethyl benzene are the major species of aromatics, with the total contribution of 16.6%. Several species*

*mentioned above are key for the formation of secondary air pollution, such as ethylene, propylene, toluene, mp-xylene and ethyl benzene (Huang et al., 2011). It illustrates that the biomass burning emission control is urgently needed for the air quality improvement. Detailed NMVOC species emission is shown in the Supplement (Fig. S5)."*

*Lines 18-19 on Page 21: "Several species with higher contribution to NMVOC (e.g., ethylene, propylene, toluene, mp-xylene and ethyl benzene) are key species for the formation of secondary air pollution."*

P2, Ln17 change "critical" to "significant"

**Response:**

We thank you very much for your suggestion. We have made the revision in revised manuscript in Line 24 on Page 2:

*"...Biomass burning is also a significant source of greenhouse gases such as methane ($CH_4$) and carbon dioxide ($CO_2$) ..."*

P2, L2: "The amount of straw outdoor burning in China in 2009 is 0.215 billion tons (MA, 2011)." This should statement should be qualified e.g. change: "is" to "was estimated as". Also, please provide a couple lines describing the data and the source of data, since the citation is not readily accessible.

**Response:**

Thanks very much for your comment. We have made the corresponding modification in the revised manuscript in Lines 11-12 on Page 3:

*"...The amount of in-field crop residue burning in China in 2009 was estimated as 0.215 billion Mg. The data is obtained from the government report on the investigation and evaluation of crop straw resources in various provinces in China (MA, 2011)."*

In addition, we have added the sources of the report in the reference list in the revised manuscript:

*"MA: Investigation and Evaluation Report on Crop Straw Resources in China, Ministry of Agriculture, 2011, available at http://www.kjs.moa.gov.cn/, (in Chinese)."*

P4 L3: Please provide a citation for the "energy statistical yearbook" and provide a brief (1 sentence) description of the yearbook. P4, L7 "statistical yearbook" is this the "energy statistical yearbook"? Please clarify. P4, L11 "yearbook" is this the "energy statistical yearbook"? Please clarify.

**Response:**

We thank you very much for your comment. We have added the corresponding citation, description and clarification. Detailed content in the revised manuscript was listed below.

*Lines 20-23 on Page 4: "Moreover, because of the lack of firewood consumption record in the China Energy Statistical Yearbook (NBSC, 2009-2015), few studies have developed a comprehensive biomass burning emission inventory in China in recent years. China Energy Statistical Yearbook provides official information on the energy construction, production and consumption, including the detailed firewood consumption in various regions. However, the firewood consumption data is no longer contained in the NBSC (2009-2015) since 2008."*

We have added the corresponding citation in the reference list in the revised manuscript:

*"National Bureau of Statistics of China (NBSC): China Energy Statistical Yearbook 2009−2015, China Statistics Press, Beijing, 2009−2015, available at http://www.stats.gov.cn/tjsj/tjcbw/, (in Chinese)."*

P4, L7 Here "statistical yearbook" refer generically to all statistical yearbook. Actually, the detailed firewood consumption could be not obtained from any other yearbook in addition to energy statistical yearbook.

P4, L11 Here "yearbook" refer generically to all statistical yearbook (e.g. China Statistical Yearbook, China Rural Statistical Yearbook, etc.) in China.

We have made the corresponding revision in Lines 25-28 on Page 4 and Lines 1-3 on Page 5:

*"...First, not all biomass burning sources have been included in recent years, especially since 2008, because of the lack of firewood consumption data in the various statistical yearbooks (e.g. China Energy Statistical Yearbook, China statistical*

*yearbook, China rural statistical yearbook). Second, the source-specific EFs used in emission estimation need to be updated based on the systematic combing of local tests in the latest research. Third, the proportion of crop straw domestic burning and in-field crop residue burning, which could reflect the recent conditions of different provinces in China needs to be investigated. Fourth, the current biomass burning emission inventory for China is generally at province resolution because detailed activity data cannot be directly obtained from the various statistical yearbooks in China…"*

P5, L1 delete "including" before "domestic combustion" The authors use the term "field burning" to refer to burning of crop residue in the field, and grassland and forest fires. The widely used terminology in biomass burning research refers to in-field crop residue burning, grassland, and forest fires as "open burning". The author should use this terminology not "field burning" when referring to the combined crop residue, grassland, and forest burning. The use of "field burning" by the authors is inconsistent with accepted terminology and is confusing, a forest is not a "field".

**Response:**

We thank you very much for your comment. We have made the corresponding modification:

  *"The biomass burning considered in this study is mainly divided into two categories, domestic combustion and open burning. Domestic combustion mainly involves crop straw, firewood and livestock excrement (mainly used in pastoral and semi-pastoral areas) burning. Open burning includes in-field crop residue burning, forest and grassland fire."*

Similar description has been revised through the full text in the revised manuscript.

P5, L11 What is the unit "a" in Mg/a?

**Response:**

We thank you very much for your comment. We have changed the unit "Mg/a" to "Mg/yr" in the manuscript. In addition, all the "a" in the unit have been changed to "yr" through the full text in the revised manuscript.

P5, L23: Please clarify the source of data in Figure S1 (statistical yearbook?) and note that it is prefecture level.

**Response:**

Thanks very much for your suggestion. The source of data in Figure S1 is from China Statistical Yearbook (NBSC, 2013b). We have added the corresponding content in the revised manuscript in Lines 24-26 on Page 6 and Lines 1-2 on Page 7, and in the supplement in Figure S1:

Lines 24-26 on Page 6 and Lines 1-2 on Page 7: *"There are currently no statistics on the amount of each crop yield at the county resolution ($P_{i,k}$) in various yearbooks in China. Therefore, in this study, we conducted a correlation analysis between grain yield and crop yield at prefecture resolution, and found a good correlation (R = 0.747, detailed analysis is provided in the Supplement, Fig. S1). The grain yield at prefecture resolution was summarized from China Statistical Yearbook in 2012 (NBSC, 2013b). The crop yield at prefecture resolution was summarized from statistical yearbooks edited by National Bureau of Statistics in 2012 for each province.…"*

*Figure S1 in the revised supplement:*

*"S1 The correlation between crop yield and grain yield at prefecture resolution.*

[Figure]

*Figure S1 The correlation between crop yield and grain yield at prefecture resolution.*
Note: [a] NBSC (2013); [b] a range of statistical yearbooks edited by National Bureau of Statistics in 2012 for each province.

Table 3. Are the superscripts denoting reference for $N_k$ also the references for $D_k$ and

CE$_k$? Please clarify.

**Response:**

Thank you very much for the comment. We are truly sorry for the missing reference of D$_k$ and CE$_k$. The reference for D$_k$ and CE$_k$ is He et al. (2015). We have add the corresponding reference in Table 3 on Page 38:

*Table 3 Residue-to-production ratio (N$_k$), dry matter fraction (D$_k$) and combustion efficiency (CE$_k$) of crop straw used in this study.*

| Crops | N$_k$ | D$_k$ [f] | CE$_k$ [f] |
|---|---|---|---|
| Corn | 1.269[a] | 0.87 | 0.92 |
| Wheat | 1.3[b] | 0.89 | 0.92 |
| Cotton | 3[b] | 0.83 | 0.9 |
| Sugar cane | 0.3[c] | 0.45 | 0.68 |
| Potato | 0.5[d] | 0.45 | 0.68 |
| Peanut | 1.5[b] | 0.94 | 0.82 |
| Rapeseed | 1.5[d] | 0.83 | 0.9 |
| Sesame | 2.2[d] | 0.83 | 0.9 |
| Sugar beet | 0.1[b] | 0.45 | 0.9 |
| Hemp | 1.7[e] | 0.83 | 0.9 |
| Rice | 1.323[a] | 0.89 | 0.93 |
| Soybean | 1.6[d] | 0.91 | 0.68 |

[a] *Zhang et al. (1990).* [b] *Bi et al. (2010).* [c] *Han et al. (2002).* [d] *NATESC (1999).* [e] *Gao et al. (2009).* [f] *He et al. (2015).*

Section 2.2.2 Firewood. Please clarify exactly which regression equation(s) were used to predict firewood consumption.

**Response:**

Thanks very much for your comment. According our correlation analysis between firewood consumption and other factors that may have a relationship with the firewood (rural population, gross agricultural and timber yield), we found that the rural population and firewood consumption have the best correlation relationship. Because the regression equation is various for the different historical years (Figure 1, 1998-2007), the regression analysis results was used to find the main factor which could be applied to calculate the detailed firewood consumption. Then the detailed firewood

consumption was estimated based on the rural population.

P6, L26 What is the unit "a" in ton/a? Is a = "annum"? If so I recommend using year ("yr") instead. Also, is this metric ton?

**Response:**

We thank you very much for your suggestion. We have changed the "a" to "yr" in the units through the full text in the revised manuscript. In addition, we have changed the "ton" to "Mg" in the units through the full text.

P6, L26: "damaged area" = "burned area"? I assume by "damage area" the authors mean burned area. I recommend the authors use "burned area" instead of "damaged area" for consistency with biomass burning literature and accepted terminology. From an ecological standpoint a burned forest or grassland is not generally "damaged" since fire is a natural part of many if not most ecosystems.

**Response:**

We thank you very much for your suggestion. We are truly sorry for the confusing description. In fact, the "damaged area" in the original manuscript means "burned area". We have made the corresponding modification through the full text in the revised manuscript.

P6, L29: More details are needed on the data used and the method used to determine the spatial and temporal distribution of burned area. 1. Describe the "damaged area" data from NBSC (2013c) and NBSC (2013d). a. Is the data county level, prefecture level, province level? b. What is the time resolution of the data (annual, monthly, daily)? c. How was the data collected, e.g. is it based on administrative reports from local land management agencies? d. Does the dataset include both wildfires and fires used for ecosystem management, e.g. clearing logging debris or rangeland burning for grazing? 2. Provide a web link to where the references NBSC (2013c) and NBSC (2013d) can be accessed.

**Response:**

We thank you very much for your comment.

1. The response about the "damaged area" data from NBSC (2013c) and NBSC (2013d):

1.a. The data of damaged area from NBSC (2013c) and NBSC (2013d) is at provincial-level;

1.b. The time resolution of the data is annual resolution;

1.c. The data was collected from the China Statistical Yearbook published by National Bureau of Statistics of China;

1.d. This dataset did not include the fires used for ecosystem management.

Considering the low temporal and spatial resolution of statistics data for burned area, we have updated our methods and data employed in this study, and have re-calculated the pixel-based emission of forest and grassland fire using the bottom-up method. The daily burned area data is derived from the moderate-resolution imaging spectroradiometer (MODIS) direct broadcast burned area satellite product (MCD64A1; http://modis-fire.umd.edu) with a primary spatial resolution of 500 m. Detailed description about the methodology employed for the estimation of the biomass burning emission owing to forest and grassland fire was listed in Sect. 2.2.3 of the revised manuscript:

*"2.2.3 Forest and grassland burning*

The burning mass of forest/grassland can be calculated from the annual mass of forest/grassland burned (Mg/yr) as Eq. (3):

$$A = \left( \sum_{j=1}^{10} BAx, j \times FLx, j \times CFj \right) \times 10^{-6},$$

(3)

where subscripts j, and x represent the land cover type, and location, respectively, $BA_{x,j}$ is the burned area ($m^2$) of land cover type j at x, $FL_{x,j}$ is the biomass fuel loading (the aboveground biomass density in this study; $g/m^2$) of land cover type j at x, and $CF_j$ is the combustion factor (the fraction of burned aboveground biomass) of land cover type j.

*Burned area data for 2012 were derived from the moderate-resolution imaging spectroradiometer (MODIS) direct broadcast burned area product (MCD64A1; http://modis-fire.umd.edu). This product employs an automated algorithm for mapping*

*MODIS post-fire burned areas, and deriving the approximate burn date within each burn cell combined with surface reflectance, land cover products, and daily active fires. The MCD64A1 product has a primary spatial resolution of 500 m and a temporal resolution of 1 month. The extent of burning over a Julian day and its temporal uncertainty are specified for each burn cell. The burned areas within an approximate Julian day can be extracted from the original 500 m resolution map.*

*Earlier research on the estimation of FL values for forest and grassland typically employed an averaged value of aboveground biomass density. However, these values do not well reflect the spatial variations of FL for each vegetation type. In this study, numerous local FL were collected for each province and vegetation type. The type of vegetation burned in each pixel was determined by the 1 km resolution MODIS Land Cover product produced by Ran et al. (2010). We considered 10 vegetation types as forest and grassland (i.e., evergreen needleleaf forest, evergreen broadleaf forest, deciduous needleleaf forest, deciduous broadleaf forest, mixed forest, closed shrublands, open shrublands, woody savannas, savannas, and grassland). The values of FL employed in this study are listed in Table 4. As for CF, it has usually been set as a constant in previous literature. In our paper, CF values were collected for each vegetation type, and the CF in each pixel was determined by the MODIS Land Cover product and the CF of typical vegetation. The CF of forest, closed shrublands, open shrublands, woody savannas, and grassland were set as 0.25, 0.5, 0.85, 0.4, and 0.95, respectively (Michel et al., 2005; Kasischke et al., 2000; Hurst et al., 1994)."*

2. The web link of the references NBSC (2013c) has been added in the reference list in the revised manuscript. The references NBSC (2013d) have been deleted in the revised manuscript:

*"National Bureau of Statistics of China (NBSC): China Statistical Yearbook 2013, China Statistics Press, Beijing, 2013c, **available at http://www.stats.gov.cn/tjsj/ndsj/**, (in Chinese)."*

P6, L31: A more detailed description of the forest and grassland biomass and combustion efficiency data is needed: 1. Provide a web link to where the references Tian et al. (2003), Lu et al. (2011), and EPD (2013) can be accessed. If they are not

accessible, the work is not reproducible. 2. What forest components does the biomass number listed in Table 4 include? Organic soil/duff, litter, down dead wood, understory herbs and shrubs? 3. Is the forest biomass derived from forest inventory data? 4. The combustion efficiency of forests is $0.1 - 0.2$. Is this because much of the forest biomass numbers include boles and branches of live trees which do not burn? 5. Does grassland category include shrub lands? 6. Please comment on how the biomass loadings and consumption estimates used in this study compare with those used in previous global emission inventories (e.g. GFED, van der Werf et al., 2010; FiNN, Wiedinmyer et al., 2011) and surveys of fuel consumption (e.g. van Leeuwen et al., 2014) and studies of grassland biomass in China (e.g. Ni, 2004; Ma et al. 2016; Zhao et al., 2014) 7. The value of 1800 kg/ha (180 $g/m^2$) used in this study compares reasonably well with Ni (2004) study of northern China Northern (325.5 $g/m^2$).

**Response:**

Thanks very much for your comment.

Considering the coarse resolution of statistical data and the lack of fires used for ecosystem management, we have changed the dataset and method employed in this study, and we re-calculated the pixel-based emission of forest and grassland fire. Detailed description could be found in the response to the comment mentioned above..

Besides, we further elaborate on the biomass fuel loadings and combustion factor data. As the values of biomass fuel loadings are various from vegetation types and provinces, in this study, numerous local biomass fuel loadings were collected for various vegetation types and provinces (Fang et al., 1996; Fang et al., 1998; Pu et al., 2004; Hu et al., 2006). Combustion factor of various vegetation types were derived from Michel et al. (2005), Levine et al. (2000), Kasischke et al. (2000), and Hurst et al. (1994).

Detailed description about the methodology employed for the estimation of the biomass burning emission owing to forest and grassland fire was listed in Sect. 2.2.3 of the revised manuscript. Please see the response to the comment mentioned above.

In addition, the specific responses to the comments are listed below:

1. The web link of the references Tian et al. (2003), Lu et al. (2011), and EPD (2014)

has been added in the reference list in the revised manuscript

*"Tian, X. R., Shu, L. F. and Wang, M. Y.: Direct Carbon Emissions from Chinese Forest Fires, 1991−2000, Fire Safety Science, 12, 6−10, 2003, http://hzkx.ustc.edu.cn/ch/reader/view_abstract.aspx?flag=1&file_no=2003120 02&journal_id=hzkx, (in Chinese).*

*Lu, B., Kong, S. F., Han, B., Wang, X. Y. and Bai, Z. P.: Inventory of Atmospheric Pollutants Discharged from Biomass Burning in China Continent in 2007, China Environmental Science, 31, 186−194, 2011, http://manu36.magtech.com.cn/Jweb_zghjkx/CN/Y2011/V31/I2/186, (in Chinese).*

*EPD: Guide for compiling atmospheric pollutant emission inventory for biomass burning, Environmental Protection Department, 2014, http://www.zhb.gov.cn/gkml/hbb/bgg/201501/t20150107_293955.htm, (in Chinese)."*

2. The forest components in the revised manuscript include trunk, branch and leaves of trees.

3. The forest biomass in the revised manuscript is not derived from forest inventory data. The forest inventory data only provides the trunks biomass of the forest. In addition, the data used in our paper also involves branches and leaves biomass of trees.

4. The combustion efficiency of forests in the revised manuscript was set as 0.25 according to Michel et al. (2005).

5. Grassland category in the revised manuscript including woody savannas, savannas and grasslands. shrub lands are included in the forest category.

6. The detailed description on the estimation of biomass loadings and consumption in this study could be found in the response to the comment mentioned above (Sect. 2.2.3). The biomass fuel loadings data are collected from the research made in China (Fang et al., 1996; Fang et al., 1998; Pu et al., 2004; Hu et al., 2006), and the data is often used in recent studies on the estimation of biomass burning emission and proved to be credible. (Song et al., 2008; Qiu et al., 2016).

7. In revised manuscript, the biomass fuel loadings of grassland are different from province and vegetation type.

P7, L1 Units for grassland biomass are kg/ha while other quantities are listed as kg/hm$^2$. While these units are equivalent, please be consistent by using either ha or hm$^2$ throughout the manuscript.

**Response:**

Accepted. We have made the corresponding modification through the full text in the revised manuscript.

Table 4. Note the units for forest biomass.

**Response:**

Accepted. The unit of forest biomass is g/m$^2$, which has been added in Table 4 in the revised manuscript.

P7, L10: Provide a web link to where the references EOCAIY (2013) and NBSC (2013b) can be accessed

**Response:**

Thanks very much for your comment.

The web link of NBSC (2013b) has been added in the reference list in the revised manuscript:

*"National Bureau of Statistics of China (NBSC): China Statistical Yearbook for Regional Economy 2013, China Statistics Press, Beijing, 2013b, available at http://www.stats.gov.cn/tjsj/tjcbw/, (in Chinese)."*

China Animal Industry Yearbook (EOCAIY 2013) is edited by the editorial committee of Chinese animal husbandry, which is a reference about the information of animal husbandry and veterinary medicine, feed and forage industry. It is widely used in recent research, such as Kang et al. (2016) and Huang et al. (2012). However, it is not publicly available online.

Section 2.3 EFs The VOC emissions factors for forest, grasslands, open residue burning,

and feces burning seem quite low compared to those reported in extensive reviews such as Akagi et al. (2011). I imagine the difference is that the VOC category in Table 5 & 6 include only a subset of VOC present in biomass smoke and measured in other studies. Please comment in the differences.

**Response:**

Thanks very much for your comment. The NMVOC emission factors were updated based on a systematic combination of localized measurements conducted in China. According to our examination about the references selected for In-field crop residue burning and feces burning in this study, NMVOC emission factor include alkane, alkene, alkyne and aromatics with C2-C12. In the revised manuscript, the EF for forest and grassland fire was selected from Akagi et al. (2011) due to the lack of localized measurement. The emission of NMVOCs species has been revised according to the species corresponding to emission factor.

Section 2.4 Spatial Distribution Is the land use data of Ran et al. (2010) publicly available? If so, please provide a web link where it may be accessed.

**Response:**

Thanks very much for your comment. The spatial distribution of the land use data is MODIS land cover data which is processed by Ran et al. (2010). The web link of the reference has been added in the reference list in the revised manuscript:

*"Ran, Y. H., Li, X. and Lu, L.: Evaluation of four remote sensing based land cover products over China, Int. J. Remote Sens., 31, 391−401, available at http://www.tandfonline.com/doi/abs/10.1080/01431160902893451, 2010."*

Figure 2 is not very useful. It should be replaced with or augmented with table that provides the total emissions and percent of each species by source.

**Response:**

We thank you very much for your suggestion. As the total emissions of each species have already been mentioned in the manuscript and in the Table 7, we marked the percent of each species by source in the Fig. 2.

[Figure]

*Figure 2. Contribution of different source to the total biomass burning emissions in China, 2012.*

Results and discussion. Please include a table providing total annual fuel consumption and emissions for the 12 species in Table 7 by source. This will is needed to compare current paper to previous studies that may have focused on only a subset of sources.

**Response:**

We thank you very much for your suggestion. We have added the total annual fuel consumption by source in order to discuss the result (Sect. 3.1.1 in lines 15-25 on Page 11 and lines 1-13 on Page 12). The emissions for the 12 species by source can be calculated through the total emission in Table 7 and the percent of each species showed in Fig. 2.

lines 5-8 on Page 12: *"…In addition to the sources mentioned above, the contribution of livestock excrement burning, forest and grassland fire is relatively small. It is mainly due to the small amount of biomass fuel consumption. The biomass fuel consumption of these three biomass sources are 10614 Gg, 6647 Gg and 505 Gg, respectively, which is*

*significantly lower than that of straw domestic combustion (201582 Gg), in-field crop residue burning (147178 Gg) and firewood combustion (127250 Gg)…"*

P9 L15-16 Please note the total annual burned area of forest and grasslands.

**Response:**

We thank you very much for your comment. The total burned area of forest and grasslands are 3587 and 4241 $km^2$, respectively. As the discussion content in P9 L15-16 in the original manuscript is about the contributions by each biomass burning sources. The fuel consumption has more direct influence on emission estimation compared with burned area of forest and grasslands. Therefore, we gave the fuel consumption here:

*"…In addition to the sources mentioned above, the contribution of livestock excrement burning, forest and grassland fire is relatively small. It is mainly due to the small amount of biomass fuel consumption. The biomass fuel consumption of these three biomass sources are 10614 Gg, 6647 Gg and 505 Gg, respectively, which is significantly lower than that of straw domestic combustion (201582 Gg), in-field crop residue burning (147178 Gg) and firewood combustion (127250 Gg). The contribution of livestock excrement burning to PM10, PM2.5, NH3, EC, OC, CO2 and CH4 is 2.52%, 2.47%, 3.44%, 1.52%, 1.96%, 1.67% and 2.10%, respectively. The contribution of forest and grassland fire to biomass burning emissions to most chemical species in China is small (0.9–3.7%), except for the contribution of forest fire to Hg emissions (14.0%)."*

Figures 4 & 5 are difficult to read and the data would be better presented as tables, perhaps in the supplement.

**Response:**

Thanks very much for your comment. Considering the occupied space of many information in Figures 4 & 5, the result is more suitable to present through figures. A furthermore quality improvement of the Figure 4 and Figure 5 has been made. The data in the figures could be read. In addition, the reader can get the detailed data freely through contacting us after the paper acceptation. Considering the importance of the

result, it is better to present it in the main body of the manuscript.

P13, L13-14: Are specific crops typically burned harvest season, sowing season, or both? Or does it vary by region and practice?

**Response:**

Thanks very much for your comment. The specific crops typically burned in harvest season or sowing season, and it varies according to the burning habit in different regions. For example, wheat crop straw in the north often burned in its harvest season while rice crop straw in the south often burned in its sowing season to clear the cultivated land and increasing the soil fertility for the next sowing. In addition, due to the difference of climate conditions, the harvest and sowing season vary in various regions. Therefore, we discussed the temporal variation in biomass burning emission in different regions. We have revised the explanation in Lines 5-8 on Page 17:

*"Burning activity mainly occurs in the harvest season (crop residue burning) or crop sowing season (clearing the cultivated land and increasing the soil fertility for the next sowing), and it varies according to the burning habit in different regions. In addition, the sowing and harvest seasons vary in different regions because of the climate conditions. Because of the differences in burning activity and climate conditions in various regions, monthly emission features vary regionally and to consider this, we divided China into seven areas…"*

Section 3.6 Please describe how parameters were estimated for the PDFs used in the Monte Carlo simulation.

**Response:**

We thank you very much for your comment. We have made the corresponding modification in Lines 2-10 on Page 19:

*"The Monte Carlo method is used to analyse the uncertainty of this emission inventory, which was used in uncertainties estimation for many inventories studies (e.g., Streets et al., 2003; Zhao et al., 2011; Zhao et al., 2012). Activity data (Zheng et al., 2009) and EFs (Zhao et al., 2011) are assumed to be normal distributions. The coefficients of variation (CV, the standard deviation divided by the mean) of activity data and emission factors were obtained from literature review. CV of activity data for*

*firewood and crop straw burning were set as 20% (Zhao et al., 2011; Ni et al., 2015). As the data source of activity data for livestock excrement is same as the crop straw burning (i.e., government statistic data), CV is also set as 20%. MCD64A1 burned data products has been shown to be reliable in big fires (Giglio et al., 2013), and the CV of burned area of forest and grassland fire is from the reported standard deviation (Giglio et al., 2010). The biomass fuel loadings (Saatchi et al., 2011; Shi et al., 2015) and combustion factor (van der Werf et al., 2010) of forest and grassland fire were within a CV of approximately 50%. The CV of EF for each pollutant for each biomass burning type is shown in the supplement S8 and S9."*

Supplement S8 and S9 have been added in the revised supplement:

**S8 CV (coefficients of variation) of biomass domestic burning emission factors.**

| | Material | $SO_2$ | $NO_x$ | $PM_{10}$ | $PM_{2.5}$ | NMVOC | $NH_3$ | CO | EC | OC | $CO_2$ | $CH_4$ | Hg |
|---|---|---|---|---|---|---|---|---|---|---|---|---|---|
| Domestic burning | Corn | 0.5[*] | 0.02[a] | 0.5[*] | 0.27[b] | 0.5[*] | 0.5[*] | 0.85[a] | 0.34[b] | 0.44[b] | 0.04[a] | 0.5[*] | 0.05[c] |
| | Wheat | 0.5[*] | 0.16[a] | 0.5[*] | 0.23[b] | 0.5[*] | 0.5[*] | 0.89[a] | 0.76[b] | 0.29[b] | 0.07[a] | 0.5[*] | 0.12[c] |
| | Cotton | 0.5[*] | 0.5[*] | 0.5[*] | 0.26[b] | 0.5[*] | 0.5[*] | 0.5[*] | 0.39[b] | 0.55[b] | 0.5[*] | 0.5[*] | 0.33[c] |
| | Cane | 0.5[*] | 0.5[*] | 0.5[*] | 0.26[b] | 0.5[*] | 0.5[*] | 0.5[*] | 0.63[b] | 0.45[b] | 0.5[*] | 0.5[*] | 0.32[c] |
| | Potato | 0.5[*] | 0.5[*] | 0.5[*] | 0.26[b] | 0.5[*] | 0.5[*] | 0.5[*] | 0.63[b] | 0.45[b] | 0.5[*] | 0.5[*] | 0.53[c] |
| | Peanut | 0.5[*] | 0.5[*] | 0.5[*] | 0.26[b] | 0.5[*] | 0.5[*] | 0.5[*] | 0.63[b] | 0.45[b] | 0.5[*] | 0.5[*] | 0.03[c] |
| | Rape | 0.5[*] | 1.21[d] | 0.5[*] | 0.15[b] | 0.26[d] | 0.5[*] | 0.26[d] | 0.63[b] | 0.45[b] | 0.5[*] | 0.5[*] | 0.3[c] |
| | Sesame | 0.5[*] | 1.78[d] | 0.5[*] | 0.26[b] | 0.24[d] | 0.5[*] | 0.29[d] | 0.63[b] | 0.45[b] | 0.5[*] | 0.5[*] | 0.3[c] |
| | Beet | 0.5[*] | 0.5[*] | 0.5[*] | 0.26[b] | 0.5[*] | 0.5[*] | 0.5[*] | 0.63[b] | 0.45[b] | 0.5[*] | 0.5[*] | 0.3[c] |
| | Hemp | 0.5[*] | 0.5[*] | 0.5[*] | 0.26[b] | 0.5[*] | 0.5[*] | 0.5[*] | 0.63[b] | 0.45[b] | 0.5[*] | 0.5[*] | 0.3[c] |
| | Rice | 0.5[*] | 0.05[a] | 0.5[*] | 0.29[b] | 0.5[*] | 0.5[*] | 0.06[a] | 0.65[b] | 0.5[b] | 0.01[a] | 0.5[*] | 0.46[c] |
| | Soybean | 0.5[*] | 1.78[d] | 0.5[*] | 0.26[b] | 0.76[d] | 0.5[*] | 0.44[d] | 0.63[b] | 0.45[b] | 0.5[*] | 0.5[*] | 0.74[c] |
| | Firewood | 0.5[*] | 1.42[d] | 0.5[*] | 0.16[b] | 0.15[d] | 0.5[*] | 0.39[d] | 0.46[b] | 0.35[b] | 0.5[*] | 0.5[*] | 1.17[c] |
| | Feces | 0.8[*] | 0.8[*] | 0.8[*] | 0.8[*] | 0.8[*] | 0.8[*] | 0.8[*] | 0.8[*] | 0.8[*] | 0.8[*] | 0.8[*] | 0.8[*] |

Table S1 CV (coefficients of variation) of biomass domestic burning emission factors.

Note: Lowercase letters indicate the data source.

Sources are from the following: [a] Zhang et al. (2008). [b] Li et al. (2009). [c] Chen et al. (2013). [d] Zhang et al. (2013). [*] Expert judgment data from Wei et al. (2011).

**S9 CV (coefficients of variation) of biomass open burning emission factors.**

| | Material | $SO_2$ | $NO_x$ | $PM_{10}$ | $PM_{2.5}$ | NMVOC | $NH_3$ | CO | EC | OC | $CO_2$ | $CH_4$ | Hg |
|---|---|---|---|---|---|---|---|---|---|---|---|---|---|
| Open burning | Corn | 0.45[b] | 0.42[b] | 0.5[*] | 0.09[b] | 0.53[b] | 0.76[b] | 0.08[b] | 0.33[b] | 0.39[b] | 0.01[b] | 0.22[b] | 0.05[a] |
| | Wheat | 0.67[b] | 0.52[b] | 0.5[*] | 0.54[b] | 0.25[b] | 0.38[b] | 0.41[b] | 0.32[b] | 0.26[b] | 0.03[b] | 0.25[b] | 0.12[a] |
| | Cotton | 0.5[*] | 0.5[*] | 0.5[*] | 0.5[*] | 0.5[*] | 0.5[*] | 0.5[*] | 0.5[*] | 0.5[*] | 0.5[*] | 0.5[*] | 0.33[a] |
| | Cane | 0.5[*] | 0.32[d] | 0.19[d] | 0.16[d] | 0.71[d] | 0.5[*] | 0.61[d] | 1.57[d] | 0.2[d] | 0.18[d] | 0.5[*] | 0.32[a] |
| | Potato | 0.5[*] | 0.5[*] | 0.5[*] | 0.5[*] | 0.5[*] | 0.5[*] | 0.5[*] | 0.5[*] | 0.5[*] | 0.5[*] | 0.5[*] | 0.53[a] |
| | Peanut | 0.5[*] | 0.5[*] | 0.5[*] | 0.5[*] | 0.5[*] | 0.5[*] | 0.5[*] | 0.5[*] | 0.5[*] | 0.5[*] | 0.5[*] | 0.03[a] |
| | Rape | 0.5[*] | 0.5[*] | 0.5[*] | 0.5[*] | 0.5[*] | 0.5[*] | 0.5[*] | 0.5[*] | 0.5[*] | 0.5[*] | 0.5[*] | 0.3[a] |
| | Sesame | 0.5[*] | 0.5[*] | 0.5[*] | 0.5[*] | 0.5[*] | 0.5[*] | 0.5[*] | 0.5[*] | 0.5[*] | 0.5[*] | 0.5[*] | 0.3[a] |
| | Beet | 0.5[*] | 0.5[*] | 0.5[*] | 0.5[*] | 0.5[*] | 0.5[*] | 0.5[*] | 0.5[*] | 0.5[*] | 0.5[*] | 0.5[*] | 0.3[a] |
| | Hemp | 0.5[*] | 0.5[*] | 0.5[*] | 0.5[*] | 0.5[*] | 0.5[*] | 0.5[*] | 0.5[*] | 0.5[*] | 0.5[*] | 0.5[*] | 0.3[a] |
| | Rice | 0.5[*] | 0.8[d] | 0.88[d] | 0.17[d] | 0.75[d] | 0.5[*] | 1.19[d] | 1.38[d] | 1.53[d] | 0.14[d] | 0.5[*] | 0.46[a] |
| | Soybean | 0.5[*] | 0.5[*] | 0.5[*] | 0.5[*] | 0.5[*] | 0.5[*] | 0.5[*] | 0.5[*] | 0.5[*] | 0.5[*] | 0.5[*] | 0.74[a] |
| | Evergreen Needleleaf Forest | 0.3[c] | 0.39[c] | 0.25[d] | 0.25[d] | 0.31[e] | 0.66[e] | 0.38[e] | 1[f] | 0.62[f] | 0.08[e] | 0.52[e] | 0.52[g] |
| | Evergreen Broadleaf Forest | 0.4[e] | 0.54[e] | 0.25[d] | 0.25[d] | - | 1.58[h] | 0.29[e] | 0.6[e] | 0.57[e] | 0.04[e] | 0.39[e] | 0.52[g] |
| | Deciduous Needleleaf Forest | 0.3[c] | 0.23[c] | 0.25[d] | 0.25[d] | 0.31[e] | 0.66[e] | 0.38[r] | 1[f] | 0.62[f] | 0.08[e] | 0.52[r] | 0.52[g] |
| | Deciduous Broadleaf Forest | 0.3[c] | 0.46[e] | 0.25[d] | 0.25[d] | 0.79[e] | 0.27[e] | 0.19[e] | 0.33[e] | 0.52[e] | 0.02[e] | 0.18[e] | 0.52[g] |
| | Mixed Forest | 0.3[c] | 0.46[e] | 0.25[d] | 0.25[d] | 0.62[e] | 0.27[e] | 0.19[e] | 0.33[e] | 0.52[e] | 0.02[e] | 0.18[e] | 0.52[g] |
| | Closed Shrublands | 0.44[e] | 0.21[e] | 0.25[d] | 0.25[d] | 0.48[e] | 0.33[e] | 0.25[e] | 0.4[f] | 0.18[f] | 0.02[e] | 0.35[e] | 0.74[h,g] |
| | Open Shrublands | 0.44[e] | 0.21[e] | 0.25[d] | 0.25[d] | 0.48[e] | 0.33[e] | 0.25[e] | 0.4[f] | 0.18[f] | 0.02[e] | 0.35[e] | 0.74[h,g] |
| | Woody Savannas | 0.44[e] | 0.21[e] | 0.25[d] | 0.25[d] | 0.48[e] | 0.33[e] | 0.25[e] | 0.4[f] | 0.18[f] | 0.02[e] | 0.35[e] | 0.52[h] |
| | Savannas | 0.63[e] | 0.29[e] | 0.25[d] | 0.25[d] | 0.25[e] | 0.8[e] | 0.29[e] | 0.5[e] | 0.46[e] | 0.02[e] | 0.6[e] | 0.52[h] |
| | Grasslands | 0.63[e] | 0.29[e] | 0.25[d] | 0.25[d] | 0.25[e] | 0.8[e] | 0.29[e] | 0.5[e] | 0.46[e] | 0.02[e] | 0.6[e] | 0.52[h] |

Table S2 CV (coefficients of variation) of biomass open burning emission factors.

Note: Lowercase letters indicate the data source.

*Sources are from the following: [a] Chen et al. (2013). [b] Li et al. (2007). [c] Andreae and Rosenfeld (2008). [d] Song et al. (2009). [e] Akagi et al. (2011).[f] McMeekin et al. (2008). [g] Friedli et al. (2003). [h] Streets et al. (2005). * Expert judgment data from Wei et al. (2011).*

Figure S4. Please note the data sources used to derive the non-carbon PM components.

**Response:**

We thank you very much for your suggestion. The $PM_{2.5}$ speciation is obtained from Li et al., (2007) and Waston et al., (2001), which have been described in Sect. 2.6 in the original manuscript. In addition, we have added the data sources of the $PM_{2.5}$ speciation in Figure S4.

[Figure]

*Figure S4 Emission of $PM_{2.5}$ species from biomass burning.*

*Note:Species in others include Al,Si,Mg,Fe,Pb,Zn,Ba,Ti,Ni,Cr,Mn,Sr,V,Cd,As,Zr,Se,Ag,Sb,Sc,Mo,Ga,Tl,Co and Hg. $PM_{2.5}$ speciation profile is obtained from Li et al., (2007) and Waston et al., (2001).*

TECHINCAL The manuscript contains many minor grammatical errors, here are a few: P13, Ln3-4: Change "This is because the main contribution of these species emission sources is from straw outdoor burning" to "This is because straw outdoor burning is the main source for these species" P13, Ln 4: change "The outdoor burning straw mainly occurs in..." to "The outdoor burning of straw occurs mainly in..." P13, L19 insert "a" between "have" and "relatively" and change "peak" to "peaks" P13, L20: change "discrepancies" to "differences" P13, L28: change "while" to "where" P14, L7: change 'peak" to "peaks" P14, L14: Change "Besides" to "Additionally" P16, L23-25 Sentence beginning "More localized EF of..." is jumbled and must be rewritten.
REFERENCES Akagi et al. (2011) Atmos. Chem. Phys. 11, 4039-4072.

Ma, A. et al. Carbon storage in Chinese grassland ecosystems: Influence of different integrative methods. Sci. Rep. 6, 21378; doi: 10.1038/srep21378 (2016).

Ni (2004) Plant Ecology, 174, 217-234.

van der Werf et al. (2010) Atmos. Chem. Phys., 10, 1170711735

van Leeuwen et al. (2014) Biogeosciences, 11, 7305-7329.

Wiedinmyer et al. (2011)Geosci. ModelDev.,4,625-641

Zhao et al. (2014)RemoteSens. 6,5368-5386.

**Response:**

We thank you very much for your helpful suggestion. We have made the corresponding modification in the revised manuscript.

Reference:

[revised manuscript text omitted]

[a] Fang et al. (2015). [b] Zhao et al. (2015). [c] Tian et al. (2011). [d] Bao et al. (2014). [e] Chang et al. (2012). [f] Liu et al. (2010). [g] Wang and Zhao (2011). [h] Qin and Ge (2012). [i] Huang (2012a). [j] Liu et al. (2014). [k] Li et al. (2013a). [l] Li et al. (2013b). [m] Zhang et al. (2015). [n] Hou et al. (2013). [o] EPD (2014).

* The result from our questionnaire.

**Table 3. Residue-to-production ratio ($N_k$), dry matter fraction ($D_k$) and combustion efficiency ($CE_k$) of crop straw used in this study.**

[revised manuscript text omitted]

**(a) In-field crop residue burning**

**(b) Crop straw domestic burning**

**(c) Total crop straw burning**

Corn · Rice · Wheat · Soybean · Cotton · Sugar cane · Potato · Peanut · Rape · Sesame · Sugar beet · Hemp

**Figure 3. Contributions of 12 crop straw typess burning to total pollutant straws burning emissions for various species.**

[Figure]

**Figure 4. Contributions of different biomass sources to the emission in each province (Gg).**

Note: The numbers 1–12 represent the species SO$_2$×10, NO$_x$, NMVOC, NH$_3$×10, EC×10, OC, CO/10, PM$_{10}$, PM$_{2.5}$, CO$_2$/100, CH$_4$ and Hg×1000000, respectively.

[Figure]

**Figure 5. Contributions of different crop straw types to the emission in each province (Gg).**

Note: The numbers 1–12 represent the species SO$_2$×10, NO$_x$, NMVOC, NH$_3$×10, EC×10, OC, CO/10, PM$_{10}$, PM$_{2.5}$, CO$_2$/100, CH$_4$ and Hg×1000000, respectively.

[Figure]

(a) The annual emissions of PM$_{2.5}$ in each county

(b) PM$_{2.5}$ emissions intensities per unit area

(c) PM$_{2.5}$ emissions intensities per capita

(d) The number of annual PM$_{2.5}$ emissions within different range of county resolution

**Figure 6. Biomass emission inventory at county resolution and intensity (PM$_{2.5}$).**

[Figure]

**Figure 7. Gridded distribution of PM₂.₅ annual emissions.**

[Figure]

5  **Figure 8. Monthly variation of different biomass sources emission for each chemical species.**

[Figure]

**Figure 9. Monthly variation of different biomass sources emission for PM$_{2.5}$ emissions in different regions.**

[Figure]

Figure 10. Comparison of the emissions inventory derived by this study with the emissions estimated by previous research.

*Supplement of*

**A comprehensive biomass burning emission inventory with high spatial and temporal resolution in China**

Ying Zhou [1,2], Xiaofan Xing [1,2], Jianlei Lang [1,2], Dongsheng Chen [1,2], Shuiyuan Cheng [1,2,3], Lin Wei[1,2], Xiao Wei [4], Chao Liu [5]

[1] Key Laboratory of Beijing on Regional Air Pollution Control, Beijing University of Technology, Beijing 100124, China

[2] College of Environmental & Energy Engineering, Beijing University of Technology, Beijing 100124, China

[3] Collaborative Innovation Center of Electric Vehicles, Beijing 100081, China

[4] Beijing Municipal Research Institute of Environmental Protection, Beijing 100037, China

[5] Environmental Meteorological Center of China Meteorological Administration, Beijing 100081, China

*Correspondence to:* Ying Zhou (y.zhou@bjut.edu.cn) and Shuiyuan Cheng (bjutpaper@gmail.com)

**Supporting Information**

**Section S1:** Figure S1 The correlation between crop yield and grain yield at prefecture

5      resolution.

**Section S2:** Figure S2 Map showing the prefecture and county resolution.

**Section S3:** The details about questionnaire field survey.

**Section S4:** The detailed description about the MODIS fire data and calculation method and

equation of gridded emission

10     **Section S5:** Figure S3 Daily $PM_{2.5}$ biomass burning emissions variation in 2012.

**Section S6:** Figure S4 Emission of $PM_{2.5}$ species from biomass burning.

**Section S7:** Figure S5 Emission of NMVOCs species from biomass burning.

**Section S8:** Table S1 CV (coefficients of variation) of biomass domestic burning emission
factors.

15     **Section S9:** Table S2 CV (coefficients of variation) of biomass open burning emission factors

**S1 The correlation between crop yield and grain yield at prefecture resolution.**

[Figure]

**Figure S1 The correlation between crop yield and grain yield at prefecture resolution.[a]**

Note: [a] NBSC (2013b); [b] a range of statistical yearbooks edited by National Bureau of Statistics in 2012 for each province.

**S2 Map showing the prefecture and county resolution.**

[Figure]

1. Beijing
2. Tianjin
3. Hebei
4. Shanxi
5. Inner-Mongolia
6. Liaoning
7. Jilin
8. Heilongjiang
9. Shanghai
10. Jiangsu
11. Zhejiang
12. Anhui
13. Fujian
14. Jiangxi
15. Shandong
16. Henan
17. Hubei
18. Hunan
19. Guangdong
20. Guangxi
21. Hainan
22. Chongqing
23. Sichuan
24. Guizhou
25. Yunnan
26. Tibet
27. Shaanxi
28. Gansu
29. Qinghai
30. Ningxia
31. Xinjiang

(a) Province level

(b) Prefecture level

(c) County level

**Figure S2 Map showing the prefecture and county resolution.**

**S3 The details about questionnaire field survey.**

A questionnaire was designed to conduct field investigation during face-to-face interviews with rural resident, in order to obtain the percentage of crop straw indoor burning and outdoor burning and uneven temporal distribution coefficient in several provinces with limited literature reports, including Tianjin, Hebei, Inner Mongolia, Heilongjiang, Shanghai, Zhejiang, Anhui, Jiangxi and Guangdong provinces. Respondents need to provide the detailed address, main cultivated crop type. They selected from a list of cooking and heating fuels, including specific crop straw, firewood, coal, gas, electricity or solar, livestock excrement and other detailed fuels not existing in the list. They also need to provide approximate proportion of crop straw domestic combustion and in field burning, and selected the month of burning the straw as waste, and heating period. The investigation was launch in the representative regions in each province mentioned above, with the integrative consideration about the geographical location, economic development level and population intensity. All the surveyors were trained and tested in their understanding of the questionnaire content. Ultimately, we received 2478 valid questionnaire responses, and at least 200 valid questionnaires in each province.

**S4 The detailed description about the MODIS fire data and calculation method and equation of gridded emission**

**4.1 Detailed description about the MODIS fire data**

For the spatiotemporal distributions of biomass open burning, satellite remote sensing has excellent characteristics of wide coverage, high resolution and strong temporal reliability. As a result, satellite remote sensing has been increasingly applied to solving temporal and spatial emission distributions in recent years. The MODIS satellite fire data were taken from FIRM (Fire Information for Resource Management System). The MODIS Thermal Anomalies/Fire 5-Min L2 Swath Product (MOD14/MYD14) within 1km resolution was used in this study. The MOD14 were provided by the Terra satellite with overpass times at 10:30 AM and 10:30 PM local time, while MYD14 were provided by Aqua at 1:30 AM and 1:30 PM local time.

**4.2 Detailed calculation method and equation of gridded emission**

The mass of biomass emission in each grid of biomass open burning and indoor burning was calculated using Eqs. (1) and (2), respectively, as follows:

$$E_{m-outdoor} = \frac{FC_m}{FC_n} \times E_{n-outdoor} \tag{1}$$

$$E_{m-indoor} = \frac{PO_m}{PO_n} \times E_{n-indoor} \tag{2}$$

where $m$ is the $m$-th grid and $n$ represents the $n$-th county; $E_{m-outdoor}$ and $E_{n-outdoor}$ represent the emissions of the $m$-th grid and $n$-th county for biomass outdoor burning (in-field crop residue burning), respectively; $E_{m-indoor}$ and $E_{n-indoor}$ represent the emissions of the $m$-th grid and $n$-th county for biomass indoor burning, respectively; $FC_m$ represents the number of typical fire points of the $m$-th grid; $FC_n$ is the number of total typical fire points of the $n$-th county; $PO_m$ is the number of typical population of the $m$-th grid; finally, $PO_n$ is the number of typical population of the $n$-th county.

**S5 Daily PM₂.₅ biomass burning emissions variation in 2012.**

[Figure]

Figure S3 Daily PM₂.₅ biomass burning emissions variation in 2012.

**S6 Emission of PM2.5 species from biomass burning**

[Figure]

**Figure S4 Emission of PM$_{2.5}$ species from biomass burning.**

Note: Species in others include Al,Si,Mg,Fe,Pb,Zn,Ba,Ti,Ni,Cr,Mn,Sr,V,Cd,As,Zr,Se,Ag,Sb,Sc,Mo,Ga,Tl,Co and Hg. PM$_{2.5}$ speciation profile is obtained from Li et al., (2007) and Waston et al., (2001).

**S7 Emission of NMVOC species from biomass burning.**

[Figure]

**Figure S5 Emission of NMVOC species from biomass burning.**

Note: *Species in others include aldehyde, ethers, alcohols, esters, ketone and acids.

**S8 CV (coefficients of variation) of biomass domestic burning emission factors.**

| | Material | $SO_2$ | $NO_x$ | $PM_{10}$ | $PM_{2.5}$ | NMVOC | $NH_3$ | CO | EC | OC | $CO_2$ | $CH_4$ | Hg |
|---|---|---|---|---|---|---|---|---|---|---|---|---|---|
| | Corn | 0.5[*] | 0.02[a] | 0.5[*] | 0.27[b] | 0.5[*] | 0.5[*] | 0.85[a] | 0.34[b] | 0.44[b] | 0.04[a] | 0.5[*] | 0.05[c] |
| | Wheat | 0.5[*] | 0.16[a] | 0.5[*] | 0.23[b] | 0.5[*] | 0.5[*] | 0.89[a] | 0.76[b] | 0.29[b] | 0.07[a] | 0.5[*] | 0.12[c] |
| | Cotton | 0.5[*] | 0.5[*] | 0.5[*] | 0.26[b] | 0.5[*] | 0.5[*] | 0.5[*] | 0.39[b] | 0.55[b] | 0.5[*] | 0.5[*] | 0.33[c] |
| | Cane | 0.5[*] | 0.5[*] | 0.5[*] | 0.26[b] | 0.5[*] | 0.5[*] | 0.5[*] | 0.63[b] | 0.45[b] | 0.5[*] | 0.5[*] | 0.32[c] |
| | Potato | 0.5[*] | 0.5[*] | 0.5[*] | 0.26[b] | 0.5[*] | 0.5[*] | 0.5[*] | 0.63[b] | 0.45[b] | 0.5[*] | 0.5[*] | 0.53[c] |
| Domestic burning | Peanut | 0.5[*] | 0.5[*] | 0.5[*] | 0.26[b] | 0.5[*] | 0.5[*] | 0.5[*] | 0.63[b] | 0.45[b] | 0.5[*] | 0.5[*] | 0.03[c] |
| | Rape | 0.5[*] | 1.21[d] | 0.5[*] | 0.15[b] | 0.26[d] | 0.5[*] | 0.26[d] | 0.63[b] | 0.45[b] | 0.5[*] | 0.5[*] | 0.3[c] |
| | Sesame | 0.5[*] | 1.78[d] | 0.5[*] | 0.26[b] | 0.24[d] | 0.5[*] | 0.29[d] | 0.63[b] | 0.45[b] | 0.5[*] | 0.5[*] | 0.3[c] |
| | Beet | 0.5[*] | 0.5[*] | 0.5[*] | 0.26[b] | 0.5[*] | 0.5[*] | 0.5[*] | 0.63[b] | 0.45[b] | 0.5[*] | 0.5[*] | 0.3[c] |
| | Hemp | 0.5[*] | 0.5[*] | 0.5[*] | 0.26[b] | 0.5[*] | 0.5[*] | 0.5[*] | 0.63[b] | 0.45[b] | 0.5[*] | 0.5[*] | 0.3[c] |
| | Rice | 0.5[*] | 0.05[a] | 0.5[*] | 0.29[b] | 0.5[*] | 0.5[*] | 0.06[a] | 0.65[b] | 0.5[b] | 0.01[a] | 0.5[*] | 0.46[c] |
| | Soybean | 0.5[*] | 1.78[d] | 0.5[*] | 0.26[b] | 0.76[d] | 0.5[*] | 0.44[d] | 0.63[b] | 0.45[b] | 0.5[*] | 0.5[*] | 0.74[c] |
| | Firewood | 0.5[*] | 1.42[d] | 0.5[*] | 0.16[b] | 0.15[d] | 0.5[*] | 0.39[d] | 0.46[b] | 0.35[b] | 0.5[*] | 0.5[*] | 1.17[c] |
| | Feces | 0.8[*] | 0.8[*] | 0.8[*] | 0.8[*] | 0.8[*] | 0.8[*] | 0.8[*] | 0.8[*] | 0.8[*] | 0.8[*] | 0.8[*] | 0.8[*] |

**Table S1 CV (coefficients of variation) of biomass domestic burning emission factors.**

Note: Lowercase letters indicate the data source.

Sources are from the following: [a] Zhang et al. (2008). [b] Li et al. (2009). [c] Chen et al. (2013). [d] Zhang et al. (2013). [*] Expert judgment data from Wei et al. (2011).

**S9 CV (coefficients of variation) of biomass open burning emission factors**

| Material | $SO_2$ | $NO_x$ | $PM_{10}$ | $PM_{2.5}$ | NMVOC | $NH_3$ | CO | EC | OC | $CO_2$ | $CH_4$ | Hg |
|---|---|---|---|---|---|---|---|---|---|---|---|---|
| Corn | 0.45[b] | 0.42[b] | 0.5[*] | 0.09[b] | 0.53[b] | 0.76[b] | 0.08[b] | 0.33[b] | 0.39[b] | 0.01[b] | 0.22[b] | 0.05[a] |
| Wheat | 0.67[b] | 0.52[b] | 0.5[*] | 0.54[b] | 0.25[b] | 0.38[b] | 0.41[b] | 0.32[b] | 0.26[b] | 0.03[b] | 0.25[b] | 0.12[a] |
| Cotton | 0.5[*] | 0.5[*] | 0.5[*] | 0.5[*] | 0.5[*] | 0.5[*] | 0.5[*] | 0.5[*] | 0.5[*] | 0.5[*] | 0.5[*] | 0.33[a] |
| Cane | 0.5[*] | 0.32[d] | 0.19[d] | 0.16[d] | 0.71[d] | 0.5[*] | 0.61[d] | 1.57[d] | 0.2[d] | 0.18[d] | 0.5[*] | 0.32[a] |
| Potato | 0.5[*] | 0.5[*] | 0.5[*] | 0.5[*] | 0.5[*] | 0.5[*] | 0.5[*] | 0.5[*] | 0.5[*] | 0.5[*] | 0.5[*] | 0.53[a] |
| Peanut | 0.5[*] | 0.5[*] | 0.5[*] | 0.5[*] | 0.5[*] | 0.5[*] | 0.5[*] | 0.5[*] | 0.5[*] | 0.5[*] | 0.5[*] | 0.03[a] |
| Rape | 0.5[*] | 0.5[*] | 0.5[*] | 0.5[*] | 0.5[*] | 0.5[*] | 0.5[*] | 0.5[*] | 0.5[*] | 0.5[*] | 0.5[*] | 0.3[a] |
| Sesame | 0.5[*] | 0.5[*] | 0.5[*] | 0.5[*] | 0.5[*] | 0.5[*] | 0.5[*] | 0.5[*] | 0.5[*] | 0.5[*] | 0.5[*] | 0.3[a] |
| Beet | 0.5[*] | 0.5[*] | 0.5[*] | 0.5[*] | 0.5[*] | 0.5[*] | 0.5[*] | 0.5[*] | 0.5[*] | 0.5[*] | 0.5[*] | 0.3[a] |
| Hemp | 0.5[*] | 0.5[*] | 0.5[*] | 0.5[*] | 0.5[*] | 0.5[*] | 0.5[*] | 0.5[*] | 0.5[*] | 0.5[*] | 0.5[*] | 0.3[a] |
| Rice | 0.5[*] | 0.8[d] | 0.88[d] | 0.17[d] | 0.75[d] | 0.5[*] | 1.19[d] | 1.38[d] | 1.53[d] | 0.14[d] | 0.5[*] | 0.46[a] |
| Soybean | 0.5[*] | 0.5[*] | 0.5[*] | 0.5[*] | 0.5[*] | 0.5[*] | 0.5[*] | 0.5[*] | 0.5[*] | 0.5[*] | 0.5[*] | 0.74[a] |
| Evergreen Needleleaf Forest | 0.3[c] | 0.39[c] | 0.25[d] | 0.25[d] | 0.31[e] | 0.66[e] | 0.38[e] | 1[f] | 0.62[f] | 0.08[e] | 0.52[e] | 0.52[g] |
| Evergreen Broadleaf Forest | 0.4[e] | 0.54[e] | 0.25[d] | 0.25[d] | - | 1.58[h] | 0.29[e] | 0.6[e] | 0.57[e] | 0.04[e] | 0.39[e] | 0.52[g] |
| Deciduous Needleleaf Forest | 0.3[c] | 0.23[c] | 0.25[d] | 0.25[d] | 0.31[e] | 0.66[e] | 0.38[r] | 1[f] | 0.62[f] | 0.08[e] | 0.52[r] | 0.52[g] |
| Deciduous Broadleaf Forest | 0.3[c] | 0.46[e] | 0.25[d] | 0.25[d] | 0.79[e] | 0.27[e] | 0.19[e] | 0.33[e] | 0.52[e] | 0.02[e] | 0.18[e] | 0.52[g] |
| Mixed Forest | 0.3[c] | 0.46[e] | 0.25[d] | 0.25[d] | 0.62[e] | 0.27[e] | 0.19[e] | 0.33[e] | 0.52[e] | 0.02[e] | 0.18[e] | 0.52[g] |
| Closed Shrublands | 0.44[e] | 0.21[e] | 0.25[d] | 0.25[d] | 0.48[e] | 0.33[e] | 0.25[e] | 0.4[f] | 0.18[f] | 0.02[e] | 0.35[e] | 0.74[h,g] |
| Open Shrublands | 0.44[e] | 0.21[e] | 0.25[d] | 0.25[d] | 0.48[e] | 0.33[e] | 0.25[e] | 0.4[f] | 0.18[f] | 0.02[e] | 0.35[e] | 0.74[h,g] |
| Woody Savannas | 0.44[e] | 0.21[e] | 0.25[d] | 0.25[d] | 0.48[e] | 0.33[e] | 0.25[e] | 0.4[f] | 0.18[f] | 0.02[e] | 0.35[e] | 0.52[h] |
| Savannas | 0.63[e] | 0.29[e] | 0.25[d] | 0.25[d] | 0.25[e] | 0.8[e] | 0.29[e] | 0.5[e] | 0.46[e] | 0.02[e] | 0.6[e] | 0.52[h] |
| Grasslands | 0.63[e] | 0.29[e] | 0.25[d] | 0.25[d] | 0.25[e] | 0.8[e] | 0.29[e] | 0.5[e] | 0.46[e] | 0.02[e] | 0.6[e] | 0.52[h] |

(Left vertical label: Open burning)

**Table S2 CV (coefficients of variation) of biomass open burning emission factors**

Note: Lowercase letters indicate the data source.

Sources are from the following: [a] Chen et al. (2013). [b] Li et al. (2007). [c] Andreae and Rosenfeld (2008). [d] Song et al. (2009). [e] Akagi et al. (2011). [f] McMeekin et al. (2008). [g] Friedli et al. (2003). [h] Streets et al. (2005). * Expert judgment data from Wei et al. (2011).

---

## Author Response (AR2)

**Authors' Responses to Reviewer Comments**

Manuscript: A comprehensive biomass burning emission inventory with high spatial and temporal resolution in China (Ref. No.: acp-2016-560)

We are very grateful to the respectful referee for the careful and insightful review. The comments and suggestions have contributed greatly to improving our paper. Our point-by-point responses to the comments are listed as follows.

**Authors Reply to Referee 1**

Page 8 Line 23-26: As for CF, it has usually been set as a constant in previous literature. In our paper, CF values were collected for each vegetation type, and the CF in each pixel was determined by the MODIS Land Cover product and the CF of typical vegetation. The CF of the forest, closed shrublands, open shrublands, woody savannas, and grassland were set as 0.25, 0.5, 0.85, 0.4, and 0.95, respectively (Michel et al., 2005; Kasischke et al., 2000; Hurst et al., 1994).

The constant CF cannot reflect the spatial variation on the fraction of burned aboveground biomass when a fire happens. CF was controlled by both land cover type and moisture condition. Land cover type decides the maximum and minimum of the combustion efficiency. But the moisture condition determines how much the fuel can be burned during fires. If the fuel is too dry, almost 90% can be combusted, but when it is too wet, only 5% can be burned (van der Werf et al., 2006; 2010). This condition (ranging from 5 to 90%) significantly affects the final emission estimation. Many existing studies dealt with the problem by using vegetation fraction, leaf area index or NDVI to reflect the real condition of CF pixel by pixel since they vary greatly (Zhang et al., 2008, Atmos Environ).

**Response:**

We thank you very much for your comments. In our paper, a comprehensive biomass

burning emission inventory, including domestic straw burning, in-field straw burning, firewood and livestock excrement combustion, and forest and grassland fire for mainland China was developed. As for the part mentioned by the reviewer, the CF was used for the estimation of the forest and grassland fires emission. According to our result, the contribution to total emission for most pollutants of the forest and grassland fire ranged from 0.9% to 3.7%. In other studies, the forest and grassland fire is also not the main biomass burning emission source in China (Lu et al., 2011; Huang et al., 2012). Therefore, considering the information we could obtain currently, we used specific CFs according to the vegetation type in each pixel based on literature review (Michel et al., 2005; Kasischke et al., 2000; Hurst et al., 1994). He et al. (2011), Chen et al. (2013), and Zhang et al. (2013) used the similar approach to consider CFs. Moreover, according to the current studies, vegetation type could reflect the moisture condition to some extent (Chang, 1986; Niu, 2000; Ren and Lin, 2013). According to the consideration mentioned above, the CF value selection would not have important impact on the emission inventory result in this study.

Human waste open burning accounted for a large proportion in developing countries due to without efficient burning facility, especially in rural area. China is a good example. The study from Wiedinmyer et al. (2014) estimated the global emissions of trace gases, particulate matter, and hazardous air pollutants from open burning of domestic waste, China is the largest contributor worldwide. Please refer to the large emissions from this part.

**Response:**

We thank you very much for your suggestion. The human waste burning indeed accounted for a large proportion in developing countries. However, the aim of this study is a comprehensive emission inventory of biomass burning. According to our previous response, the open burning of the biomass waste has been included in our paper, and the human waste irrelevant to the biomass is not the target in this study. Moreover, Few EFs of human waste burning are categorized into specific waste types in current studies (the detailed literature review could be found in the previous response). The specific

human waste production in rural areas is difficult to obtain currently in China. If we attempt to estimate the emissions of human waste burning based on non-specific EFs and waste production in our biomass burning study, the results of the biomass burning emission inventory will be overestimated due to the introduced emissions that are irrelevant to biomass burning. With more studies on the specific characteristics and EFs of human waste burning, further study could be conduct about the detailed and reliable emission inventory of human waste burning.

The Monte Carlo model was used to allocate the uncertainty. How did the author run this model? There is a strange phenomenon in this process since the input data for emission estimate are very with Street et al (2003) and van der Werf et al. (2006; 2010), Huang et al. (2011). And the uncertainties you defined in this study is the same with others, therefore, how the uncertainty ranges are so narrow than the others? That means why your estimation has more confidence than others. This is unbelievable.

**Response:**

Thanks very much for your comments. The details about the Monte Carlo model running has been described in Sect. 3.6 (Lines 11-19 on Page 18) of the revised manuscript:

*"Activity data (Zheng et al., 2009) and EFs (Zhao et al., 2011) are assumed to be normal distributions. The coefficients of variation (CV, the standard deviation divided by the mean) of activity data and EFs were obtained from literature review. CV of activity data for firewood and straw burning were set as 20% (Zhao et al., 2011; Ni et al., 2015). As the data source of activity data for livestock excrement is same as the crop straw burning (i.e., government statistic data), CV is also set as 20%. MCD64A1 burned data product has been shown to be reliable in big fires (Giglio et al., 2013), and the CV of burned area of forest and grassland fire is from the reported standard deviation (Giglio et al., 2010). The biomass fuel loadings (Saatchi et al., 2011; Shi et al., 2015) and combustion factors (van der Werf et al., 2010) of forest and grassland fire were within a CV of approximately 50%. The CV of EF for each pollutant for each biomass burning type is shown in the supplement S8 and S9. The range of emissions*

*were calculated by averaging 20000 Monte Carlo simulations with a 95% confidence interval……"*

It could be found from the supplement S8 and S9 that CV of EFs in our paper is generally lower than that in Street et al. (2003). This leads to the narrower uncertainty range of emission estimation than others to some extent.

The author stress the Emission Factors (EFs) in their study. The emission factors indeed have spatial variations with geographical characteristics. I welcome the authors to employ the regional EF to reflect the real condition. All EF actually can only be derived from experiment, the authors just used the EF from literature. Your study China covers a large area with strong geographical characteristics, the EF in different ecosystem located in different climatic zones vary greatly. Only one EF value cannot reflect the real situation in China.

The authors contributed their originality to the other studies by highlighting the employed EF in China, but the EF you used actually has minor differences or we can say they are identical to many studies outside China (Andreae and Merlet, 2001; Akagi et al., 2011). therefore, you cannot conclude assertively at Page 20 Line 3-5 below.

Though the uncertainty exists in this study, compared with the limited research of national and comprehensive emission with uncertainty analysis (Table 8), our emission inventory is relatively reliable due to the selection of localized and specific crop EFs.

Besides, the emission is calculated by using activity data and emission factors. Before excluding the potential different activity data between this study and others, it is irresponsible to judge a conclusion that localized EF is the only reason that contributes your emissions comparable to others.

**Response:**
Thanks very much for your suggestion.

First, in order to develop the comprehensive and detailed biomass burning emission inventory which could reflect the actual situation in China, we have try to take full

account of the domestic and in-field straw burning EFs measured in China for 12 pollutants and the specific biomass burning source (including 12 crop straw) according to the extensive literature review. Based on this premise, it's difficult to further select the region-specific EFs for various areas in China due to the limited measurement. That is why few study on the biomass burning emission inventory is developed based on the EFs with regional difference. With more studies on the EFs measurements in different region in China, the emission inventory could be further improved based on the detailed EFs which could reflect spatial variations with geographical characteristics.

Second, the Table 5 and Table 6 show that most of the literature we mentioned is from the measurement in China and they show obvious difference to the studies outside China. For example, the in-field wheat $EF_{SO2}$ in Li et al. (2007) is 0.85 while is 0.4 in Andreae and Merlet (2001), the domestic rice $EF_{nox}$ in Zhang et al. (2008) is 1.81 while is 1.1 in Andreae and Merlet (2001). We think that the localized EFs selection as much as possible for the various pollutants and the specific biomass burning source is a significant factor for the comprehensive emission inventory improvement.

In addition to the EFs, detailed activity data are also important for a reliable emission inventory, such as detailed activity data with high spatial resolution, the province-specific domestic/in-field straw burning percentage that reflects the status of China in recent years, and so on. The detailed activity data could reduce the uncertainty of emission inventory to some extent because they could reflect the actual situation better. As for the sentence mentioned by the reviewer, we have made the corresponding modification in the revised manuscript in Lines 1-3 on Page 19:

"*…As the detailed activity data could also reduce the uncertainty of emission inventory to some extent because they could reflect the actual situation better, in spite of the uncertainty exists in this study, our emission inventory is relatively reliable due to the selection of localized EFs and the detailed activity data.*"

Figure 7 has a fundamental serious problem. Based on the reply from the authors, we think that. But please refer to the urban center of Beijing, the $PM_{2.5}$ annual emissions is very low, there was scarce open burning, firewood burning. But comparing to other

counties in North China Plain, their amounts are similar to Beijing. If you attribute the low emission in Beijing center to its urban area. That means the counties with low emissions in Shandong, Jiangsu and Anhui provinces are all urban areas, which are larger than Beijing. This is a paradox. The strange results appeared many polygons with obvious administrative boundaries, where the emissions were significantly lower than its surrounding areas. The authors failed in improving this figures and understanding the real meaning.

Please refer to Huang et al. (2011) on biomass burning emissions in China.

**Response:**

Thanks very much for your comment. It should be noted that there are several urban areas surrounded by suburban and rural areas. The main fuel used in these urban areas is commodity energy (e.g., coal, natural gas, and electricity) rather than biomass fuel. There is little crop yield, cultivated land and rural population. Therefore, in Fig. 7, the biomass burning emission of these areas were significantly lower than its surrounding areas, such as the Dongcheng and Xicheng district in Beijing, the Bincheng and Chengyang district in Shandong, the Qixia and Yuhuatai district in Jiangsu, and the Shushan and Yaohai district in Anhui. The central urban areas in Shandong, Jiangsu, and Anhui provinces are indeed larger than those in Beijing.

As for the literature mentioned by the reviewer (Huang et al, 2011), the EFs used for emission estimation were from one literature (Cao et al., 2005). The emission was allocated by the population density. Because the spatial allocation figures in Huang et al. (2011) represent the distribution for the total emission including not only biomass burning but also other source categories, it could not reflect the spatial distribution characteristic of the biomass burning emission. The biomass burning emission in our study were first estimated based on the straw data at county scale and the EFs summarized from local tests in the latest researches, and then allocated to grid cells according to the rural population density. We think the grid emission allocated from an emission inventory with improved preliminary resolution (i.e., county-level resolution) based on the appropriate surrogate could better reflect the actual situation.

The author stressed the updated biomass burning emissions in China. The study period was selected to be 2012. There is no sign to show the reason for 2012 selection. A comprehensive biomass burning emissions inventory lasting for only one year has little implications. China released strict controls and restrictions on its open biomass burning recently with the decreasing amount of crop residue burning from year to year, therefore, only one-year emission inventory has no meaning.

**Response:**

Thanks very much for your comment. As for the novelty and contribution of this study, we have given a detailed statement from several perspectives which could be found in the reply to the first comment from Referee #1 in our former response letter. As for the trends analysis, there are several studies that focused on the inter annual variation of open biomass burning emission (Song et a., 2009; van der Werf et al., 2010; Shon, 2015; Sun et al., 2016).

In Figure 9, the largest monthly variations of the $PM_{2.5}$ were found in in-field straw. Actually the in-field straw has strong seasonal variations peaking during field crop rotation twice or three times within a year. The farmer burns the residue for land clearing for next plantation. That means at least two peaks can be found on emissions. And then the emissions will be very low. But we cannot find the monthly variations in the South.

**Response:**

Thanks very much for your comment. The seasonal variation peaking twice or three times within a year could be found in Figure 9 for the in-field straw burning emission of various regions. As for the south regions, there are three relatively higher in-field straw burning emission occurred in February, April and August than other months. These periods are consistent with local sowing and harvest times in south region. February, April, and August are the sowing season of beans, the harvest season of the first-round and second-round crop (e.g., rice), respectively (CAAS, 1984; MOA, 2000). The corresponding description could be found in Lines 26-27 on Page 16 and Lines 1-4 on Page 17 of the revised manuscript:

[revised manuscript text omitted]

van der Werf, G. R., Randerson, J. T., Giglio, L., Collatz, G. J., Mu, M., Kasibhatla, P.

S., Morton, D. C., DeFries, R. S., Jin, Y., and van Leeuwen, T. T.: Global fire emissions and the contribution of deforestation, savanna, forest, agricultural, and peat fires (1997–2009), Atmos. Chem. Phys., 10, 11707-11735, doi:10.5194/acp-10-11707-2010, 2010.

Zhang, H. F., Ye, X. N., Cheng, T. T., Chen, J. M., Yang, X., Wang, L. and Zhang, R. Y.: A laboratory study of agricultural crop residue combustion in China: Emission factors and emission inventory, Atmos. Environ. 42, 8432−8441, doi: 10.1016/j.atmosenv.2008.08.015, 2008.

Zhang, Y. S., Shao, M., Lin, Y., Luan, S. J., Mao, N., Chen, W. T. and Wang, M.: Emission inventory of carbonaceous pollutants from biomass burning in the Pearl River Delta Region, China, Atmos. Environ., 76, 189−199, doi: 10.1016/j.atmosenv.2012.05.055, 2013.

Zhao, Y., Nielsen, C. P., Lei, Y., McElroy, M. B. and Hao, J.: Quantifying the uncertainties of a bottom−up emission inventory of anthropogenic atmospheric pollutants in China, Atmos. Chem. Phys., 11, 2295−2308, doi: 10.5194/acp-11-2295-2011, 2011.

Zheng, J. Y., Zhang, L. J., Che, W. W., Zheng, Z. Y. and Yin, S. S.: A highly resolved temporal and spatial air pollutant emission inventory for the Pearl River Delta region, China and its uncertainty assessment, Atmos. Environ., 43, 5112−5122, doi: 10.1016/j.atmosenv.2009.04.060, 2009.

**Authors Reply to Referee 2 #**

In many places I found myself a bit confused by the usage of crop straw, unsure if the authors were referring to in-field burning of crop residue or domestic burning. I suggest the authors consistently use "in-field straw" to describe in-field burning of crop straw and consistently use "domestic straw" when referring to domestic burning of crop straw. I imagine the authors use "crop residue" since not all crop residue is straw, e.g. residue from potatoes, beets, peanuts, and cotton. This should be clarified early on for the

readers. Where the author are referring to all in-field burning of crop residue types they should consistently use "in-field crop residue" not "in-field straw".

**Response:**

Thanks very much for your comments, which were very helpful for improving our paper. We are truly sorry about this unclear description. According to the reviewer's advice, we have made the corresponding modification about the "in-field straw" and "domestic straw" through the full text in the revised manuscript. The "straw burning" without specific description means the total straw burning including in-field and domestic burning. In our paper, the straw includes crop stalk and residue. These statements have been clarified in the "General description" in Sect.2.1 (Lines 3-6 on Page 6):

*"The biomass burning considered in this study is mainly divided into two categories, domestic burning and open burning. Domestic burning mainly involves domestic straw (straw burned as fuel indoors), firewood, and livestock excrement (mainly used in pastoral and semi-pastoral areas) burning. Open burning includes in-field straw burning (straw burned as waste outdoors, including crop stalk and residue), forest and grassland fire. Straw burning without specific description in this paper refers to the total straw burning including in-field and domestic straw burning…"*

In the manuscript text the authors need to define all chemical formulas upon their first use, e.g. at

P2, L13 "Active trace gases (e.g., $SO_2$, $NO_x$, NMVOCs, $NH_3$) released from biomass burning are the major precursors of…" and other places in the Introduction.

**Response:**

Thanks very much for your suggestion. We have made the corresponding modification in the revised manuscript in Lines 18-20 on Page 2, Lines 24-26 on Page 2 and Line 1 on Page 3, Line 28 on Page 3 and Lines 1-2 on Page 4, and Lines 12-14 on Page 5:

*Lines 18-20 on Page 2: "Active trace gases (e.g., sulfur dioxide ($SO_2$), nitrogen oxides ($NO_x$), non-methane volatile organic compounds (NMVOCs), ammonia ($NH_3$)) released from biomass burning are the major precursors of secondary*

*inorganic/organic aerosols and tropospheric ozone (O₃) in the atmosphere."*

*Lines 24-26 on Page 2 and Line 1 on Page 3: "Primary particles (e.g., elemental carbon (EC) and organic carbon (OC)) discharged by biomass burning not only impact visibility, but also have an influence on climate due to the positive effects of the absorption of light and cloud condensation (IPCC, 2011). Biomass burning is also a significant source of greenhouse gases such as methane (CH₄) and carbon dioxide (CO₂) (Andreae and Merlet, 2001), which contribute to global warming (Sun et al., 2016)."*

*Line 28 on Page 3 and Lines 1-2 on Page 4: "Zhang et al. (2008) measured CO₂, carbon monoxide (CO), nitric oxide(NO), nitrogen dioxide (NO₂), NOₓ, and PM EFs of rice, wheat, and corn straw. And Wang et al. (2009) launched a study on characteristics of gaseous pollutants from biofuel stoves in China."*

*Lines 12-14 on Page 5: "The gaseous and particulate pollutants examined in this research included SO₂, NOx, particulate matter with a diameter below 10 μm (PM₁₀), particulate matter with a diameter below 2.5 μm (PM₂.₅), NMVOC, NH₃, CO, EC, OC, CO₂, CH₄, and mercury (Hg)."*

Frequent use of "higher" when authors should use "largest" or "highest" or need an explicit comparison, for example:

  P1 L25 "…wheat straw burning has higher contribution to CO and Hg emissions." – "higher" compared to what?

  P1 L26: "Heilongjiang, Shandong, and Henan provinces located in northeast and central-south region of China have higher emissions."

And similar misuse in many other places. These instances need to be corrected.

**Response:**

Thanks very much for your comments. We are truly sorry for the confusing description. We have made the corresponding modification through the full text in the revised manuscript.

Some examples in the revised manuscript are mentioned below:

*Lines 25-27 on Page 1 and Line 1 on Page 2: "…As for the straw burning emission of various crops, corn straw burning has the largest contribution to all of the pollutants*

*considered except for CH$_4$; rice straw burning has highest contribution to CH$_4$ and the second largest contribution to other pollutants except for SO$_2$, OC, and Hg; wheat straw burning is the second largest contributor to SO$_2$, OC, and Hg and the third largest contributor to other pollutants…"*

*Lines 3-4 on Page 2: "Heilongjiang, Shandong, and Henan provinces located in northeast and central-south region of China have higher emissions compared with other provinces in China."*

P1 L21-22: Based on Figure 2, perhaps state that: "Domestic straw burning is the largest source of biomass burning emissions for all the pollutants considered except for NH$_3$, EC (firewood), and NO$_x$ (in-field crop residue)"

**Response:**

We thank you for your suggestion. We have revised the expression in Lines 22-23 on Page 1:

*"Domestic straw burning is the largest source of biomass burning emissions for all the pollutants considered except for NH$_3$, EC (firewood), and NO$_x$ (in-field straw)."*

P1 L 24: "As for the straw burning emission of various crops, corn straw burning has the largest contribution to EC, NO$_x$ and SO$_2$ emissions"

Here any everywhere else specify "in-field" or "domestic" straw burning."

**Response:**

Thanks very much for your comment. According to our reply to the first comment of Referee 2 #, we have made the corresponding modification about the "in-field straw" and "domestic straw" through the full text in the revised manuscript. In addition, the "straw burning" without specific description refers to the total straw burning including in-field and domestic burning. Here is an example of it.

P2, L2: "The temporal distribution shows that April, May, June and October are the top four months with higher emissions, due to the in-field crop residue burning."

This statement is a bit confusing as written and not too useful once deciphered. Change

to something like: "The months of April, May, June and October account for X% of emissions from in-field crop residue burning".

**Response:**

Thanks very much for your comment. We have made the modification in the revised manuscript in Lines 6-7 on Page 2:

*"The months of April, May, June and October account for 65% of emissions from in-field crop residue burning."*

P2, L3: "While as for EC, the emission in February, January, October and December are relatively higher due to the biomass domestic burning in heating season."

See previous comment.

**Response:**

Thanks very much for your comment. We have made the corresponding modification in the revised manuscript in Lines 8-9 on Page 2:

*"While as for EC, the emissions in February, January, October, November and December are relatively higher than other months due to the domestic biomass burning in heating season."*

P2, L3: P2, L4: "There's regional difference in monthly variation due to the diversity of main planted crop and the climate conditions."

"Variation" of what? Please clarify.

**Response:**

Thanks very much for your comment. We have made the corresponding modification in the revised manuscript in Lines 9-10 on Page 2:

*"There's regional difference in monthly variation of emission due to the diversity of main planted crop and the climate conditions."*

P2, 13: Biomass burning emission have an important role in climate system, independent of anthropogenic forcing / change. Suggest delete "change"

**Response:**

Thanks very much for your suggestion. We have made the corresponding modification through the full text in the revised manuscript.

P5, L25: Since E and A are in units of Mg/yr and EF is in units of g/kg, the equation should multiplied by 0.001 (.001 kg per g). This must be a typo as the emission estimates in this study are similar to other inventories (Fig 10) and not high by a factor 100,000.

**Response:**

Thanks very much for your comment. We are truly sorry about this mistake. We have revised the equation in the revised manuscript in Line 12 on Page 6:

$$"E_i = \sum (A_i \times EF_{i,j})/1000 \qquad\qquad (1)$$

P10, L2-4: This first two sentences are awkward and bit confusing. I believe the authors intend something like: "Detailed speciation of NMVOC and PM$_{2.5}$ emissions in necessary to model gas and aerosol chemistry and simulate the impact of biomass burning on atmospheric composition and it has received extensive attention by domestic scholars in recent years (refs…)"

**Response:**

Thanks very much for your suggestion. We have made the corresponding revision in revised manuscript in Lines 14-15 on Page 10:

*"Detailed speciation of NMVOC and PM$_{2.5}$ emissions is necessary to model gas and aerosol chemistry and simulate the impact of biomass burning on atmospheric composition and it has received extensive attention by domestic scholars in recent years (Song et al., 2007; Li et al., 2007c; Liu et al., 2008)."*

P10 L18-19: Based on Figure 2, perhaps rephrase to stress that domestic straw burning is the largest source of biomass burning emissions for all the pollutants considered except for NH$_3$, EC (firewood), and NO$_x$ (in-field crop residue).

**Response:**

Thanks very much for your comment. We have made the corresponding revision in revised manuscript in Lines 9-10 on Page 11:

"*...Domestic straw burning is the largest source of biomass burning emissions for $SO_2$ (57.8%), $PM_{10}$ (42.8%), $PM_{2.5}$ (42.0%), NMVOC (49.2%), CO (58.1%), OC (41.9%), $CO_2$ (38.8%), $CH_4$ (53.2%), and Hg (37.4%)...*"

P11, L15-17: "Among the various crops, corn straw burning has large contribution to all of the chemical species except for $CH_4$. Rice straw has the largest contribution to $CO_2$, NMVOC, $CH_4$ and $NH_3$ emissions, accounting for 32.90%, 32.43%, 31.61% and 30.12%, respectively;"

These statements are contradictory. Which has the larger contribution corn or rice? Should this read "corn straw burning has the largest contribution to all of the chemical species…" and "Rice straw has the second largest contribution to…"?

**Response:**

Thanks very much for your comment. We are truly sorry for the confusing description. We have made the corresponding modification in the revised manuscript in Lines 8-11 on Page 12:

"*Among the various crops, corn straw burning has the largest contribution to all of the pollutants except for $CH_4$. Rice straw burning is the largest contributor to $CH_4$ and the second largest contributor to other pollutants except for $SO_2$, OC, and Hg. Wheat straw burning is the second largest contributor to $SO_2$, OC, and Hg and the third largest contributor to other pollutants.*"

P11, L21-22: "In addition, Fig. 3a and Fig. 3b indicate that for most of the chemical species, the contribution of in-field corn residue burning is larger than that of domestic burning, except for $SO_2$, EC and $CO_2$."

This statement is incorrect. Figure 3a and Figure 3b show percentages within groups (in-field and domestic). The magnitudes of emissions from in-field and domestic cannot be inferred from Fig 3a and Fig 3b. This statement must be supported by a different figure or table.

**Response:**

Thanks very much for your suggestion. We are truly sorry for the unclear description. Through the Fig. 3a and Fig. 3b, we want to show the contribution of each straw burning emission to the in-field and domestic straw burning emission, respectively. We have revised the expression in revised manuscript in Line 16-26 on Page 12:

*"...In addition, Fig. 3a and Fig. 3b show the contribution of each straw burning emission to the in-field and domestic straw burning emission, respectively. Similar to Fig. 3c, corn, rice, and wheat straw are the main contributors whether for in-field or domestic burning emission. However, the dominant contributor of certain pollutants are different in in-field and domestic straw burning: for $SO_2$ and $CO_2$, rice straw is the largest contributor to in-field straw burning emission while corn straw is the largest contributor to domestic straw burning emission; for $NO_x$ and VOC, corn straw contributes most to in-field straw burning emission while rice straw contributes most to domestic straw burning emission; for CO and $CH_4$, corn straw has the largest contribution to in-field straw burning emission while wheat straw has the largest contribution to domestic straw burning emission."*

[Figure]

**Figure 3. Contributions of 12 crop straw types to total straws burning emissions for various species.**

P11, L22-24: "Contrary to that for corn straw, emissions of all chemical species (except for $SO_2$, $NO_x$ and EC) from wheat straw domestic burning is greater than those from in-field crop residue burning. For rice straw, the contribution of in-field crop residue burning to $NO_x$, $PM_{10}$, $PM_{2.5}$, NMVOC, EC and OC emissions is larger than domestic burning."

These statements are cannot be supported by Figure 3. Please refer the reader to figures or tables that support these statements.

**Response:**

Thanks very much for your suggestion. We are truly sorry for the unclear description. Similar to the reply mentioned above, the unclear statements have been revised. The modified content could be found in revised manuscript in Line 16-26 on Page 12:

*"...In addition, Fig. 3a and Fig. 3b show the contribution of each straw burning emission to the in-field and domestic straw burning emission, respectively. Similar to Fig. 3c, corn, rice, and wheat straw are the main contributors whether for in-field or domestic burning emission. However, the dominant contributor of certain pollutants are different in in-field and domestic straw burning: for $SO_2$ and $CO_2$, rice straw is the largest contributor to in-field straw burning emission while corn straw is the largest contributor to domestic straw burning emission; for $NO_x$ and VOC, corn straw contributes most to in-field straw burning emission while rice straw contributes most to domestic straw burning emission; for CO and $CH_4$, corn straw has the largest contribution to in-field straw burning emission while wheat straw has the largest contribution to domestic straw burning emission."*

[Figure]

**Figure 3. Contributions of 12 crop straw types to total straws burning emissions for various species.**

*Technical*

*The manuscript has many instances where English usage needs to be improved. Here is a list of some, but not all instances.*

*P3, L1: insert "frequently" between "more" and "burned"*

*P3, L10: insert "associated" between "the" and "environmental"*

*P3, L12: change "inventory" to "inventories"*

*P3, L24: spelling, change "in-filed" to "in-field"*

*P6, L22: Maybe change "investigation" to "review"*

*P6, L22: change "collect" to "estimate" or "derive"*

*P9, L3: change "research" to "review"*

*P9, L6: insert "burning" between "biomass" and "emission" and change "emission" to "emissions"*

*P10, L5: new paragraph not needed.*

*P10, L16: delete "the" before "domestic straw…"*

*P11, L10: delete "change" it is not needed, see earlier comment for P2, L13.*

*P12, L8: change "shown" to "provided"*

*P14, L 7: Change "The most of high values" to "Most of the high values"*

*P15, L6: Change "Figure 8 shows the 12 species emissions in each month…" to "Figure 8 shows the monthly emissions of all 12 species considered"*

*P15, L7-8: "Besides, the in-field burning of crop residue mainly in the harvest season and thus shows the obvious monthly variation features."*

*This statement is unclear and needs to be rewritten. Do the authors mean to say that "the monthly variability in in-field emissions reflects the timing of the harvest seasons"?*

*P16, L16: change "The total PM2.5 emission of biomass burning emission …" to "Total PM2.5 emissions from biomass burning…"*

*P18, L11: change "The emission involves…" to "The emission inventory includes…"*

*Figure 2 caption: add "s" "source" and delete "the"*

*Figure 6d caption is awkward change to something like: "The distribution of county level annual $PM_{2.5}$ emissions"*

**Response:**

Thanks very much for your careful review and suggestion. We have made the corresponding modification through the full text in the revised manuscript. The details could be found in the revised manuscript.

**Reference**

[revised manuscript text omitted]

**Section S8:** Table S1 CV (coefficients of variation) of biomass domestic burning emission factors.

**Section S9:** Table S2 CV (coefficients of variation) of biomass open burning emission factors

**S1 The correlation between crop yield and grain yield at prefecture resolution.**

[Figure]

**Figure S1 The correlation between crop yield and grain yield at prefecture resolution.**

Note: [a] NBSC (2013); [b] a range of statistical yearbooks edited by National Bureau of Statistics in 2012 for each province.

**S2 Maps showing the prefecture and county resolution.**

[Figure]

| | | | |
|---|---|---|---|
| 1. | Beijing | 17. | Hubei |
| 2. | Tianjin | 18. | Hunan |
| 3. | Hebei | 19. | Guangdong |
| 4. | Shanxi | 20. | Guangxi |
| 5. | Inner-Mongolia | 21. | Hainan |
| 6. | Liaoning | 22. | Chongqing |
| 7. | Jilin | 23. | Sichuan |
| 8. | Heilongjiang | 24. | Guizhou |
| 9. | Shanghai | 25. | Yunnan |
| 10. | Jiangsu | 26. | Tibet |
| 11. | Zhejiang | 27. | Shaanxi |
| 12. | Anhui | 28. | Gansu |
| 13. | Fujian | 29. | Qinghai |
| 14. | Jiangxi | 30. | Ningxia |
| 15. | Shandong | 31. | Xinjiang |
| 16. | Henan | | |

(a) Province level

(b) Prefecture level

(c) County level

Figure S2 Maps showing the prefecture and county resolution.

**S3 The details about questionnaire field survey.**

A questionnaire was designed to conduct field investigation during face-to-face interviews with rural resident, in order to obtain the percentage of domestic and in-field straw burning and uneven temporal distribution coefficient in several provinces with limited literature reports, including Tianjin, Hebei, Inner Mongolia, Heilongjiang, Shanghai, Zhejiang, Anhui, Jiangxi and Guangdong provinces. Respondents need to provide the detailed address, main cultivated crop type. They selected from a list of cooking and heating fuels, including specific crop straw, firewood, coal, gas, electricity or solar, livestock excrement and other detailed fuels not existing in the list. They also need to provide approximate proportion of crop straw domestic combustion and in field burning, and selected the month of burning the straw as waste, and heating period. The investigation was launched in the representative regions in each province mentioned above, with the integrative consideration about the geographical location, economic development level and population intensity. All the surveyors were trained and tested in their understanding of the questionnaire content. Ultimately, we received 2478 valid questionnaire responses, and at least 200 valid questionnaires in each province.

**S4 The detailed description about the MODIS fire data and calculation method and equation of gridded emission**

**4.1 Detailed description about the MODIS fire data**

For the spatiotemporal distributions of biomass open burning, satellite remote sensing has excellent characteristics of wide coverage, high resolution and strong temporal reliability. As a result, satellite remote sensing has been increasingly applied to solving temporal and spatial emission distributions in recent years. The MODIS satellite fire data were taken from FIRM (Fire Information for Resource Management System). The MODIS Thermal Anomalies/Fire 5-Min L2 Swath Product (MOD14/MYD14) within 1km resolution was used in this study. The MOD14 were provided by the Terra satellite with overpass times at 10:30 AM and 10:30 PM local time, while MYD14 were provided by Aqua at 1:30 AM and 1:30 PM local time.

**4.2 Detailed calculation method and equation of gridded emission**

The mass of biomass emission in each grid of biomass open burning and domestic burning was calculated using Eqs. (1) and (2), respectively, as follows:

$$E_{m-open} = \frac{FC_m}{FC_n} \times E_{n-open} \tag{1}$$

$$E_{m-domestic} = \frac{PO_m}{PO_n} \times E_{n-domestic} \tag{2}$$

where $m$ is the $m$-th grid and $n$ represents the $n$-th county; $E_{m-open}$ and $E_{n-open}$ represent the emissions of the $m$-th grid and $n$-th county for biomass open burning (in-field straw burning), respectively; $E_{m-domestic}$ and $E_{n-domestic}$ represent the emissions of the $m$-th grid and $n$-th county for biomass domestic burning, respectively; $FC_m$ represents the number of typical fire points of the $m$-th grid; $FC_n$ is the number of total typical fire points of the $n$-th county; $PO_m$ is the number of typical population of the $m$-th grid; finally, $PO_n$ is the number of typical population of the $n$-th county.

**S5 Daily PM₂.₅ biomass burning emissions variation in 2012.**

[Figure]

**Figure S3 Daily PM₂.₅ biomass burning emissions variation in 2012.**

**S6 Emission of PM2.5 species from biomass burning**

[Figure]

**Figure S4 Emission of PM2.5 species from biomass burning.**

Note: Species in others include Al,Si,Mg,Fe,Pb,Zn,Ba,Ti,Ni,Cr,Mn,Sr,V,Cd,As,Zr,Se,Ag,Sb,Sc,Mo,Ga,Tl,Co and Hg. PM2.5 speciation profile is obtained from Li et al., (2007) and Waston et al., (2001).

**S7 Emission of NMVOC species from biomass burning.**

[Figure]

**Figure S5 Emission of NMVOC species from biomass burning.**

Note: *Species in others include aldehyde, ethers, alcohols, esters, ketone and acids.

**S8 Table S1 CV (coefficients of variation) of biomass domestic burning emission factors.**

| | Material | $SO_2$ | $NO_x$ | $PM_{10}$ | $PM_{2.5}$ | NMVOC | $NH_3$ | CO | EC | OC | $CO_2$ | $CH_4$ | Hg |
|---|---|---|---|---|---|---|---|---|---|---|---|---|---|
| Domestic burning | Corn | 0.5[*] | 0.02[a] | 0.5[*] | 0.27[b] | 0.5[*] | 0.5[*] | 0.85[a] | 0.34[b] | 0.44[b] | 0.04[a] | 0.5[*] | 0.05[c] |
| | Wheat | 0.5[*] | 0.16[a] | 0.5[*] | 0.23[b] | 0.5[*] | 0.5[*] | 0.89[a] | 0.76[b] | 0.29[b] | 0.07[a] | 0.5[*] | 0.12[c] |
| | Cotton | 0.5[*] | 0.5[*] | 0.5[*] | 0.26[b] | 0.5[*] | 0.5[*] | 0.5[*] | 0.39[b] | 0.55[b] | 0.5[*] | 0.5[*] | 0.33[c] |
| | Cane | 0.5[*] | 0.5[*] | 0.5[*] | 0.26[b] | 0.5[*] | 0.5[*] | 0.5[*] | 0.63[b] | 0.45[b] | 0.5[*] | 0.5[*] | 0.32[c] |
| | Potato | 0.5[*] | 0.5[*] | 0.5[*] | 0.26[b] | 0.5[*] | 0.5[*] | 0.5[*] | 0.63[b] | 0.45[b] | 0.5[*] | 0.5[*] | 0.53[c] |
| | Peanut | 0.5[*] | 0.5[*] | 0.5[*] | 0.26[b] | 0.5[*] | 0.5[*] | 0.5[*] | 0.63[b] | 0.45[b] | 0.5[*] | 0.5[*] | 0.03[c] |
| | Rape | 0.5[*] | 1.21[d] | 0.5[*] | 0.15[b] | 0.26[d] | 0.5[*] | 0.26[d] | 0.63[b] | 0.45[b] | 0.5[*] | 0.5[d] | 0.3[c] |
| | Sesame | 0.5[*] | 1.78[d] | 0.5[*] | 0.26[b] | 0.24[d] | 0.5[*] | 0.29[d] | 0.63[b] | 0.45[b] | 0.5[*] | 0.5[*] | 0.3[c] |
| | Beet | 0.5[*] | 0.5[*] | 0.5[*] | 0.26[b] | 0.5[*] | 0.5[*] | 0.5[*] | 0.63[b] | 0.45[b] | 0.5[*] | 0.5[*] | 0.3[c] |
| | Hemp | 0.5[*] | 0.5[*] | 0.5[*] | 0.26[b] | 0.5[*] | 0.5[*] | 0.5[*] | 0.63[b] | 0.45[b] | 0.5[*] | 0.5[*] | 0.3[c] |
| | Rice | 0.5[*] | 0.05[a] | 0.5[*] | 0.29[b] | 0.5[*] | 0.5[*] | 0.06[a] | 0.65[b] | 0.5[b] | 0.01[a] | 0.5[*] | 0.46[c] |
| | Soybean | 0.5[*] | 1.78[d] | 0.5[*] | 0.26[b] | 0.76[d] | 0.5[*] | 0.44[d] | 0.63[b] | 0.45[b] | 0.5[*] | 0.5[*] | 0.74[c] |
| | Firewood | 0.5[*] | 1.42[d] | 0.5[*] | 0.16[b] | 0.15[d] | 0.5[*] | 0.39[d] | 0.46[b] | 0.35[b] | 0.5[*] | 0.5[*] | 1.17[c] |
| | Feces | 0.8[*] | 0.8[*] | 0.8[*] | 0.8[*] | 0.8[*] | 0.8[*] | 0.8[*] | 0.8[*] | 0.8[*] | 0.8[*] | 0.8[*] | 0.8[*] |

Note: Lowercase letters indicate the data source.

Sources are from the following: [a] Zhang et al. (2008). [b] Li et al. (2009). [c] Chen et al. (2013). [d] Zhang et al. (2013). [*] Expert judgment data from Wei et al. (2011).

**S9 Table S2 CV (coefficients of variation) of biomass open burning emission factors**

| | Material | $SO_2$ | $NO_x$ | $PM_{10}$ | $PM_{2.5}$ | NMVOC | $NH_3$ | CO | EC | OC | $CO_2$ | $CH_4$ | Hg |
|---|---|---|---|---|---|---|---|---|---|---|---|---|---|
| Open burning | Corn | 0.45[b] | 0.42[b] | 0.5[*] | 0.09[b] | 0.53[b] | 0.76[b] | 0.08[b] | 0.33[b] | 0.39[b] | 0.01[b] | 0.22[b] | 0.05[a] |
| | Wheat | 0.67[b] | 0.52[b] | 0.5[*] | 0.54[b] | 0.25[b] | 0.38[b] | 0.41[b] | 0.32[b] | 0.26[b] | 0.03[b] | 0.25[b] | 0.12[a] |
| | Cotton | 0.5[*] | 0.5[*] | 0.5[*] | 0.5[*] | 0.5[*] | 0.5[*] | 0.5[*] | 0.5[*] | 0.5[*] | 0.5[*] | 0.5[*] | 0.33[a] |
| | Cane | 0.5[*] | 0.32[d] | 0.19[d] | 0.16[d] | 0.71[d] | 0.5[*] | 0.61[d] | 1.57[d] | 0.2[d] | 0.18[d] | 0.5[*] | 0.32[a] |
| | Potato | 0.5[*] | 0.5[*] | 0.5[*] | 0.5[*] | 0.5[*] | 0.5[*] | 0.5[*] | 0.5[*] | 0.5[*] | 0.5[*] | 0.5[*] | 0.53[a] |
| | Peanut | 0.5[*] | 0.5[*] | 0.5[*] | 0.5[*] | 0.5[*] | 0.5[*] | 0.5[*] | 0.5[*] | 0.5[*] | 0.5[*] | 0.5[*] | 0.03[a] |
| | Rape | 0.5[*] | 0.5[*] | 0.5[*] | 0.5[*] | 0.5[*] | 0.5[*] | 0.5[*] | 0.5[*] | 0.5[*] | 0.5[*] | 0.5[*] | 0.3[a] |
| | Sesame | 0.5[*] | 0.5[*] | 0.5[*] | 0.5[*] | 0.5[*] | 0.5[*] | 0.5[*] | 0.5[*] | 0.5[*] | 0.5[*] | 0.5[*] | 0.3[a] |
| | Beet | 0.5[*] | 0.5[*] | 0.5[*] | 0.5[*] | 0.5[*] | 0.5[*] | 0.5[*] | 0.5[*] | 0.5[*] | 0.5[*] | 0.5[*] | 0.3[a] |
| | Hemp | 0.5[*] | 0.5[*] | 0.5[*] | 0.5[*] | 0.5[*] | 0.5[*] | 0.5[*] | 0.5[*] | 0.5[*] | 0.5[*] | 0.5[*] | 0.3[a] |
| | Rice | 0.5[*] | 0.8[d] | 0.88[d] | 0.17[d] | 0.75[d] | 0.5[*] | 1.19[d] | 1.38[d] | 1.53[d] | 0.14[d] | 0.5[*] | 0.46[a] |
| | Soybean | 0.5[*] | 0.5[*] | 0.5[*] | 0.5[*] | 0.5[*] | 0.5[*] | 0.5[*] | 0.5[*] | 0.5[*] | 0.5[*] | 0.5[*] | 0.74[a] |
| | Evergreen Needleleaf Forest | 0.3[c] | 0.39[c] | 0.25[d] | 0.25[d] | 0.31[e] | 0.66[e] | 0.38[e] | 1[f] | 0.62[f] | 0.08[e] | 0.52[e] | 0.52[g] |
| | Evergreen Broadleaf Forest | 0.4[e] | 0.54[e] | 0.25[d] | 0.25[d] | - | 1.58[h] | 0.29[e] | 0.6[e] | 0.57[e] | 0.04[e] | 0.39[e] | 0.52[g] |
| | Deciduous Needleleaf Forest | 0.3[c] | 0.23[c] | 0.25[d] | 0.25[d] | 0.31[e] | 0.66[e] | 0.38[r] | 1[f] | 0.62[f] | 0.08[e] | 0.52[r] | 0.52[g] |
| | Deciduous Broadleaf Forest | 0.3[c] | 0.46[e] | 0.25[d] | 0.25[d] | 0.79[e] | 0.27[e] | 0.19[e] | 0.33[e] | 0.52[e] | 0.02[e] | 0.18[e] | 0.52[g] |
| | Mixed Forest | 0.3[c] | 0.46[e] | 0.25[d] | 0.25[d] | 0.62[e] | 0.27[e] | 0.19[e] | 0.33[e] | 0.52[e] | 0.02[e] | 0.18[e] | 0.52[g] |
| | Closed Shrublands | 0.44[e] | 0.21[e] | 0.25[d] | 0.25[d] | 0.48[e] | 0.33[e] | 0.25[e] | 0.4[f] | 0.18[f] | 0.02[e] | 0.35[e] | 0.74[h,g] |
| | Open Shrublands | 0.44[e] | 0.21[e] | 0.25[d] | 0.25[d] | 0.48[e] | 0.33[e] | 0.25[e] | 0.4[f] | 0.18[f] | 0.02[e] | 0.35[e] | 0.74[h,g] |
| | Woody Savannas | 0.44[e] | 0.21[e] | 0.25[d] | 0.25[d] | 0.48[e] | 0.33[e] | 0.25[e] | 0.4[f] | 0.18[f] | 0.02[e] | 0.35[e] | 0.52[h] |
| | Savannas | 0.63[e] | 0.29[e] | 0.25[d] | 0.25[d] | 0.25[e] | 0.8[e] | 0.29[e] | 0.5[e] | 0.46[e] | 0.02[e] | 0.6[e] | 0.52[h] |
| | Grasslands | 0.63[e] | 0.29[e] | 0.25[d] | 0.25[d] | 0.25[e] | 0.8[e] | 0.29[e] | 0.5[e] | 0.46[e] | 0.02[e] | 0.6[e] | 0.52[h] |

Note: Lowercase letters indicate the data source.

Sources are from the following: [a] Chen et al. (2013). [b] Li et al. (2007). [c] Andreae and Rosenfeld (2008). [d] Song et al. (2009). [e] Akagi et al. (2011). [f] McMeekin et al. (2008). [g] Friedli et al. (2003). [h]

Streets et al. (2005). * Expert judgment data from Wei et al. (2011).